# Why Differentially-Private Local SGD
# – An Analysis of Biased Synchronized-Only Iterates

## Abstract

We argue to use Differentially-Private Local Stochastic Gradient Descent (DP-LSGD) in both centralized and distributed setups, and explain why DP-LSGD enjoys higher clipping efficiency and produces less clipping bias compared to classic Differentially-Private Stochastic Gradient Descent (DP-SGD). For both convex and non-convex optimization, we present generic analysis on noisy synchronized-only iterates in LSGD, the building block of federated learning, and study its applications to differentially-private gradient methods with clipping-based sensitivity control. We point out that given the current *decompose-then-compose* framework, there is no essential gap between the privacy analysis of centralized and distributed learning, and DP-SGD is a special case of DP-LSGD. We thus build a unified framework to characterize the clipping bias via the second moment of local updates, which initiates a direction to systematically instruct DP optimization by variance reduction. We show DP-LSGD with multiple local iterations can produce more concentrated local updates and then enables a more efficient exploitation of the clipping budget with a better utility-privacy tradeoff. In addition, we prove that DP-LSGD can converge faster to a small neighborhood of global/local optimum compared to regular DP-SGD. Thorough experiments on practical deep learning tasks are provided to support our developed theory.

## 1 Introduction

Local Stochastic Gradient Descent (LSGD) [1, 2] and (Local/Client-Level) Differential Privacy (DP) [3, 4, 5] are two popular methods to address the issues of communication efficiency and data privacy, respectively. Rooted in the *FedAvg* framework first proposed in [6], instead of communicating and synchronizing on the local updates from each user at each iteration, LSGD [1] randomly samples participants to perform gradient descent on their local data in parallel and only aggregates their local updates periodically. Though LSGD is a simple generalization of SGD to a distributed setup with a lower synchronization frequency, empirically it is known to produce promising performance, with regard to both communication efficiency and convergence rate [7]. When each user holds i.i.d. data, LSGD provably achieves a linear speedup in the number of users with also asymptotic improvements on the communication overhead over regular distributed SGD to produce equivalent accuracy [1, 2].

As for privacy preservation, DP [3, 8] provides a semantically precise way to quantify the data leakage from any processing. At a high level, DP is an input-independent guarantee which ensures that an adversary cannot infer the participation of an individual datapoint easily from the release. For example, the classic $(\epsilon, \delta)$-DP with small security parameters $\epsilon$ and $\delta$ implies a large Type I or Type II error for an adversarial hypothesis testing to guess whether an arbitrary individual is involved in the processing [9]. In DP research, one key problem is to determine the *sensitivity*, the worst-case influence/change on the output of the objective processing after arbitrarily replacing an individual in an input set. Only

with tractable sensitivity, one can then apply proper randomization/perturbation such as the Gaussian or Laplace mechanism [10] to produce required security parameters. Unfortunately, sensitivity is in general NP-hard to compute [11]. To this end, in practice, a commonly-applied alternative is the *decompose-then-compose* framework: a complicated processing is first (approximately) decomposed into several simpler (possibly adaptive) subroutines such as mean estimation, each of whose sensitivity is controllable. A *white-box* adversary is then assumed who can observe the intermediate computations, and an upper bound on the privacy loss is derived by the composition of the leakage from the virtual release in each step [12].

In the applications of machine learning, where the processing function returns a model trained on possibly sensitive data, arguably the most popular and generic DP privatization method is DP-SGD [13, 14]. As a representative of the above-mentioned decompose-then-compose framework, DP-SGD views the SGD as a sequence of adaptive gradient mean estimations. To ensure a bounded sensitivity guarantee, each per-sample gradient is clipped, usually, in $l_2$-norm [14] to some constant $c$, which is essentially a projection to an $l_2$-norm ball of radius $c$. Noise, which is determined by both the number of iterations $T$ and the clipping threshold $c$ (sensitivity bound), is then added to the clipped stochastic gradient in each iteration to produce satisfied DP parameters $(\epsilon, \delta)$ under $T$-fold composition. A wider dimension and a longer convergence time $T$ will consequently require a larger DP noise. Though the implementation of DP-SGD does not require any additional assumptions on either model or training data, it is notorious for heavy utility loss, especially for deep learning. Moreover, the understanding of the clipping bias from this artificial sensitivity control remains limited. In general, due to the bias, clipped SGD will not converge even without noise perturbation [15, 16].

Given the artificial assumption that DP-SGD releases the intermediate computations, there is no essential gap between the privacy analysis of the centralized and local SGD, except that in the distributed setup one may apply different DP metrics such as Local DP (LDP) [4] or client-level DP [5] to consider the privacy preservation for each user's local data. More interestingly, it is worth noting the connection among different problems in federated learning and DP-SGD that are essentially equivalent. First, it is not hard to see that DP-SGD is a special case of DP-LSGD. DP-SGD can be viewed as: $n$ nodes, each holds a sample, and a virtual server collects the clipped stochastic gradient from a subset of sampled nodes *in every iteration*, and publishes a noisy gradient descent. DP-LSGD can be similarly defined where the only difference is that the server may not synchronize on each iteration, but clips and aggregates a linear combinations of local gradients, *periodically*. Thus, as a primary concern in federated learning, a smaller communication overhead in a lower synchronization/aggregation frequency would also imply less leakage and a smaller composition bound of privacy loss. On the other hand, the study on the utility loss by perturbation and artificial sensitivity control (clipping) could also be used to analyze federated learning with compressed communication [17] where there exists quantification error in broadcasted local updates. Therefore, in this paper, we aim to provide a unified analysis for both noisy LSGD and DP-LSGD/SGD to get new insights. Before we can build useful theory to capture these concerns from different perspectives, several technical challenges need to be addressed.

**Utility of "Synchronized/Published" Iterate Only**: Many existing convergence results [2, 18, 19, 20, 21] on non-private LSGD are developed on the (weighted) average of all iterates. These include the intermediate iterates produced during the local updates from each user/node, which will not be exposed or shared. To properly characterize the effect of perturbation, a more appropriate and realistic convergence guarantee is to measure the performance of synchronized (shared) iterates only. This is also important to help understand the practical performance of LSGD as neither the server nor users have access to all intermediate computations. Such measurement is especially necessary when we apply LSGD in a private version: the utility of concern is only with respect to the released outputs, and anything assumed to be published would incur privacy loss and increase the scale of DP noise.

**Clipping Bias and Data Heterogeneity**: In practice, tight sensitivity of many data processing algorithms is intractable and thus a very popular but artificial control is clipping. However, clipping could also bring non-negligible bias. In general, there is no convergence guarantee for clipped SGD if we only assume the stochastic gradient is of bounded variance [15], though under more restrictive assumptions, for example, when the stochastic gradient is in a symmetric [15] or light-tailed [22] distribution, or provided generalized smoothness [23], some (near) convergence results are known. A concise characterization of such clipping bias still largely remains open, especially for deep learning. The bias is even more complicated in the more general DP-LSGD. To provide meaningful theory to instruct systematic bias reduction, we do not want to assume Lipschitz continuity or bounded

gradient, which may make the analysis trivial and impractical. Thus, the desired analysis essentially captures the scenario given heavy data heterogeneity, and the results should not require a bounded difference among the local updates.

In this paper, through tackling the above-mentioned challenges, we aim to provide useful and intuitive theory to understand practical performance of LSGD and instruct optimization with DP guarantees. In particular, we want to explain how DP-LSGD out-performs regular DP-SGD. We summarize our contributions as follows.

1. With only a mild assumption that the stochastic gradient is of bounded variance, we present the convergence analysis on the released-only iterates of LSGD under perturbation for both convex and non-convex smooth optimization in Theorem 3.1 and 3.2. In particular, for the general convex case, we show more powerful last iterate convergence, which could be of independent interest in developing generic last-iterate analysis with unbounded gradients.

2. We then generalize our results to study the utility of DP-LSGD, where DP-SGD becomes a special case. In particular, we use the incremental norm of local update (see Definition 4.1) to characterize the clipping bias and show DP-LSGD has a faster convergence rate to a small neighborhood of global/local optimum as compared to DP-SGD.

3. We further show LSGD behaves as an efficient variance reduction of local update, where multiple local GDs with a small learning rate cancel out substantial sampling noise, and enable more efficient clipping compared to DP-SGD. Thorough experiments show that DP-LSGD produces a much sharpened utility-privacy tradeoff in practical deep learning.

## 1.1 Related Works

**Convergence Analysis of LSGD**: With the increasing scale of both training data and models, federated learning has become an important paradigm in modern machine learning, where LSGD and its variants form the building block. Though the idea of LSGD can be traced back to earlier works [24, 25], the theoretical convergence analysis has only been proved recently. A common strategy to show convergence is to consider a virtual average of all the intermediate iterates produced by each user, and keep track of the divergence (dissimilarity) between the virtual average and the local iterate. In the setup where each user holds i.i.d. data, Stich in [1] studied strongly-convex optimization with LSGD and showed a linear speedup in the number of users/nodes. [26] presented non-convex analysis under the Lipschitz continuity assumption where the divergence of local update is also bounded.

For the more general applications with heterogeneous data, [27] studied the convex case with local GD (without sampling on either users or users' local data) but still under Lipschitz continuity. [2] presented more generic and tighter analysis for LSGD without assumptions on bounded gradient for both strongly and general convex optimization. Further generalization of LSGD to the decentralized setup under arbitrary network topology was considered in [19, 28, 29]. However, many existing works [2, 19, 28] only showed the convergence rate relying on all the intermediate averages. To our knowledge, the first generic analysis for synchronized-only iterates was shown in [30]. [30] proposed Scaffold, a generalized LSGD with careful correction on the client-drift caused by data heterogeneity. Compared to existing works, in this paper, we prove more powerful last-iterate analysis for general convex optimization with clipping and perturbation for privacy. It is also worth mentioning that with a different motivation, there is another line of works also studying noisy LSGD to capture the effect of compressed local updates to further save the communication cost. But, in most existing related works [17, 31], the compression error is assumed to be independent with zero-mean. As we need to study DP-LSGD with clipped local update, which introduces bias in the local update generation, in this paper we present more involved analysis to handle such adaptive and biased perturbation.

**Convergence Analysis of DP-SGD and DP-LSGD**: Asymptotically, under Lipschitz continuity, DP-SGD is known to produce a tight utility-privacy tradeoff [32, 33], where no bias is produced given a clipping threshold larger than the Lipschitz constant. However, without Lipschitz continuity, practical understanding of DP-SGD remains limited. On one hand, negative examples are shown in [15, 16] where clipped-SGD in general will not converge, and in practice clipped-SGD does produce bias and has a lower convergence rate, especially in deep learning applications compared to regular SGD [16]. On the other hand, under more restrictive assumptions on the stochastic gradient distribution, clipped-SGD can be shown to (nearly) converge [15, 22, 23]. A generic characterization on the clipping bias still largely remains open. As a consequence, there is little known meaningful theory to

148 systematically instruct optimization algorithms with DP guarantees, and most existing private deep
149 learning works are empirical, which aim to search for the optimal model and hyperparameters for
150 objective training data [34, 35, 36]. As for DP-LSGD, to our knowledge the only known theoretical
151 result that captures the clipping bias is [16]. However, [16] still assumes globally bounded gradient
152 compared to bounded second moment as assumed in our results, and its main motivation is to study
153 the clipping effect in client-level DP. In this paper, we show more intuitive and generic analysis of
154 DP-LSGD for both convex and non-convex optimization, and our motivations are also very different.
155 We set out to provide usable quantification on the utility loss due to clipping and *we argue to apply*
156 *DP-LSGD both in the centralized and distributed setup*, since DP-LSGD can significantly reduce the
157 clipping bias with a faster convergence rate.

## 2 Preliminaries

159 We focus on the classic Empirical Risk Minimization (ERM) problem. Given a dataset $\mathcal{D} =$
160 $\{(x_i, y_i), i = 1, 2, \cdots, n\}$, the loss function is defined as $F(w) = \frac{1}{n} \cdot \sum_{i=1}^{n} f(w, x_i, y_i) = \frac{1}{n} \cdot$
161 $\sum_{i=1}^{n} f_i(w)$. We will consider the cases where the loss function $f_i(w) : \mathcal{W} \to \mathbb{R}^+$ is convex or
162 non-convex. $w^* = \arg\min_w F(w)$ represents the global optimum. Some formal definitions about
163 the properties of the objective loss function are defined as follows.

**Definition 2.1** (Smoothness). *A function $f$ is $\beta$-smooth on $\mathcal{W}$ if the gradient $\nabla f(w)$ is $\beta$-Lipschitz*
165 *such that for all $w, w' \in \mathcal{W}$, $\|\nabla f(w) - \nabla f(w')\| \leq \beta \|w' - w\|$.*

**Definition 2.2** (Convexity and Strong Convexity). *A function $f(w)$ is $\lambda$-convex on $\mathcal{W}$ if for all*
167 $w, w' \in \mathcal{W}$, $\frac{\lambda}{2} \|w - w'\|^2 \leq f(w) - f(w') - \langle \nabla f(w'), w - w' \rangle$. *We call $f(w)$ general convex if*
168 $\lambda = 0$, *and $f(w)$ is strongly convex if $\lambda > 0$.*

**Assumption 2.1** (Bounded Variance of Stochastic Gradient). *For any $w \in \mathcal{W}$ and an index $i$ that is*
170 *randomly selected from $\{1, 2, \cdots, n\}$, there exists $\tau > 0$ such that $\mathbb{E}[\|\nabla F(w) - \nabla f_i(w)\|^2] \leq \tau$.*

171 Assumption 2.1 is the only additional assumption we need for the analysis of non-private LSGD
172 without clipping. We formally present the non-private LSGD algorithm in Algorithm 1 which uses
173 non-clipped local update (3). The whole process is formed of $T$ phases. In each phase, by $q$-Poisson
174 sampling, in expectation $(nq)$ many users will be selected to perform $K$ local gradient descents
175 on their local data before broadcasting the local update. To match the DP-LSGD where the local
176 function $f_i(w)$ held by each user may only be determined by a single datapoint, we do not consider
177 an additional stochastic gradient oracle on the local function in Algorithm 1, but only assume random
178 sampling on the user level at each phase. However, our results can be easily generalized to the
179 scenario with stochastic local gradient. Moreover, we assume Poisson sampling in Algorithm 1 so as
180 to match the setup of DP-LSGD, since given current studies on privacy amplification by sampling,
181 Poisson sampling can produce the tightest results [37] (and has become the most popular option in
182 practice [36, 38]). In the following, we introduce the definition of DP.

**Definition 2.3** (Differential Privacy [38]). *Given a universe $\mathcal{X}^*$, we say that two datasets $X, X' \subseteq \mathcal{X}^*$*
184 *are adjacent, denoted as $X \sim X'$, if $X = X' \cup x$ or $X' = X \cup x$ for some additional datapoint*
185 $x \in \mathcal{X}$. *A randomized algorithm $\mathcal{M}$ is said to be $(\epsilon, \delta)$-differentially-private (DP) if for any pair of*
186 *adjacent datasets $X, X'$ and any event set $O$ in the output domain of $\mathcal{M}$, it holds that*

$$\mathbb{P}(\mathcal{M}(X) \in O) \leq e^\epsilon \cdot \mathbb{P}(\mathcal{M}(X') \in O) + \delta.$$

187 In Definition 2.3, we apply the unbounded DP definition as adopted in most existing DP-SGD works
188 [16, 35, 38], where the two adjacent datasets are defined to differ in one datapoint. One may also
189 apply the bounded DP definition [8] by defining the adjacent datasets as arbitrarily replacing a
190 datapoint. However, as a stronger definition, bounded DP will also face a larger sensitivity bound.

191 We can now formally describe DP-LSGD and DP-SGD. In (2) of Algorithm 1, a clipping operation
192 on a vector $v$ with threshold $c$ is defined as $\mathcal{CP}(v, c) = v \cdot \min\{1, c/\|v\|\}$, which ensures a bounded
193 sensitivity up to $c$. Using the clipped local update (2), by selecting $Q^{(t)}$ to be proper DP noise,
194 Algorithm 1 captures DP-SGD when $K = 1$ and DP-LSGD for general $K \geq 1$. DP-LSGD (SGD) is
195 essentially an LSGD (SGD) with clipped local update (per-sample gradient) and additional DP noise.
196 Running for $T$ iterations with a total privacy budget $(\epsilon, \delta)$, one may select $Q^{(t)} \sim \mathcal{N}(0, \sigma^2 \cdot \boldsymbol{I}_d)$
197 where $\sigma = \tilde{O}(qc\sqrt{T \log(1/\delta)}/\epsilon)$ by the composition bound [38]. The privacy analysis and the noise
198 bound are identical for both DP-LSGD and DP-SGD given the same clipping threshold $c$.

---
**Algorithm 1** (Differentially Private) Local SGD with Noisy (Clipped) Periodic Averaging
---
1: **Input:** A system of $n$ workers where each holds a local loss function $F(w) = f_i(w)$, sampling rate $q$, update step size $\eta$, local update length $K$ and global synchronization number $T$, clipping threshold $c$, and initialization $\bar{w}^{(0)}$ with synchronization noise $Q^{(1:T)}$.
2: **for** $t = 1, 2, \cdots, T$ **do**
3:     Implement i.i.d. sampling to select an index batch $S^{(t)} = \{[1], \cdots, [B_t]\}$ from $\{1, 2, \cdots, n\}$ of size $B_t$.
4:     **for** $i = 1, 2, \cdots, B_t$ in parallel **do**
5:         $w_{[i]}^{(t,0)} = \bar{w}^{(t-1)}$.
6:         **for** $k = 1, 2, \cdots, K$ **do**
7:

$$w_{[i]}^{(t,k)} = w_{[i]}^{(t,k-1)} - \eta \nabla f_{[i]}(w_{[i]}^{(t,k-1)}). \tag{1}$$

8:         **end for**
9:         Clip the local update as $\Delta w_{[i]}^{(t)} = \mathcal{CP}(w_{[i]}^{(t,K)} - \bar{w}^{(t-1)}, c)$
10:     **end for**
11:     **if** to ensure Differential Privacy with clipping **then**
12:

$$\bar{w}^{(t)} = \bar{w}^{(t-1)} + \frac{1}{nq} \cdot \left( \sum_{i=1}^{B_t} \Delta w_{[i]}^{(t)} \right) + Q^{(t)} \tag{2}$$

13:     **else**
14:

$$\bar{w}^{(t)} = \frac{1}{nq} \cdot \left( \sum_{i=1}^{B_t} w_{[i]}^{(t,K)} \right) + Q^{(t)}. \tag{3}$$

15:     **end if**
16: **end for**
17: **Output**: $\bar{w}^{(t)}$ for $t = 1, 2, \cdots, T$.

---

We want to stress again that our motivation to study DP-LSGD is not because we only focus on the federated setup, but to provide a unified analysis of the clipping bias and argue for using DP-LSGD *even in the centralized setup*. Our results are straightforwardly applicable to distributed learning with local DP [4] or client-level DP [5], where the only difference is that we may add a larger noise $Q^{(t)}$ determined by the number of local datapoints or the users involved, respectively, for these stronger DP definitions. As for the possible communication restriction where we need to add discrete noise of finite precision, one may replace the Gaussian noise by the Binomial mechanism [39].

## 3 Convergence of Synchronized-Only Iterate in Noisy Non-Clipped LSGD

In this section, we will study the convergence analysis of LSGD in Algorithm 1 using the non-clipped local update (3) for both convex and non-convex optimization.

**Theorem 3.1** (Last-iterate Convergence of Noisy LSGD in General Convex Optimization). *For an objective function $F(w) = \frac{1}{n} \cdot \sum_{i=1}^n f_i(w)$ where $f_i(w)$ is convex and $\beta$-smooth with variance-bounded gradient (Assumption 2.1), when $\eta < \min\{\frac{\beta}{\sqrt{24}K}, \frac{1}{\beta}, \frac{1}{2\beta+3K\beta/(nq)}\}$, $\log(TK) \geq 2$, and $Q^{(t)}$ is an independent noise such that $\mathbb{E}[Q^{(t)}] = 0$ and $\mathbb{E}[\|Q^{(t)}\|^2] \leq \bar{\mathcal{Q}}$, for some parameter $\bar{\mathcal{Q}}$ for $t = 1, 2, \cdots, T$, Algorithm 1 with (3) ensures*

$$\mathbb{E}[F(\bar{w}^{(T)})] \leq \left( \frac{\|\bar{w}^{(0)} - w^*\|^2}{\eta(TK+1)} + \log(TK+1)\left(6\eta\tau/(nq) + 8K^2\beta\tau\eta^2 + \bar{\mathcal{Q}}/\eta\right) \right.$$

$$\left. + 5\eta\beta^2(\log(TK)+1)\left(\|\bar{w}^{(0)} - w^*\|^2 + T\left(8\beta\eta^3 K^3\tau + \frac{12K^3\beta^2\eta^4\tau + 3K^2\eta^2\tau}{nq} + \bar{\mathcal{Q}}\right)\right) \right)$$

$$= \tilde{O}\left( \frac{\|\bar{w}^{(0)} - w^*\|^2}{\sqrt{TK}} + \frac{\tau}{\sqrt{TK}nq} + \frac{K\tau}{T} + \sqrt{TK}\bar{\mathcal{Q}} \right), \text{ if } \eta = O(1/\sqrt{TK}).$$

The proof can be found in Appendix A. To prove Theorem 3.1, with a careful analysis on $\|\bar{w}^{(t)} - w^*\|^2$, we develop a new last-iterate analysis framework, different from existing works [40, 41, 42] which must count on the assumption of bounded gradient. In Theorem 3.1, we need to assume the noise $Q$ to be independent and of zero-mean. Because we do not assume Lipschitz continuity of $F(w)$, we cannot provide a meaningful upper bound of the deviation between $F(w)$ and $F(w + Q)$ for arbitrary $w$ and $Q$ in general. However, provided the Lipschitz assumption, Theorem 3.1 can be easily generalized to handle biased perturbation. In Section 4, with an additional assumption on the similarity of the local functions (Assumption 4.2), we will show how to handle the clipping bias as a special biased noise. When there is no noise $\mathcal{Q} = 0$, provided that $K = O(T^{1/3}/(nq)^{2/3})$, we show LSGD achieves $\tilde{O}(\frac{\|\bar{w}^{(0)} - w^*\|^2 + \tau/(nq)^{2/3}}{\sqrt{TK}})$ last-iterate convergence in general-convex optimization.

We now study the non-convex scenario.

**Theorem 3.2** (Synchronized-only Iterate Convergence of Noisy LSGD in Non-convex Optimization).
*For an arbitrary objective function $F(w) = \frac{1}{n} \cdot \sum_{i=1}^n f_i(w)$, where $f_i(w)$ is $\beta$-smooth and satisfies Assumption 2.1, and for arbitrary perturbation (not necessarily independent or of zero mean) where $\mathbb{E}[\|Q^{(t)}\|^2] \leq \bar{\mathcal{Q}}$, when $\eta < \min\{\frac{\beta}{\sqrt{24K}}, \frac{1}{4\beta K}\}$, Algorithm 1 with (3) ensures that*

$$
\begin{aligned}
\mathbb{E}[\frac{\sum_{t=1}^T \|\nabla F(\bar{w}^{(t-1)})\|^2}{T}] &\leq \frac{4F(\bar{w}^{(0)})}{TK\eta} + \frac{16\eta^2\tau\beta^2 K^2}{nq} + \frac{4(1+\beta\eta)\sum_{t=1}^T \mathbb{E}[\|Q_i^{(t)}\|^2]}{\eta^2 KT} \\
&= O(\frac{\tau^{1/3}}{T^{2/3}(nq)^{1/3}} + \frac{T^{2/3}\tau^{2/3}K\bar{\mathcal{Q}}}{(nq)^{2/3}}),
\end{aligned}
\tag{4}
$$

*when we select $\eta = O(\frac{(nq)^{1/3}}{T^{1/3}K\tau^{1/3}})$. In particular, when $Q^{(t)}$ is independent and $\mathbb{E}[Q^{(t)}] = 0$, and $\eta = \Theta(1/K)$, then*

$$
\mathbb{E}[\frac{\sum_{t=1}^T \|\nabla F(\bar{w}^{(t-1)})\|^2}{T}] \leq O(\frac{F(\bar{w}^{(0)})}{\eta TK} + \tau + \frac{\sum_{t=1}^T \beta\mathbb{E}[\|Q^{(t)}\|^2]}{\eta TK}) = O(\frac{1}{T} + \tau + \bar{\mathcal{Q}}).
$$

The proof can be found in Appendix B. In Theorem 3.2, we provide an analysis on the effect of generic perturbation, which can also be used to capture the clipping bias in DP-LSGD. When there is no perturbation, Theorem 3.2 has two implications. First, we show to ensure $\min \mathbb{E}[\|\nabla F(\bar{w}^{(t)})\|^2] \leq \kappa$, we need $T = O(\frac{\sqrt{\tau/(nq)}}{\kappa^{3/2}})$, which is tighter than the state-of-the-art results $O(\frac{\tau/(nq)}{\kappa^2} + \frac{\sqrt{\tau}}{\kappa^{3/2}})$ in [30]. Second, compared to $O(1/T^{2/3})$, we also show that LSGD can converge faster in $O(1/T)$ to a $\tau$-*neighborhood* of a saddle point. This is helpful to understand the practical performance of DP-LSGD with bias, as discussed in Section 4.2.

As a final remark, we want to mention it is possible to improve the convergence rate from $O(1/T^{2/3})$ to $O(1/T)$ via careful variance reduction or error feedback mechanism, such as Scaffold [30] or FedLin [43]. However, the proper implementation of those advanced tricks in DP-LSGD with additional sensitivity control is not clear. As a first step to systematically study the generic clipping bias, in this paper we only focus on the regular LSGD. We will explain and discuss possible generalizations in Section 6.

## 4 Utility and Clipping Bias of DP-LSGD and DP-SGD

In this section, we move to study DP-LSGD with clipped local update (2) in Algorithm 1. To have a clear comparison with DP-SGD, we still consider the centralized setup and $F(w) = 1/n \cdot f_i(w)$ where each local function $f_i(w)$ is determined by a single sample. To capture the clipping bias, we need to introduce a new term, termed *incremental norm*.

**Definition 4.1** (Incremental Norm). *Consider applying the private and clipping version of Algorithm 1 with (2) on $F(w) = \sum_{i=1}^n f_i(w)$. In the $t$-th phase, we define $\Psi_i^{(t)} = \mathbf{1}(\|\Delta w_i^{(t)}\| > c) \cdot (\|\Delta w_i^{(t)}\| - c)$ as the incremental norm of the local update from $f_i(w)$ compared to the clipping threshold $c$, for $t = 1, 2, \cdots, T$.*

In Definition 4.1, the incremental norm $\Psi_i^{(t)}$ simply quantifies the difference between the norm of the local update and its clipped version from $f_i(w)$. In the following, we will always assume the DP noise injected $\mathbb{E}[\|Q^{(t)}\|^2] = \sigma^2 d$, following the classic privacy analysis of DP-SGD [38].

It is not hard to observe that the clipped local update is essentially a scaled version of the original update, and thus virtually one may view DP-LSGD as a generalization of noisy LSGD but each local update applies a different and adaptively-selected learning rate. To show meaningful characterization on the difference among those learning rates, we need the following assumption as a generalization of bounded-variance stochastic gradient.

**Assumption 4.1** (Incremental norm of Bounded Second Moment). *When applying the clipped version of Algorithm 1 via (2) on an objective function $F(w) = \frac{1}{n} \cdot f_i(w)$, $\mathbb{E}\big[\big(\sum_{i=1}^n (\Psi_i^{(t)})^2\big)/n\big]$ is upper bounded by $\mathcal{B}^2$, for some global parameter $\mathcal{B}$ for $t = 1, 2, \cdots, T$.*

Assumption 4.1 basically states that in expectation the square of $l_2$-norm of each local update is bounded. Assumption 4.1 also suggests that $\mathbb{E}\big[\big(\sum_{i=1}^n \Psi_i^{(t)}\big)/n\big] \leq \mathcal{B}$.

## 4.1 Utility of DP-LSGD in Convex Optimization

Another assumption we need for the anlysis of DP-LSGD on general convex optimization is the similarity among the local functions.

**Assumption 4.2** ($\gamma$ Similarity). *For $F(w) = 1/n \cdot \sum_{i=1}^n f_i(w)$, local functions $f_i$ are of $\gamma$-similarity to $F$ such that for any $w \in \mathcal{W}$, $|f_i(w) - F(w)| \leq \gamma$, for some constant $\gamma > 0$.*

The main reason why we need this additional Assumption 4.2 is because we do not assume Lipschitz continuity of $F(w)$. Thus, we alternatively consider to use the similarity among local functions to characterize the deviation of the evaluation of $F(\cdot)$ on biased iterates.

**Theorem 4.1** (Last-iterate of DP-LSGD in General Convex Optimization). *For an arbitrary objective function $F(w) = \frac{1}{n} \cdot \sum_{i=1}^n f_i(w)$ where $f_i(w)$ is convex and $\beta$-smooth, and under Assumptions 2.1, 4.1 and 4.2 when $\eta = O(1/\sqrt{TK})$ and $Q^{(t)}$ is independent DP noise such that $\mathbb{E}[Q^{(t)}] = 0$ and $\mathbb{E}[\|Q^{(t)}\|^2] = \sigma^2 d$, $t = 1, 2, \cdots, T$, then DP-LSGD with clipping threshold $c$ ensures that*

$$
\frac{c}{c+\mathcal{B}} \cdot \mathbb{E}[F(\bar{w}^{(T)}) - F(w^*)] = \tilde{O}\big((\frac{1}{\sqrt{TK}} + \frac{K}{nT})\|\bar{w}^{(0)} - w^*\|^2
$$
$$
+ (\frac{K}{nT} + \frac{1}{\sqrt{TK}})(1 + \frac{K^{3/2}}{\sqrt{T}} + \frac{K}{nq})\tau + (\frac{K^{3/2}}{\sqrt{T}n} + 1)\frac{\gamma\mathcal{B}}{c+\mathcal{B}} + \sqrt{TK}\sigma^2 d\big). \tag{5}
$$

*When $K = O(nq)$ and $K = O(T)$, and for $(\epsilon, \delta)$-DP, where $\sigma = \tilde{O}(\frac{c\sqrt{T\log(1/\delta)}}{n\epsilon})$, we have that*

$$
\mathbb{E}[F(\bar{w}^{(T)}) - F(w^*)]
$$
$$
= \tilde{O}\big(\underbrace{\frac{c+\mathcal{B}}{c} \cdot \big(\frac{\|\bar{w}^{(0)} - w^*\|^2}{\sqrt{TK}} + (\frac{1}{\sqrt{TK}} + \frac{K}{T})\tau\big)}_{(A)} + \underbrace{\frac{\gamma\mathcal{B}}{c}}_{(B)} + \underbrace{\frac{c+\mathcal{B}}{c} \cdot \frac{T^{3/2}K^{1/2}\log(1/\delta)dc^2}{n^2\epsilon^2}}_{(C)}\big).
$$

The proof can be found in Appendix C. We focus on a practical scenario where $\mathcal{B} = O(c)$, i.e., the incremental norm of local updates is in the same order of the clipping threshold $c$ selected, and thus $(c + \mathcal{B})/c = O(1)$. From Theorem 4.1, we show the last-iterate utility of DP-LSGD is captured by three terms: (A) a similar convergence rate as regular LSGD, (B) a clipping bias, and (C) the DP noise variance. First, ignoring the bias and noise, DP-LSGD still enjoys a convergence rate $\tilde{O}(\frac{\|\bar{w}^{(0)} - w^*\|^2}{\sqrt{TK}} + (\frac{1}{\sqrt{TK}} + \frac{K}{T})\tau)$, which is slightly worse compared to Theorem 3.2 with $\tilde{O}(\frac{\|\bar{w}^{(0)} - w^*\|^2}{\sqrt{TK}} + (\frac{1}{\sqrt{TKnq}} + \frac{K}{T})\tau)$ as a consequence of clipping which essentially applies different learning rates in each local update. Second, the clipping bias is captured by $(\gamma\mathcal{B})/c$. This matches our intuition that a larger incremental norm $\mathcal{B}$ combined with a smaller clipping threshold $c$ will imply a more significant change on the local update and thus a larger bias. The last accumulated perturbation term is determined by the noise injected across each phase with an effect of $\tilde{O}(\frac{T^{3/2}K^{1/2}\log(1/\delta)dc^2}{n^2\epsilon^2})$ for $(\epsilon, \delta)$-DP under $T$-fold composition.

As we consider the very generic setup with non-trival clipping, Theorem 3.2 cannot be directly compared to the classic DP-utility tradeoff [32] given Lipschitz continuity, where a utility loss $\tilde{\Theta}(\sqrt{d}/n\epsilon)$

is tight for convex optimization under $(\epsilon, \delta)$-DP. However, we have the following interesting observations. First, when we take the clipping threshold $c = O(\eta) = O(1/\sqrt{TK})$ and $K = O(T \cdot d/(n^2\epsilon^2))$, DP-LSGD achieves the same optimal rate $\tilde{O}(\sqrt{d}/n\epsilon)$ [33] ignoring the clipping bias. Second and more important, when the stochastic gradient variance $\tau$ is in the same order of the clipping bias $O(\gamma\mathcal{B}/c)$, then by selecting $c = \Theta(\eta)$ and $K = \Theta(T)$, Theorem 4.1 suggests that DP-LSGD will converge in $O(1/T)$ to an $O(\gamma\mathcal{B}/c + \frac{d}{n^2\epsilon^2})$ neighborhood of the global optimum. As a comparison, when we select $K = 1$ in Theorem 4.1, it becomes the analysis of DP-SGD but the convergence rate to the neighborhood of global optimum in the same scale $O(\gamma\mathcal{B}/c + \frac{d}{n^2\epsilon^2})$ is only $O(1/\sqrt{T})$. Moreover, as we will show in the next section, the local update bound $\mathcal{B}$ in DP-SGD with $K = 1$ in practice would be much larger than that of DP-LSGD with a relatively larger $K$. As a simple generalization, we also include an analysis of DP-LSGD on strongly-convex functions in Appendix D, and we move our focus to the non-convex optimization in the following.

## 4.2 Utility of DP-LSGD in Non-convex Optimization

**Theorem 4.2** (DP-LSGD in Non-convex Optimization). *For $F(w) = \frac{1}{n} \cdot \sum_{i=1}^{n} f_i(w)$ where $f_i(w)$ is $\beta$-smooth and satisfies Assumptions 2.1 and 4.1, when $\eta = O(1/K)$, DP-LSGD ensures that*

$$\mathbb{E}\left[\frac{\sum_{t=1}^{T}\|\nabla F(\bar{w}^{(t-1)})\|^2}{T}\right] \leq \frac{4F(\bar{w}^{(0)})}{TK\eta} + \frac{16\eta^2\tau\beta^2K^2}{nq} + \frac{4(1+\beta\eta)(\mathcal{B}^2/q + \sigma^2 d)}{\eta^2 K}. \quad (6)$$

*When we select $\eta = O(\frac{1}{\sqrt{TK}})$ and $K = \Theta(T)$, for $(\epsilon, \delta)$-DP we have that*

$$\mathbb{E}\left[\frac{\sum_{t=1}^{T}\|\nabla F(\bar{w}^{(t-1)})\|^2}{T}\right] = \tilde{O}\left(\frac{F(\bar{w}^{(0)})}{T} + \frac{\tau}{nq} + \frac{\mathcal{B}^2 T}{q} + \frac{d}{n^2\epsilon^2}\right). \quad (7)$$

The proof can be found in Appendix E. For the analysis of DP-LSGD in non-convex optimization, we do *not* need Assumption 4.2 on the similarity among local functions and Theorem 4.2 is simply obtained by substituting the clipping error from each phase into Theorem 3.2. To have a more clear picture, we still consider a practical scenario when $\mathcal{B} = \mathcal{B}_0 \cdot \eta$ for some constant $\mathcal{B}_0$ and the variance $\tau$ is also some constant. Then, from (7) we have that

$$\mathbb{E}\left[\frac{\sum_{t=1}^{T}\|\nabla F(\bar{w}^{(t-1)})\|^2}{T}\right] = O\left(\frac{F(\bar{w}^{(0)})}{T} + \frac{1}{nq} + \frac{\mathcal{B}_0^2}{q} + \frac{d}{n^2\epsilon^2}\right) = \tilde{O}\left(\frac{1}{T} + \frac{1}{q} + \frac{d}{n^2\epsilon^2}\right).$$

In other words, similar to the convex case, DP-LSGD will converge at a rate of $O(1/T)$ to an $\tilde{O}(1 + d/(n^2\epsilon^2))$ neighborhood of a saddle point given some constant sampling rate $q$. As a comparison, for DP-SGD when $K = 1$, from Theorem 3.2 we can only ensure an $O(1/\sqrt{T})$ convergence rate to a same $\tilde{O}(1 + d/(n^2\epsilon^2))$ neighborhood.

# 5 Why DP-LSGD Produces Less Bias and Better SNR

Throughout the previous section, we showed that asymptotically DP-LSGD enjoys a faster convergence rate to a neighborhood of (global/local) optimum compared to DP-SGD. We characterized the clipping bias mainly based on the second moment upper bound $\mathcal{B}^2$ of the incremental norm $\Psi_i^{(t)}$ of local updates. In this section, we proceed to empirically study the $\Psi_i^{(t)}$, and the tradeoff between clipping bias and DP (Gaussian) noise in practical deep learning tasks. We will explain why DP-LSGD could produce smaller bias and enable more efficient clipping compared to DP-SGD.

To produce good utility-privacy tradeoff, a proper selection of the clipping threshold $c$ is important. Many existing works are devoted to optimizing the selection of $c$ by either grid searching [35] or adaptive fine-tuning [44]. A smaller $c$ requires less DP noise. But, as a tradeoff shown in Theorem 4.1 and 4.2, a smaller $c$ and a consequently a larger $\mathcal{B}$ will also lead to a heavier clipping bias. Thus, from the perspective of signal-to-noise ratio (SNR), an ideal scenario is that the $l_2$-norm of each local update is *concentrated* such that we can maximize the efficiency of the clipping power $c$ with a small clipping effect for most local updates. Interpreted via our developed theory of clipping bias, it is expected that given the clipping threshold $c$, the incremental norm $\Psi_i^{(t)}$ would be small, captured by $\mathcal{B}$ in (5) and (7). In Fig. 1 (a,b), we plot various statistics of the incremental norm $\Psi_i^{(t)}$

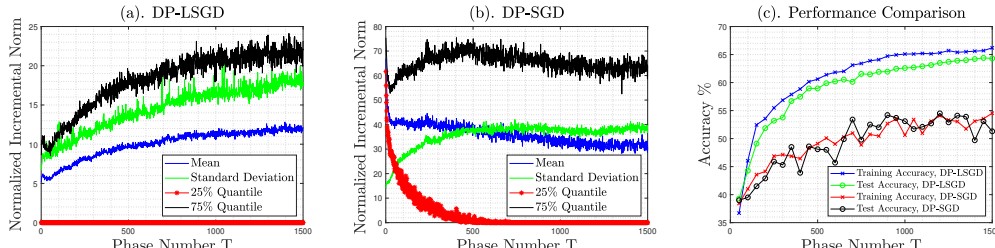

Figure 1: Training ResNet 20 on CIFAR10 with DP-LSGD ($K = 10, \eta = 0.025, c = 1$) and DP-SGD ($K = 1, \eta = 1, c = 1$) under ($\epsilon = 2, \delta = 10^{-5}$)-DP, with expected batch size 1000.

for DP-LSGD and DP-SGD, respectively, on training CIFAR10 [45]. By our analysis, DP-LSGD usually should apply a smaller learning rate $\eta$. To have a fair comparison, we consider the normalized incremental norm $\Psi_i^{(t)}/\eta$. Given the same clipping threshold, comparing Fig. 1 (a) and (b), the mean of normalized incremental norm, captured by $\mathcal{B}/\eta$ in our theorems, of DP-LSGD is only around 32% compared to that of DP-SGD. The corresponding standard deviation is around only 40% compared to that of DP-SGD. One may also compare the 25% and 75% quantiles, which suggest that more local updates bear less clipping influence in DP-LSGD and thus enjoying a higher clipping efficiency. We also report the comparison when training ResNet20 [46] on SVHN [47] in Fig. 2 in Appendix F with similar observations. Details of experiment setups and the anonymous GitHub code link can be found in Appendix F.

| Dataset and Method \ $\epsilon$ | 1.5 | 2.0 | 2.5 | 3.0 | 3.5 | 4.0 |
|---|---|---|---|---|---|---|
| CIFAR10, DP-LSGD ($K = 10$) | 59.4($\pm$0.5) | 64.0($\pm$0.3) | 66.2($\pm$0.4) | 67.7($\pm$0.3) | 68.7($\pm$0.2) | 69.9($\pm$0.3) |
| CIFAR10, DP-SGD ($K = 1$) | 49.8($\pm$1.2) | 58.7($\pm$1.0) | 59.9($\pm$1.2) | 60.6($\pm$0.8) | 62.1($\pm$0.6) | 62.8($\pm$0.6) |
| SVHN, DP-LSGD ($K = 10$) | 83.2($\pm$0.4) | 84.4($\pm$0.5) | 85.7($\pm$0.5) | 85.4($\pm$0.4) | 86.1($\pm$0.4) | 86.5($\pm$0.3) |
| SVHN, DP-SGD ($K = 1$) | 74.5($\pm$0.8) | 78.2($\pm$0.6) | 79.8($\pm$0.6) | 80.3($\pm$1.0) | 81.7($\pm$0.4) | 82.2($\pm$0.5) |

Table 1: **Test Accuracy** of ResNet20 on CIFAR10 and SVHN via DP-LSGD and DP-SGD under various $\epsilon$ and fixed $\delta = 10^{-5}$, with expected batch size 1000.

In Fig. 1 (c), we record the performance of DP-LSGD and DP-SGD, which coincides with our theory that DP-LSGD has a smaller clipping bias and a faster convergence rate. The smaller incremental norm in DP-LSGD is not surprising. With relatively larger $K$, for each individual function $f_i(w)$, though the $K$ local gradients are correlated and essentially determined by a single sample, the aggregation of them still averages out substantial sampling noise and makes the $l_2$-norm of local updates more concentrated. In Table 1, we include additional comparison between their performance on CIFAR10 [45] and SVHN [47]; DP-LSGD produces significant improvements.

## 6 Conclusion and Prospects

In this paper, via LSGD, we provide a unified analysis of the clipping bias and the utility loss in privacy-preserving gradient methods for both centralized and distributed setups. Provided the generic analysis, we develop the connections between the bias and the second moment of local updates. This initializes a new direction to systematically instruct private learning by connecting the research of variance reduction in distributed optimization. In this paper we only focus on regular LSGD to show its advantage over DP-SGD, but advanced acceleration methods [30, 31, 43] are known in non-private federated learning to further reduce the "local-update drift" caused by (per-sample) data heterogeneity. This could then further reduce the clipping bias given local updates of smaller variance. Thus, a promising future direction is to understand and incorporate those techniques within the sensitivity control framework. Another important issue we have not fully explored is the software implementation of DP-LSGD in the centralized case. For DP-SGD, many PyTorch libraries with fast per-sample gradient computation in low memory overhead have been developed, such as Opacus [48]. However, in all above-presented experiments, we simulate DP-LSGD in a distributed environment and compute each local update in parallel at a cost of large memory. Given limited hardware resources, this restricts the application of larger batchsize (tens of thousands) and deploying deeper neural networks, which are known to produce much better utility-privacy tradeoffs [36, 49]. We leave empirical efficiency improvement to future work.

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
