+ \big(\frac{K}{nT} + \frac{1}{\sqrt{TK}}\big)\big(1 + \frac{K^{3/2}}{\sqrt{T}} + \frac{K}{nq}\big)\tau + \big(\frac{K^{3/2}}{\sqrt{T}n} + 1\big)\frac{\gamma\mathcal{B}}{c + \mathcal{B}} + \sqrt{TK}\sigma^2 d\big).
\end{aligned}
\tag{5}
$$

*For $(\epsilon, \delta)$-DP, where $\sigma = \tilde{O}\big(\frac{c\sqrt{T \log(1/\delta)}}{n\epsilon}\big)$, we have that*

$$
\begin{aligned}
&\mathbb{E}[F(\bar{w}^{(T)}) - F(w^*)] \\
&= \tilde{O}\big(\underbrace{\frac{c + \mathcal{B}}{c} \cdot \big(\frac{\|\bar{w}^{(0)} - w^*\|^2}{\sqrt{TK}} + \big(\frac{1}{\sqrt{TK}} + \frac{K}{T}\big)\tau\big)}_{(A)} + \underbrace{\frac{\gamma\mathcal{B}}{c}}_{(B)} + \underbrace{\frac{c + \mathcal{B}}{c} \cdot \frac{T^{3/2}K^{1/2}\log(1/\delta)dc^2}{n^2\epsilon^2}}_{(C)}\big).
\end{aligned}
$$

The proof can be found in Appendix C. We focus on a practical scenario where $\mathcal{B} = O(c)$, i.e., the incremental norm of local updates is in the same order of the clipping threshold $c$ selected, and thus $(c + \mathcal{B})/c = O(1)$. From Theorem 4.1, we show the last-iterate utility of DP-LSGD is captured by three terms: (A) a similar convergence rate as regular LSGD, (B) a clipping bias, and (C) the DP noise variance. First, ignoring the bias and noise, DP-LSGD still enjoys a convergence rate $\tilde{O}\big(\frac{\|\bar{w}^{(0)} - w^*\|^2}{\sqrt{TK}} + \big(\frac{1}{\sqrt{TK}} + \frac{K}{T}\big)\tau\big)$.

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

We first present a sketch of the proof. There are two main challenges to derive the last-iterate convergence of LSGD with unbounded gradients. First, to derive the last-iterate guarantee, we need to keep track of the progress of $F(\bar{w}^{(t)}) - F(\bar{w}^{(t')})$ for different $t$ and $t'$. To support this, we still adopt the similar idea from existing works [2, 26] to consider a virtual sequence determined by the average of all intermediate updates assuming all users participate in the $t$-th phase, i.e., $\tilde{w}^{(t,k)} = \frac{1}{n} \cdot \sum_{i=1}^{n} w_i^{(t,k)}$. But instead, we show a more generic analysis on $F(\tilde{w}^{(t,k)}) - F(u)$ for arbitrary $u$ and a careful characterization of the difference between $F(\tilde{w}^{(t,k)})$ and $F(\bar{w}^{(t)})$ under sampling, given that $\bar{w}^{(t)}$ is the actual and only release. The second and more challenging problem is that we cannot straightforwardly apply classic last-iterate convergence analyses [40, 41, 42] which must count on the assumption of bounded gradient. To address this, in the proof, we alternatively use the following two kinds of upper bounds on the gradient norm

$$\|\nabla F(w)\|^2 = \|\nabla F(w) - \nabla F(w^*)\|^2 \leq \min\{\beta^2 \|w - w^*\|^2, 2\beta(F(w) - F(w^*))\},$$

which is based on the property of smoothness and convexity. With a careful analysis on $\|\tilde{w}^{(t,k)} - w^*\|^2$ for any $t$ and $k$, we propose a more generic last-iterate framework to handle unbounded and heterogeneous local update, simultaneously.

## A.1 Main Proof

Before the start, we define a virtual sequence $\hat{w}^{(t,k)} = \bar{w}^{(t-1)} + \frac{1}{nq} \sum_{i=1}^{n} 1_i^{(t)}(w_i^{(t,k)} - \bar{w}^{(t-1)})$ for those intermediate iterates produced by the users selected in the $t$-th phase. $1_i^{(t)}$ is an indicator which equals 1 iff the $i$-th user is selected in the $t$-th phase. Meanwhile, we imagine the scenario that all users participate in the $t$-th phase computation and a sequence of intermediate iterates $w_i^{(t,k)}$ for $i = 1, 2, \cdots, n$, and $k = 1, 2, \cdots, K$, is produced. We use $\tilde{w}^{(t,k)} = \frac{1}{n} \cdot \sum_{i=1}^{n} w_i^{(t,k)}$ to denote the average. It is not hard to observe that $\mathbb{E}[\hat{w}^{(t,k)}] = \tilde{w}^{(t,k)}$ conditional on $\bar{w}^{(t-1)}$. Moreover, $w_i^{(t,0)} = \tilde{w}^{(t,0)} = \bar{w}^{(t-1)}$ for $i = 1, 2, \cdots, n$. In the following, we unravel $\|\tilde{w}^{(t,k)} - u\|^2$ for arbitrary $u$ and obtain

$$\|\hat{w}^{(t,k)} - u\|^2 = \|\hat{w}^{(t,k-1)} - \frac{\eta}{nq} \sum_{i=1}^{n} 1_i^{(t)} \nabla f_i(w_i^{(t,k-1)}) - u\|^2$$

$$= \|\hat{w}^{(t,k-1)} - u\|^2 - \frac{2}{nq} \cdot \sum_{i=1}^{n} \eta 1_i^{(t)} \cdot \langle \hat{w}^{(t,k-1)} - u, \nabla f_i(w_i^{(t,k-1)}) \rangle + \|\frac{\sum_{i=1}^{n} \eta 1_i^{(t)} \nabla f_i(w_i^{(t,k-1)})}{nq}\|^2. \tag{8}$$

We first work on the last term $\|\frac{\sum_{i=1}^{n} \eta 1_i^{(t)} \nabla f_i(w_i^{(t,k-1)})}{nq}\|^2$ in (8).

**Lemma A.1.** *Conditional on $\bar{w}^{(t-1)}$,*

$$\mathbb{E}[\|\frac{\sum_{i=1}^{n} \eta 1_i^{(t)} \nabla f_i(w_i^{(t,k-1)})}{nq}\|^2] \leq \frac{10\eta^2 \beta^2}{n} \sum_{i=1}^{n} \|w_i^{(t,k-1)} - \tilde{w}^{(t,k-1)}\|^2 + \frac{6\eta^2 \tau}{nq}$$

$$+ 10\eta^2 \min\{2\beta(F(\tilde{w}^{(t,k-1)}) - F(w^*)), \beta^2 \|\tilde{w}^{(t,k-1)} - w^*\|^2\}. \tag{9}$$

Now, we move our focus to the second term $\frac{-2}{nq} \cdot \sum_{i=1}^{n} \eta 1_i^{(t)} \cdot \langle \hat{w}^{(t,k-1)} - u, \nabla f_i(w_i^{(t,k-1)}) \rangle$ of (8).

**Lemma A.2.** *Conditional on $\bar{w}^{(t-1)}$,*

$$\mathbb{E}\Big[ -\frac{2}{nq} \cdot \sum_{i=1}^{n} \eta 1_i^{(t)} \langle \hat{w}^{(t,k-1)} - u, \nabla f_i(w_i^{(t,k-1)}) \rangle \Big]$$

$$\leq 2\eta\big(F(u) - F(\tilde{w}_i^{(t,k-1)})\big) + \frac{\beta}{2n} \sum_{i=1}^{n} \|w_i^{(t,k-1)} - \tilde{w}^{(t,k-1)}\|^2\big). \tag{10}$$

522

Finally, we consider the upper bound of $\sum_{i=1}^{n} \|w_i^{(t,k-1)} - \tilde{w}^{(t,k-1)}\|^2$.

**Lemma A.3.** *When* $\eta < \frac{\beta}{\sqrt{24K}}$,

$$\sum_{i=1}^{n} \|w_i^{(t,k)} - \tilde{w}^{(t,k)}\|^2 \leq 4k^2 n\tau\eta^2. \tag{11}$$

Now, we combine Lemma A.1, A.2 and A.3 together and go back to (8). On one hand, when we adopt the upper bound of Lemma A.1 using $F(\tilde{w}^{(t,k)}) - F(w^*)$, we have

$$\mathbb{E}[\|\hat{w}^{(t,k)} - u\|^2] \leq \mathbb{E}\big[\|\hat{w}^{(t,k-1)} - u\|^2 + 20\eta^2\beta\big(F(\tilde{w}^{(t,k-1)}) - F(w^*)\big) + 2\eta(F(u) - F(\tilde{w}^{(t,k-1)}))$$
$$+ \frac{6\eta^2\tau}{nq} + (10\eta^2\beta^2 + \beta\eta) \cdot 4k^2\tau\eta^2\big]. \tag{12}$$

Sum up (12) on both sides from $k = 1, 2, \cdots, K$, and we have that

$$\mathbb{E}\Big[\sum_{k=1}^{K} 2\eta(F(\tilde{w}^{(t,k-1)}) - F(u)) - 20\eta^2\beta\big(F(\tilde{w}^{(t,k-1)}) - F(w^*)\big)\Big] \tag{13}$$
$$\leq \mathbb{E}[\|\bar{w}^{(t-1)} - u\|^2 - \|\hat{w}^{(t,K)} - u\|^2] + \frac{6K\eta^2\tau}{nq} + (10\eta^2\beta^2 + \beta\eta) \cdot 4K^3\tau\eta^2.$$

When $u = w^*$, it is noted that the left side of (13) becomes

$$\mathbb{E}\Big[\sum_{k=1}^{K} (2\eta - 20\eta^2\beta)(F(\tilde{w}^{(t,k-1)}) - F(w^*))\Big],$$

and once $\eta$ is small enough such that $2(\eta - 10\eta^2\beta) > 0$ where $\eta < 1/(10\beta)$, then the above is non-negative. In the following, we further take the perturbation $Q^{(t)}$ into accountant. It is noted that

$$\mathbb{E}[\|\bar{w}^{(t)} - u\|^2] = \mathbb{E}[\|\hat{w}^{(t,K)} + Q^{(t)} - u\|^2] = \mathbb{E}[\|\hat{w}^{(t,K)} - u\|^2] + \mathbb{E}[\|Q^{(t)}\|^2], \tag{14}$$

since $Q^{(t)}$ is independent zero-mean noise. Therefore, when we further sum up (13) for $t = 1, 2, \cdots, T$ combined with (14),

$$\mathbb{E}\Big[\frac{\sum_{t=1}^{T}\sum_{k=1}^{K} F(\tilde{w}^{(t,k)}) - F(w^*)}{TK}\Big]$$
$$\leq \frac{\|\bar{w}^{(0)} - w^*\|^2}{(2\eta - 20\eta^2\beta)TK} + \frac{(6\eta^2\tau/(nq) + (10\eta^2\beta^2 + \beta\eta) \cdot 4K^2\tau\eta^2) + \bar{\mathcal{Q}}/K}{(2\eta - 20\eta^2\beta)}. \tag{15}$$

Here, as assumed $\mathbb{E}[\|Q^{(t)}\|^2] \leq \bar{\mathcal{Q}}$. When $\eta < 1/(20\beta)$, which suggests that $(2\eta - 20\eta^2\beta) \geq \eta$ and $(10\eta^2\beta^2 + \beta\eta) \leq 2\beta\eta$, respectively, (15) can be simplified as

$$\mathbb{E}\Big[\frac{\sum_{t=1}^{T}\sum_{k=1}^{K} F(\tilde{w}^{(t,k)}) - F(w^*)}{TK}\Big] \leq \frac{\|\bar{w}^{(0)} - w^*\|^2}{\eta TK} + \big(\frac{6\eta\tau}{nq} + 8\beta K^2\tau\eta^2\big) + \bar{\mathcal{Q}}/(\eta K) \tag{16}$$

On the other hand, when we apply Lemma A.1 in (12) if we adopt the form $\beta^2\|\tilde{w}^{(t,k-1)} - w^*\|^2$ as the upper bound, we have

$$\mathbb{E}[\|\hat{w}^{(t,k)} - u\|^2] \leq \mathbb{E}\big[\|\hat{w}^{(t,k-1)} - u\|^2 + 10\eta^2\beta^2\|\tilde{w}^{(t,k-1)} - w^*\|^2 + 2\eta(F(u) - F(\tilde{w}^{(t,k-1)}))$$
$$+ \frac{6\eta^2\tau}{nq} + (10\eta^2\beta^2 + \beta\eta) \cdot 4k^2\tau\eta^2\big]. \tag{17}$$

With a similar reasoning, when $\eta < 1/(20\beta)$,

$$\mathbb{E}[F(\tilde{w}^{(t,k-1)}) - F(u)]$$
$$\leq \mathbb{E}\Big[\frac{\|\hat{w}^{(t,k-1)} - u\|^2 - \|\hat{w}^{(t,k)} - u\|^2}{2\eta} + 5\eta\beta^2\|\tilde{w}^{(t,k-1)} - w^*\|^2 + \frac{3\eta\tau}{nq} + 4k^2\beta\tau\eta^2\Big]. \tag{18}$$

However, to apply (18), we need an additional result to upper bound the term $\|\tilde{w}^{(t,k-1)} - w^*\|$, summarized as the following lemma.

**Lemma A.4.** *With the initialization $\bar{w}^{(0)}$, when $\eta < \min\{\frac{\beta}{\sqrt{24}K}, \frac{1}{20\beta}, \frac{1}{2\beta+3K\beta/(nq)}\}$, for any $k \in [0:K-1]$,*

$$\mathbb{E}[\|\tilde{w}^{(t,k)} - w^*\||] \leq \|\bar{w}^{(0)} - w^*\| + 8t\beta\eta^3 K^3\tau + (t-1)\big(\bar{\mathcal{Q}} + \frac{12K^4\beta^2\eta^4\tau + 3K^2\eta^2\tau}{nq}\big).$$

From Lemma A.4, we also have a global bound that for any $t \in [1:T]$ and $k \in [0:K]$,

$$\mathbb{E}[\|\tilde{w}^{(t,k)} - w^*\||] \leq \|\bar{w}^{(0)} - w^*\| + T\big(8\beta\eta^3 K^3\tau + \big(\bar{\mathcal{Q}} + \frac{12K^4\beta^2\eta^4\tau + 3K^2\eta^2\tau}{nq}\big)\big). \quad (19)$$

Now, for any $t_0 \in [1:T]$ and $k_0 \in [0:K-1]$, if we select $u = \tilde{w}^{(t_0,k_0)}$, stemmed from (18),

$$
\begin{aligned}
\frac{\sum_{(t,k)\in\mathcal{C}} \mathbb{E}[F(\tilde{w}^{(t,k)}) - F(\tilde{w}^{(t_0,k_0)})]}{(T-t_0+1)K - k_0} &\leq 3\eta\tau/(nq) + 4K^2\beta\tau\eta^2 \\
+ \frac{(T-t_0+1)\bar{\mathcal{Q}}}{2\eta((T-t_0+1)K - k_0)} &+ \frac{5\eta\beta^2 \sum_{(t,k)\in\mathcal{C}} \mathbb{E}[\|\tilde{w}^{(t,k)} - w^*\|^2]}{(T-t_0+1)K - k_0},
\end{aligned}
\quad (20)
$$

where $\mathcal{C} = \big((t_0,k), k = k_0, \cdots, K-1\big) \cup \big((t,k), t = t_0+1, \cdots, T, k = 0, \cdots, K-1\big)$. Finally, as we are concerning about the utility of $\mathcal{F}(\bar{w}^{(T)})$, we need to virtually implement one more gradient descent step on $\bar{w}^{(T)}$ to get an upper bound of $F(\bar{w}^{(T)}) - F(w^*)$. To be specific, we imagine one additional full gradient descent using the entire set on $\bar{w}^{(T)}$, and for any $u$, we have that

$$
\begin{aligned}
\|\tilde{w}^{(T+1,1)} - u\|^2 &= \|\bar{w}^{(T)} - u - \eta \cdot \frac{\sum_{i=1}^n \nabla f_i(\bar{w}^{(T)})}{n}\|^2 \\
&\leq \|\bar{w}^{(T)} - u\|^2 - 2\eta\big(F(\bar{w}^{(T)}) - F(u)\big) + \eta^2\|\nabla F(\bar{w}^{(T)}) - \nabla F(w^*)\|^2 \\
&\leq \|\bar{w}^{(T)} - u\|^2 - 2\eta\big(F(\bar{w}^{(T)}) - F(u)\big) + \min\eta^2\{\beta^2\|\bar{w}^{(T)} - w^*\|^2, 2\beta(F(\bar{w}^{(T)}) - F(w^*))\}.
\end{aligned}
\quad (21)
$$

Therefore, let $u = w^*$ and we can combine (16) and (21) to produce the following. Since we assume $(2\eta - 20\eta^2\beta) \geq \eta$ which also implies $2(\eta - \eta^2\beta) \geq \eta$, we have

$$
\begin{aligned}
\mathbb{E}\big[\frac{\sum_{t=1}^T \sum_{k=1}^K \big(F(\tilde{w}^{(t,k-1)}) - F(w^*)\big) + \big(F(\bar{w}^{(T)}) - F(w^*)\big)}{TK+1}\big] \\
\leq \frac{\|\bar{w}^{(0)} - w^*\|^2}{\eta(TK+1)} + \big(\frac{6\eta\tau}{nq} + 8\beta K^2\tau\eta^2\big) + \bar{\mathcal{Q}}/(\eta K).
\end{aligned}
\quad (22)
$$

Similarly, for (20), it is noted that conditional on $\bar{w}^{(t-1)}$, we have that

$$\mathbb{E}[\|\hat{w}^{(t,k)} - u\|^2] = \mathbb{E}[\|\hat{w}^{(t,k)} - \tilde{w}^{(t,k)}\|^2] + \|\tilde{w}^{(t,k)} - u\|^2, \quad (23)$$

and for $\mathbb{E}[\|\hat{w}^{(t,k)} - \tilde{w}^{(t,k)}\|^2]$ for any $t$ and $k$,

$$
\begin{aligned}
\mathbb{E}[\|\hat{w}^{(t,k)} - \tilde{w}^{(t,k)}\|^2] &= \mathbb{E}[\|(\hat{w}^{(t,k)} - \bar{w}^{(t-1)}) - (\tilde{w}^{(t,k)} - \bar{w}^{(t-1)})\|^2] \\
&= \eta^2\mathbb{E}[\|\sum_{i=1}^n \frac{(\mathbf{1}^{(t)} - q)}{nq} \cdot \sum_{l=0}^{k-1} \nabla f_i(w_i^{(t,k)})\|^2] \leq \frac{\eta^2 k(q-q^2)}{n^2 q^2} \cdot \sum_{i=1}^n \sum_{l=0}^{k-1} \|\nabla f_i(w_i^{(t,l)})\|^2 \\
&= \frac{\eta^2 k(q-q^2)}{n^2 q^2} \cdot \sum_{i=1}^n \sum_{l=0}^{k-1} \|\nabla f_i(w_i^{(t,l)}) - \nabla f_i(\tilde{w}^{(t,l)}) + \nabla f_i(\tilde{w}^{(t,l)}) - F(\tilde{w}^{(t,l)}) + \nabla F(\tilde{w}^{(t,l)}) - \nabla F(w^*)\|^2) \\
&\leq \frac{3k\eta^2}{n^2 q} \cdot \sum_{i=1}^n \sum_{l=0}^{k-1} \big(\beta^2\|w_i^{(t,l)} - \tilde{w}^{(t,l)}\|^2 + \beta^2\|\tilde{w}^{(t,l)} - w^*\|^2 + \tau\big) \\
&\leq \frac{3K\eta^2}{nq}\big(4\beta^2 K^3\tau\eta^2 + K\tau + \sum_{l=0}^{k-1} \beta^2\|\tilde{w}^{(t,l)} - w^*\|^2\big).
\end{aligned}
\quad (24)
$$

where the last line of (24) we apply Lemma A.4. Therefore, by replacing $\mathbb{E}[\|\hat{w}^{(t,k)} - u\|^2]$ with $\mathbb{E}[\|\hat{w}^{(t,k)} - \tilde{w}^{(t,k)}\|^2] + \|\tilde{w}^{(t,k)} - u\|^2$ in (18), we have that

$$\mathbb{E}[F(\tilde{w}^{(t,k-1)}) - F(u)] \leq \mathbb{E}\Big[\frac{\|\tilde{w}^{(t,k-1)} - u\|^2 - \|\tilde{w}^{(t,k)} - u\|^2 + \|\hat{w}^{(t,k-1)} - \tilde{w}^{(t,k-1)}\|^2 - \|\hat{w}^{(t,k)} - \tilde{w}^{(t,k-1)}\|^2}{2\eta}$$

$$+ 5\eta\beta^2\|\tilde{w}^{(t,k-1)} - w^*\|^2 + \frac{3\eta\tau}{nq} + 4K^2\beta\tau\eta^2\Big].$$
(25)

Now, we let $u = \tilde{w}^{(t_0,k_0)}$ in (21) and (25), combining (24) we have

$$\frac{\sum_{t=t_0}^{T}\sum_{k=k_0}^{K-1}\mathbb{E}[F(\tilde{w}^{(t,k)}) - F(\tilde{w}^{(t_0,k_0)})] + \mathbb{E}[F(\bar{w}^{(T)}) - F(\tilde{w}^{(t_0,k_0)})]}{((T - t_0 + 1)K - k_0 + 1)}$$

$$\leq 3\eta\tau/(nq) + 4K^2\beta\tau\eta^2 + \frac{(T - t_0 + 1)\bar{Q}}{2\eta((T - t_0 + 1)K - k_0 + 1)}$$

$$+ \frac{\frac{3K\eta}{nq}\big(4\beta^2 K^3\tau\eta^2 + K\tau + \sum_{l=0}^{k-1}\beta^2\|\tilde{w}^{(t,l)} - w^*\|^2\big)}{2((T - t_0 + 1)K - k_0 + 1)}$$

$$+ \frac{5\eta\beta^2\big(\sum_{t=t_0}^{T}\sum_{k=k_0+1}^{K}\mathbb{E}[\|\tilde{w}^{(t,k)} - w^*\|^2] + \mathbb{E}[\|\bar{w}^{(T)} - w^*\|^2]\big)}{(T - t_0 + 1)K - k_0 + 1}.$$
(26)

Now, we can apply the last-iterate convergence rate trick.

**Lemma A.5.** *For any sequence $y_i$, $i = 1, 2, \cdots, M$,*

$$y_M = \frac{\sum_{j=1}^{M} y_j}{M} + \sum_{j=1}^{M-1}\frac{\sum_{l=M-j+1}^{M}(y_l - y_{M-j})}{j(j+1)}$$
(27)

One can easily verify the identity in Lemma A.5.

If we take $y_j = \mathbb{E}[F(\tilde{w}^{(t,k)}) - F(w^*)]$ and $z_j = \mathbb{E}[\|\tilde{w}^{(t,k)} - w^*\|^2]$, for $j = (t-1)K + k$ and let $M = TK + 1$ where $y_{TK+1} = \mathbb{E}[F(\bar{w}^{(T)}) - F(w^*)]$ and $z_{TK+1} = \mathbb{E}[\|\bar{w}^{(T)} - w^*\|^2]$, combined with (22),(26) and Lemma A.5, we have that

$$y_{TK+1} = \mathbb{E}[F(\bar{w}^{(T)}) - F(w^*)]$$
(28)

$$= \frac{\sum_{j=1}^{TK} y_j}{TK + 1} + \sum_{j=1}^{TK}\frac{1}{j+1}\cdot\frac{\sum_{l=TK+2-j}^{TK+1}(y_l - y_{TK+1-j})}{j}$$
(29)

$$\leq \Big\{\frac{\|\bar{w}^{(0)} - w^*\|^2}{\eta(TK + 1)} + \big(\frac{6\eta\tau}{nq} + 8\beta K^2\tau\eta^2\big) + \bar{Q}/(\eta K)\Big\}$$
(30)

$$+ \sum_{j=1}^{TK}\Big\{\frac{1}{j+1}\cdot\big(\frac{3\eta\tau}{nq} + 4\beta K^2\tau\eta^2 + \frac{\bar{Q}}{2\eta} + \frac{12K^4\eta^3\beta^2\tau}{2nq} + \frac{3K^2\eta\tau}{2nq} + \frac{3K^2\eta}{nq}\max_l\{z_l\}\big) + 5\eta\beta^2\frac{\sum_{l=TK-j+2}^{TK+1} z_l}{j(j+1)}\Big\}$$
(31)

$$\leq \frac{\|\bar{w}^{(0)} - w^*\|^2}{\eta(TK + 1)} + \log(TK + 1)\big(\frac{6\eta\tau}{nq} + 8\beta K^2\tau\eta^2 + \bar{Q}/\eta + \frac{12K^4\eta^3\beta^2\tau}{2nq} + \frac{3K^2\eta\tau}{2nq} + \frac{3K^2\eta}{nq}\max_l\{z_l\}\big)$$
(32)

$$+ (5\eta\beta^2)\sum_{j=1}^{TK}\big(\frac{1}{j} - \frac{1}{TK + 1}\big)\cdot z_{TK-j+2}$$
(33)

In (30), we apply (22) on $\frac{\sum_{j=1}^{TK} y_j}{TK+1}$. In (31), we apply the results in (26) and $\frac{(T-t_0+1)\bar{Q}}{2\eta((T-t_0+1)K-k_0+1)} \leq \frac{\bar{Q}}{2\eta}$, since the number of iterates is always no less than the number of synchronization in any time interval. In (33), we use the fact that $\sum_{j=1}^{TK}\frac{1}{j+1} \leq \log(TK + 1)$ and as assumed $\log(TK) \geq 2$.

Now, with the assumption that $K^2 = O(nq)$, (33) can be further bounded as

$$y_{TK+1} < O(1) \cdot \left( \frac{\|\bar{w}^{(0)} - w^*\|^2}{\eta(TK+1)} + \log(TK+1)\left( \frac{\eta\tau}{nq} + K^2\tau\eta^2 + \bar{\mathcal{Q}}/\eta + \tau\eta \right) + \eta\left( \sum_{j=1}^{TK} \frac{1}{j} \right) \cdot \max_l \{z_l\} \right) \tag{34}$$

$$\leq O(1) \cdot \left( \frac{\|\bar{w}^{(0)} - w^*\|^2}{\eta(TK+1)} + \log(TK+1)\left( \frac{\eta\tau}{nq} + K^2\tau\eta^2 + \bar{\mathcal{Q}}/\eta + \tau\eta \right) \right. \tag{35}$$

$$\left. + \eta(\log(TK) + 1)\left( \|\bar{w}^{(0)} - w^*\|^2 + T\left( \beta\eta^3 K^3\tau + \frac{K^4\beta^2\eta^4\tau + K^2\eta^2\tau}{nq} + \bar{\mathcal{Q}} \right) \right) \right). \tag{36}$$

In (36), we apply Lemma A.4 and (19). Thus, we complete the proof.

## A.2  Proof of Lemma A.1

Conditional on $\bar{w}^{(t-1)}$, we have that

$$\mathbb{E}\left[ \left\| \frac{\sum_{i=1}^n \eta \mathbf{1}_i^{(t)} \nabla f_i(w_i^{(t,k-1)})}{nq} \right\|^2 \right]$$

$$= \mathbb{E}\left[ \left\| \frac{\sum_{i=1}^n \eta \mathbf{1}_i^{(t)} \nabla f_i(w_i^{(t,k-1)})}{nq} - \frac{\sum_{i=1}^n \eta \nabla f_i(w_i^{(t,k-1)})}{n} + \frac{\sum_{i=1}^n \eta \nabla f_i(w_i^{(t,k-1)})}{n} \right\|^2 \right]$$

$$\leq 2 \cdot \mathbb{E}\left[ \left\| \frac{\sum_{i=1}^n \eta (\mathbf{1}_i^{(t)} - q) \nabla f_i(w_i^{(t,k-1)})}{nq} \right\|^2 \right] + 2 \cdot \left\| \frac{\sum_{i=1}^n \eta \nabla f_i(w_i^{(t,k-1)})}{n} \right\|^2 \tag{37}$$

$$= \frac{2(q - q^2) \sum_{i=1}^n \|\eta \nabla f_i(w_i^{(t,k-1)})\|^2}{(nq)^2} + 2 \cdot \left\| \frac{\sum_{i=1}^n \eta \nabla f_i(w_i^{(t,k-1)})}{n} \right\|^2$$

$$\leq \frac{2\eta^2 \sum_{i=1}^n \|\nabla f_i(w_i^{(t,k-1)})\|^2}{n^2 q} + 2\eta^2 \left\| \frac{\sum_{i=1}^n \nabla f_i(w_i^{(t,k-1)})}{n} \right\|^2.$$

In the fourth line of (37), we use the fact that $\mathbf{1}_{[1:n]}^{(t)}$ are i.i.d. Bernoulli variable of mean $q$, and thus $\mathbb{E}[(\mathbf{1}_i^{(t)} - q)^2] = q(1-q)$ and $\mathbb{E}[(\mathbf{1}_i^{(t)} - q) \cdot (\mathbf{1}_j^{(t)} - q)] = 0$ for $i \neq j$. As for $\sum_{i=1}^n \|\nabla f_i(w_i^{(t,k-1)})\|^2$, we can further bound it as follows,

$$\sum_{i=1}^n \|\nabla f_i(w_i^{(t,k-1)}) - \nabla f_i(\tilde{w}^{(t,k-1)}) + \nabla f_i(\tilde{w}^{(t,k-1)}) - \nabla f_i(w^*) + \nabla f_i(w^*)\|^2 \tag{38}$$

$$\leq 3 \sum_{i=1}^n \left( \beta^2 \|w_i^{(t,k-1)} - \tilde{w}^{(t,k-1)}\|^2 + 2\beta \mathcal{D}_{f_i}(\tilde{w}^{(t,k-1)}, w^*) + \|\nabla f_i(w^*)\|^2 \right) \tag{39}$$

$$\leq 3\beta^2 \sum_{i=1}^n \|w_i^{(t,k-1)} - \tilde{w}^{(t,k-1)}\|^2 + 6\beta n \left( F(\tilde{w}^{(t,k-1)}) - F(w^*) \right) + 3n\tau. \tag{40}$$

In (39), we apply AM-GM inequality again and use the property that for convex and $\beta$-smooth function $f_i(w)$, it holds that $\|\nabla f_i(x) - \nabla f_i(y)\|^2 \leq 2\beta \mathcal{D}_{f_i}(x, y)$, where $\mathcal{D}_{f_i}(x, y) = f(x) - f(y) - \langle \nabla f(y), x - y \rangle$ is the Bregman divergence. In (40), we use the fact that $\nabla F(w^*) = 0$ and due to Assumption 2.1, the variance $\sum_{i=1}^n \|\nabla f_i(w^*) - \nabla F(w^*)\|^2 = \sum_{i=1}^n \|\nabla f_i(w^*)\|^2 \leq n\tau$.

When we apply similar decomposition tricks in (40) to the term $\left\| \frac{\sum_{i=1}^n \nabla f_i(w_i^{(t,k-1)})}{n} \right\|^2$,

$$\left\| \frac{\sum_{i=1}^n \nabla f_i(w_i^{(t,k-1)})}{n} \right\|^2$$

$$\leq \left\| \frac{\sum_{i=1}^n \nabla f_i(w_i^{(t,k-1)}) - \nabla f_i(\tilde{w}^{(t,k-1)}) + \nabla f_i(\tilde{w}^{(t,k-1)}) - \nabla f_i(w^*) + \nabla f_i(w^*)}{n} \right\|^2$$

$$\leq 2\left( \left\| \frac{\sum_{i=1}^n \nabla f_i(w_i^{(t,k-1)}) - \nabla f_i(\tilde{w}^{(t,k-1)})}{n} \right\|^2 + \left\| \frac{\sum_{i=1}^n \nabla f_i(\tilde{w}^{(t,k-1)}) - \nabla f_i(w^*)}{n} \right\|^2 \right)$$

$$\leq \frac{2\beta^2 \sum_{i=1}^n \|w_i^{(t,k-1)} - \tilde{w}^{(t,k-1)}\|^2}{n} + 4\beta \left( F(\tilde{w}^{(t,k-1)}) - F(w^*) \right),$$

since $\nabla F(w^*) = \frac{1}{n} \cdot \sum_{i=1}^{n} \nabla f_i(w^*) = 0$. Thus, (37) can be further bounded as follows:

$$
\mathbb{E}[\|\frac{\sum_{i=1}^{n} \eta \mathbf{1}_i^{(t)} \nabla f_i(w_i^{(t,k-1)})}{nq}\|^2]
$$
$$
\leq \frac{10\eta^2\beta^2}{n} \sum_{i=1}^{n} \|w_i^{(t,k-1)} - \tilde{w}^{(t,k-1)}\|^2 + 20\beta\eta^2 (F(\tilde{w}^{(t,k-1)}) - F(w^*)) + \frac{6\eta^2\tau}{nq}. \tag{41}
$$

Here, we use the fact that $q \geq 1/n$ and thus $\frac{1}{n^2 q} \leq \frac{1}{n}$. Meanwhile, it is noted that $\|\nabla f_i(\tilde{w}^{(t,k-1)}) - \nabla f_i(w^*)\|^2$ can also be bounded by $\beta^2\|\tilde{w}^{(t,k-1)} - w^*\|^2$ alternatively due to the smooth assumption. Thus, by replacing $2\beta(F(\tilde{w}^{(t,k-1)}) - F(w^*))$ in (39) and (41) with $\beta^2\|\tilde{w}^{(t,k-1)} - w^*\|^2$, we complete the proof.

### A.3 Proof of Lemma A.2

Based on the Poisson sampling assumption, conditional on $\bar{w}^{(t-1)}$,

$$
\mathbb{E}\Big[-\frac{2}{nq} \cdot \sum_{i=1}^{n} \eta \mathbf{1}_i^{(t)} \langle \tilde{w}^{(t,k-1)} - u, \nabla f_i(w_i^{(t,k-1)})\rangle\Big] = -\frac{2\eta}{n}[\sum_{i=1}^{n} \langle \tilde{w}^{(t,k-1)} - u, \nabla f_i(w_i^{(t,k-1)})\rangle].
$$

For each $i$, it is noted that

$$
-\langle \tilde{w}^{(t,k-1)} - u, \nabla f_i(w_i^{(t,k-1)})\rangle
$$
$$
= -\langle w_i^{(t,k-1)} - u, \nabla f_i(w_i^{(t,k-1)})\rangle - \langle \tilde{w}^{(t,k-1)} - w_i^{(t,k-1)}, \nabla f_i(w_i^{(t,k-1)})\rangle \tag{42}
$$
$$
\leq f_i(u) - f_i(w_i^{(t,k-1)}) + f_i(w_i^{(t,k-1)}) - f_i(\tilde{w}^{(t,k-1)}) + \frac{\beta}{2}\|w_i^{(t,k-1)} - \tilde{w}^{(t,k-1)}\|^2.
$$

In (42), we use the following facts. First, for smooth and convex function $f_i$, $\mathcal{D}_{f_i}(u, w_i^{(t,k-1)}) \geq 0$ and thus $-\langle w_i^{(t,k-1)} - u, \nabla f_i(w_i^{(t,k-1)})\rangle \leq f_i(u) - f_i(w_i^{(t,k-1)})$. Second, for the term $-\langle \tilde{w}^{(t,k-1)} - w_i^{(t,k-1)}, \nabla f_i(w_i^{(t,k-1)})\rangle$, we use the classic smooth inequality where

$$
f_i(\tilde{w}^{(t,k-1)}) \leq f_i(w_i^{(t,k-1)}) + \langle \tilde{w}^{(t,k-1)} - w_i^{(t,k-1)}, \nabla f_i(w_i^{(t,k-1)})\rangle + \frac{\beta}{2}\|w_i^{(t,k-1)} - \tilde{w}^{(t,k-1)}\|^2.
$$

Therefore, by (42), we have that

$$
-\frac{2\eta}{n}[\sum_{i=1}^{n} \langle \tilde{w}^{(t,k-1)} - u, \nabla f_i(w_i^{(t,k-1)})\rangle] \leq 2\eta\big(F(u) - F(\tilde{w}^{(t,k-1)})\big) + \frac{\beta}{2n} \sum_{i=1}^{n} \|w^{(t,k-1)} - \tilde{w}^{(t,k-1)}\|^2\big).
$$

### A.4 Proof of Lemma A.3

Given $\bar{w}^{(t-1)}$,

$$
\sum_{i=1}^{n} \big[\|w_i^{(t,k)} - \tilde{w}^{(t,k)}\|^2\big] = \eta^2 \sum_{i=1}^{n} \big[\|\sum_{l=0}^{k-1} \nabla f_i(w_i^{(t,l)}) - \frac{\sum_{j=1}^{n} \sum_{l=0}^{k-1} \nabla f_j(w_j^{(t,l)})}{n}\|^2\big] \tag{43}
$$

$$
\leq 3k\eta^2 \big[\sum_{i=1}^{n} \sum_{l=0}^{k-1} \big(\|\nabla f_i(w_i^{(t,l)}) - \nabla f_i(\tilde{w}^{(t,l)})\|^2 + \|\nabla f_i(\tilde{w}^{(t,l)}) - \nabla F(\tilde{w}^{(t,l)})\|^2 \tag{44}
$$

$$
+ \|\nabla F(\tilde{w}^{(t,l)}) - \frac{\sum_{j=1}^{n} \nabla f_j(w_j^{(t,l)})}{n}\|^2\big)\big] \tag{45}
$$

$$
\leq 3k\eta^2 \big[\big(\sum_{i=1}^{n} \sum_{l=0}^{k-1} \beta^2\|w_i^{(t,l)} - \tilde{w}^{(t,l)}\|^2\big) + kn\tau + \sum_{i=1}^{n} \sum_{l=0}^{k-1} \frac{\beta^2\|\tilde{w}^{(t,l)} - w_i^{(t,l)}\|^2}{n}\big] \tag{46}
$$

$$
\leq 3k\beta^2\eta^2(1 + 1/n) \sum_{i=1}^{n} \sum_{l=0}^{k-1} [\|w_i^{(t,l)} - \tilde{w}^{(t,l)}\|^2] + 3k^2 n\tau\eta^2. \tag{47}
$$

587  In (45), we use the fact that $\|\sum_{i=1}^{3} v_i\|^2 \leq 3\sum_{i=1}^{3}\|v_i\|^2$. In (46), we use Assumption 2.1 that the

588  variance of stochastic gradient is bounded by $\tau$ and apply the form $\nabla F(\tilde{w}^{(t,l)}) = \frac{\sum_{i=1}^{n} \nabla f_i(\tilde{w}^{(t,l)})}{n}$.

Let $M^{(k)} = \mathbb{E}[\sum_{i=1}^{n}\|w_i^{(t,k)} - \tilde{w}^{(t,k)}\|^2]$. Then, from (47), when $n \geq 1$, we have an inequality in a form

$$M^{(k)} \leq \eta^2\big(6k\beta^2 \sum_{l=0}^{k-1} M^{(l)} + 3k^2 n\tau\big),$$

589  where $M^{(0)} = \|\bar{w}^{(t-1)} - \bar{w}^{(t-1)}\|^2 = 0$. It is not hard to verify that by induction, once $\eta^2 < \frac{\beta^2}{24K^2}$,

590  $M^{(k)} \leq 4\eta^2 k^2 n\tau$.

591  ## A.5  Proof of Lemma A.4

To provide more intuition, we start from the case when $t = 1$, $\tilde{w}^{(t,0)} = \bar{w}^{(0)}$ and thus

$$\|\tilde{w}^{(1,k)} - w^*\|^2 = \|\tilde{w}^{(1,k-1)} - w^*\|^2 - 2\eta\langle\frac{\sum_{i=1}^{n} \nabla f_i(w_i^{(1,k-1)})}{n}, \tilde{w}^{(1,k-1)} - w^*\rangle + \eta^2\|\frac{\sum_{i=1}^{n} \nabla f_i(w_i^{(1,k-1)})}{n}\|^2.$$

592  As a straightforward corollary of Lemma A.1, A.2 and A.3, we can obtain a similar upper bound in a

593  form once $\eta < \min\{\frac{\beta}{\sqrt{24}K}, \frac{1}{2\beta}\}$

$$\|\tilde{w}^{(1,k)} - w^*\|^2 \leq \|\tilde{w}^{(1,k-1)} - w^*\|^2 + 2\eta\big(F(w^*) - F(\tilde{w}^{(t,k-1)}) + \frac{\beta}{2n}\sum_{i=1}^{n}\|w^{(t,k-1)} - \tilde{w}^{(t,k-1)}\|^2\big)$$

$$+ 2\eta^2\big(\frac{\beta^2 \sum_{i=1}^{n}\|w_i^{(t,k-1)} - \tilde{w}^{(t,k-1)}\|^2}{n} + 2\beta F(\tilde{w}^{(t,k-1)}) - F(w^*)\big)$$

$$\leq \|\tilde{w}^{(1,k-1)} - w^*\|^2 + 2(\eta - 2\beta\eta^2)(F(w^*) - F(\tilde{w}^{(t,k-1)})) + (\beta\eta + 2\beta^2\eta^2) \cdot 4\eta^2 K^2 \tau$$

$$\leq \|\tilde{w}^{(1,k-1)} - w^*\|^2 + 2(\eta - 2\beta\eta^2)(F(w^*) - F(\tilde{w}^{(t,k-1)})) + 8\beta\eta^3 K^2 \tau.$$

(48)

594  In (48), we apply Lemma A.3 and use the fact that $\beta\eta + 2\beta^2\eta^2 \leq 2\beta\eta$.

On the other hand, during the synchronization, it is noted that

$$\mathbb{E}[\bar{w}^{(1)}] = \mathbb{E}[\tilde{w}^{(1,K)} + Q^{(1)}] = \mathbb{E}[\tilde{w}^{(1,K)}].$$

Therefore,

$$\mathbb{E}[\|\bar{w}^{(1)} - w^*\|^2] = \mathbb{E}[\|\bar{w}^{(1)} - \tilde{w}^{(1,K)}\|^2] + \|\tilde{w}^{(1,K)} - w^*\|^2.$$

595  Moreover,

$$\mathbb{E}[\|\bar{w}^{(1)} - \tilde{w}^{(1,K)}\|^2]$$

$$= \mathbb{E}[\eta^2\|\frac{\sum_{k=1}^{K}\sum_{i=1}^{n}(1_i^{(1)} - q)\nabla f_i(w_i^{(1,k-1)})}{nq} - Q^{(1)}\|^2]$$

$$\leq \frac{K\eta^2 \sum_{k=1}^{K}\sum_{i=1}^{n}\|\nabla f_i(w_i^{(1,k-1)})\|^2}{n^2 q} + \bar{\mathcal{Q}}$$

$$\leq \frac{3K\eta^2 \sum_{k=1}^{K}\big\{\sum_{i=1}^{n}\big(\beta^2\|w_i^{(1,k-1)} - \tilde{w}^{(1,k-1)}\|^2\big) + 2\beta n(F(\tilde{w}^{(1,k-1)}) - F(w^*)) + n\tau\big\}}{n^2 q} + \bar{\mathcal{Q}}$$

$$\leq \frac{3K\eta^2\big(4\beta^2\eta^2 K^3 n\tau + 2\beta n\sum_{k=1}^{K}(F(\tilde{w}^{(1,k-1)}) - F(w^*)) + Kn\tau\big\}}{n^2 q} + \bar{\mathcal{Q}}$$

$$= \frac{12K^4\beta^2\eta^4\tau + 6K\beta\eta^2\sum_{k=1}^{K}(F(\tilde{w}^{(1,k-1)}) - F(w^*)) + 3K^2\eta^2\tau}{nq} + \bar{\mathcal{Q}}.$$

(49)

596  In the fifth line of (49), we apply Lemma A.3. From (48),

$$\|\tilde{w}^{(1,K)} - w^*\|^2 \leq \|\bar{w}^{(0)} - w^*\|^2 + 2(\eta - 2\beta\eta^2)\sum_{k=1}^{K}(F(w^*) - F(\tilde{w}^{(t,k-1)})) + 8\beta\eta^3 K^3 \tau. \quad (50)$$

Now, we combine (49) and (50). Once $2(\eta - 2\beta\eta^2) - \frac{6K\beta\eta^2}{nq} \geq 0$, which implies that $\eta \leq \frac{1}{2\beta + 3K\beta/(nq)}$,

$$\mathbb{E}[\|\bar{w}^{(1)} - w^*\|^2] \leq \|\bar{w}^{(0)} - w^*\|^2 + \frac{12K^4\beta^2\eta^4\tau + 3K^2\eta^2\tau}{nq} + 8\beta\eta^3K^3\tau + \bar{\mathcal{Q}}.$$

The remainder of the proof for the $\|\tilde{w}^{(t,k)} - w^*\|$ is straightforward as for arbitrary $t$, $\|\tilde{w}^{(t,0)} - w^*\| = \|\bar{w}^{(t-1)} - w^*\|$. Therefore, by induction reasoning, we have the bound claimed.

# B    Proof of Theorem 3.2: Synchronized-only Convergence of Noisy LSGD in Non-convex Optimization

Based on the smooth assumption of $F(w)$, we have the following classic inequality,

$$
F(\bar{w}^{(t)}) \le F(\bar{w}^{(t-1)}) + \langle \nabla F(\bar{w}^{(t-1)}), \bar{w}^{(t)} - \bar{w}^{(t-1)} \rangle + \frac{\beta}{2}\|\bar{w}^{(t)} - \bar{w}^{(t-1)}\|^2
$$

$$
= F(\bar{w}^{(t-1)}) - \langle \nabla F(\bar{w}^{(t-1)}), \frac{\eta}{nq} \sum_{i \in S^{(t)}} \sum_{k=0}^{K-1} \nabla f_i(w_i^{(t,k)}) - Q^{(t)} \rangle
$$

$$
+ \frac{\beta}{2}\|\frac{\eta}{nq} \sum_{i \in S^{(t)}} \sum_{k=0}^{K-1} \nabla f_i(w_i^{(t,k)}) - Q^{(t)}\|^2
$$

$$
= F(\bar{w}^{(t-1)})
$$

$$
- \frac{\eta}{2}\Big( \sum_{k=0}^{K-1} \big(\|\nabla F(\bar{w}^{(t-1)})\|^2 + \|\frac{1}{nq} \sum_{i \in S^{(t)}} \nabla f_i(w_i^{(t,k)})\|^2 - \|\nabla F(\bar{w}^{(t-1)}) - \frac{1}{nq} \sum_{i \in S^{(t)}} \nabla f_i(w_i^{(t,k)})\|^2\big)\Big)
$$

$$
+ \langle \nabla F(\bar{w}^{(t-1)}), Q^{(t)} \rangle + \frac{\beta}{2}\|\frac{\eta}{nq} \sum_{i \in S^{(t)}} \sum_{k=0}^{K-1} \nabla f_i(w_i^{(t,k)}) - Q^{(t)}\|^2.
$$

$$(51)$$

In (51), we simply use the fact that $\langle a, b \rangle = \frac{\|a\|^2 + \|b\|^2 - \|a-b\|^2}{2}$. For notation simplicity, we will use $g_i^{(t,k)} = \nabla f_i(w_i^{(t,k)})$ and $g^{(t,k)} = \frac{1}{nq} \cdot \sum_{i \in S_t} \nabla f_i(w_i^{(t,k)}) = \frac{1}{nq} \cdot \sum_{i \in S_t} g_i^{(t,k)}$ in the following. Using the generalized AM-GM inequality, where $\langle a, b \rangle \le \frac{1}{2}\big(\gamma\|a\|^2 + \frac{1}{\gamma}\|b\|^2\big)$ for any $\gamma > 0$, on $\langle \nabla F(w^{(t-1)}), Q^{(t)} \rangle$, we have that

$$
\langle \nabla F(w^{(t-1)}), Q^{(t)} \rangle \le \frac{\eta}{4}\|\nabla F(w^{(t-1)})\|^2 + \frac{1}{\eta}\|Q^{(t)}\|^2. \tag{52}
$$

Similarly,

$$
\frac{\beta}{2}\|\frac{\eta}{nq} \sum_{i \in S_t} \sum_{k=0}^{K-1} g_i^{(t,k)} - Q^{(t)}\|^2 \le \beta\big(\eta^2\|\frac{1}{nq} \sum_{i \in S_t} \sum_{k=0}^{K-1} g_i^{(t,k)}\|^2 + \|Q^{(t)}\|^2\big). \tag{53}
$$

Thus, putting together, we have the following by rearranging the terms in (51),

$$
(\frac{\eta K}{2} - \frac{\eta}{4})\|\nabla F(\bar{w}^{(t-1)})\|^2 \le F(\bar{w}^{(t-1)}) - F(\bar{w}^{(t)}) - \underbrace{\big(\frac{\eta}{2} \sum_{k=0}^{K-1} \|g^{(t,k)}\|^2 - \beta\eta^2\|\sum_{k=0}^{K-1} g^{(t,k)}\|^2\big)}_{(A)}
$$

$$
+ \frac{\eta}{2} \sum_{k=0}^{K-1} \|\nabla F(\bar{w}^{(t-1)}) - g^{(t,k)}\|^2 + (\frac{1}{\eta} + \beta)\|Q^{(t)}\|^2. \tag{54}
$$

Still by AM-GM inequality, it is noted that $\|\sum_{k=0}^{K-1} g^{(t,k)}\|^2 \le K \sum_{k=0}^{K-1} \|g^{(t,k)}\|^2$ and therefore term (A) is lower bounded by $(\frac{\eta}{2} - \beta\eta^2 K)\sum_{k=0}^{K-1} \|g^{(t,k)}\|^2$. For a sufficiently small learning rate $\eta$, term (A) is non-negative. Thus, to upper bound $\|\nabla F(w^{(t)})\|^2$, it suffices to keep track of $\|\nabla F(w^{(t)}) - g^{(t,k)}\|^2$.

Now, we imagine the scenario that each agent participates in the $t$-th phase without Poisson sampling and each produces intermediate $w_i^{(t,k)}$ for $i = 1, 2, \cdots, n$ and $k = 1, 2, \cdots, K$. Let $\tilde{w}^{(t,k)} = \frac{1}{n} \sum_{i=1}^{n} w_i^{(t,k)}$. It is not hard to observe that conditional on $\bar{w}^{(t-1)}$, $\mathbb{E}[\tilde{w}^{(t,k)} - \bar{w}^{(t-1)}] =$

$-\eta \mathbb{E}[\sum_{l=0}^{k-1} g^{(t,l)}]$. On the other hand, by AM-GM inequality again,

$$
\begin{aligned}
&\|\nabla F(w^{(t-1)}) - g^{(t,k)}\|^2 \\
&\leq 2\big(\|\nabla F(\bar{w}^{(t-1)}) - \nabla F(\tilde{w}^{(t,k)})\|^2 + \|\nabla F(\tilde{w}^{(t,k)}) - g^{(t,k)}\|^2\big) \\
&\leq 2\big(\beta^2\|\bar{w}^{(t-1)} - \tilde{w}^{(t,k)}\|^2 + \|\nabla F(\tilde{w}^{(t,k)}) - g^{(t,k)}\|^2\big) \\
&= 2\big(\beta^2\|\bar{w}^{(t-1)} - \tilde{w}^{(t,k)}\|^2 + \|\frac{\sum_{i=1}^n (q - 1_i^{(t)})\big(\nabla f_i(\tilde{w}^{(t,k)}) - \nabla f_i(w_i^{(t,k)})\big)}{nq}\|^2\big).
\end{aligned}
\tag{55}
$$

In (55), we use the $\beta$-smooth assumption on $\nabla F(w)$, and $1_i^{(t)}$ is an indicator which equals 1 iff the $i$-th worker/agent is selected in the $t$-th phase with probability $q$, otherwise 0. We first handle the first term $\beta^2\|\bar{w}^{(t)} - \tilde{w}^{(t,k)}\|^2$. With expectation conditional on $\bar{w}^{(t-1)}$,

$$
\begin{aligned}
\mathbb{E}[\|\bar{w}^{(t-1)} - \tilde{w}^{(t,k)}\|^2] &= \mathbb{E}[\eta^2\|\sum_{l=0}^{k-1} g^{(t,l)}\|^2] - \mathbb{E}\big[\| - (\eta \sum_{l=0}^{k-1} g^{(t,l)}) - (\bar{w}^{(t-1)} - \tilde{w}^{(t,k)})\|^2\big] \\
&\leq k\eta^2 \sum_{l=0}^{k-1} \mathbb{E}[\|g^{(t,l)}\|^2]
\end{aligned}
\tag{56}
$$

In (56), we use the following fact about the variance and second moment: for a random vector $v$ whose mean is $\mu$, $\mathbb{E}[\|v\|^2] = \mathbb{E}[\|v - \mu\|^2] + \|\mu\|^2$. As mentioned above, the expectation conditional on $\bar{w}^{(t-1)}$ $\mathbb{E}[\tilde{w}^{(t,k)} - \bar{w}^{(t-1)}] = -\eta \mathbb{E}[\sum_{l=0}^{k-1} g^{(t,l)}]$. Therefore,

$$
2\beta^2 \sum_{k=1}^K \mathbb{E}[\|\bar{w}^{(t-1)} - \tilde{w}^{(t,k)}\|^2] \leq 2\beta^2 \sum_{k=1}^K k\eta^2 \sum_{l=0}^{k-1} \mathbb{E}[\|g^{(t,l)}\|^2] \leq 2\beta^2\eta^2 K^2 \sum_{k=0}^{K-1} \mathbb{E}[\|g^{(t,k)}\|^2].
\tag{57}
$$

Now, combined the same term $\mathbb{E}[\|g^{(t,k)}\|^2]$ in (57) with (A), it is not hard to verifiy that, once $\frac{\eta}{2} - \beta\eta^2 K - \beta^2\eta^3 K^2 \geq 0$, which holds when $\eta < \frac{1}{4\beta K}$, then the expectation

$$
\mathbb{E}\big[\frac{\eta}{2} \cdot 2\beta^2 K^2\eta^2 \sum_{k=0}^{K-1} \|\sum_{l=0}^k g^{(t,l)}\|^2 - (A)\big] \leq 0.
$$

Now, we move our focus to the second term $\|\frac{1}{nq} \cdot \sum_{i=1}^n (q - 1_i^{(t)})\big(\nabla f_i(\tilde{w}^{(t,k)}) - \nabla f_i(w_i^{(t,k)})\big)\|^2$ in (55).

Based on the assumption on Poisson sampling, $1_i^{(t)}$ is independent and $\mathbb{E}[1_i^{(t)}] = q$ for $i = 1, 2, \cdots, n$. Morevoer, $\mathbb{E}[(1_i^{(t)} - q)^2] = q - q^2 < q$. Therefore, with expectation,

$$
\begin{aligned}
&\sum_{k=0}^{K-1} \mathbb{E}\big[\|\frac{\sum_{i=1}^n (q - 1_i^{(t)})\big(\nabla f_i(\tilde{w}^{(t,k)}) - \nabla f_i(w_i^{(t,k)})\big)}{nq}\|^2\big] \\
&= \sum_{k=0}^{K-1} \sum_{i=1}^n \frac{(q - q^2)\mathbb{E}[\|\nabla f_i(\tilde{w}^{(t,k)}) - \nabla f_i(w_i^{(t,k)})\|^2]}{(nq)^2} \leq \sum_{k=0}^{K-1} \sum_{i=1}^n \frac{\beta^2\mathbb{E}[\|\tilde{w}^{(t,k)} - w_i^{(t,k)}\|^2]}{n^2 q}.
\end{aligned}
\tag{58}
$$

In (58), we use the fact for $n$ random independent vectors $v_{[1:n]}$ of zero mean, $\mathbb{E}[\|\sum_{i=1}^n v_i\|^2] = \sum_{i=1}^n \mathbb{E}[\|v_i\|^2]$. On the other hand, we can apply the results of Lemma A.3 to upper bound $\sum_{i=1}^n \mathbb{E}\big[\|w_i^{(t,k)} - \tilde{w}^{(t,k)}\|^2\big]$ by $4\eta^2 k^2 n\tau$ once $\eta < \min\{\frac{\beta}{\sqrt{24}K}, \frac{1}{20\beta}\}$. Now, back to (58), we have that

$$
\sum_{k=0}^{K-1} \sum_{i=1}^n \frac{\beta^2\mathbb{E}[\|\tilde{w}^{(t,k)} - w_i^{(t,k)}\|^2]}{n^2 q} \leq \frac{4\eta^2\tau\beta^2 K^3}{nq}.
$$

With the above preparation, we are finally ready to complete the proof. Back to (54), conditional on $w^{(t-1)}$, with expectation we have that

$$(\frac{\eta K}{2} - \frac{\eta}{4})\|\nabla F(\bar{w}^{(t-1)})\|^2 \leq \mathbb{E}[F(\bar{w}^{(t-1)}) - F(\bar{w}^{(t)})] - (\frac{\eta}{2} - \beta\eta^2 K - \beta^2\eta^3 K^2) \sum_{k=0}^{K-1} \mathbb{E}[\|g^{(t,k)}\|^2]$$

$$+ \frac{\eta}{2} \cdot \frac{8\eta^2\tau\beta^2 K^3}{nq} + (\frac{1}{\eta} + \beta)\|Q^{(t)}\|^2.$$

(59)

Summing up both sides of (59) for $t = 1, 2, ..., T$, with unconditional expectation and averaging, since $\eta K/2 - \eta/4 \geq \eta K/4$ for $K \geq 1$, we obtain that once $\eta < \min\{\frac{\beta}{\sqrt{24K}}, \frac{1}{4\beta K}, \frac{1}{20\beta}\}$,

$$\mathbb{E}[\frac{\sum_{t=1}^T \|\nabla F(\bar{w}^{(t-1)})\|^2}{T}] \leq \frac{4F(\bar{w}^{(0)})}{TK\eta} + \frac{16\eta^2\tau\beta^2 K^2}{nq} + \frac{(1 + \beta\eta)\sum_{t=1}^T \mathbb{E}[\|Q^{(t)}\|^2]}{\eta^2 KT}.$$

Alternatively, especially when the perturbation $Q^{(t)}$ is independent and of zero-mean, we may consider another bound derived as follows. Still, based on the smooth assumption of $F(w)$, if we focus on each cross term between $\nabla F(\bar{w}^{(t-1)})$ and $\nabla f_i(w_i^{(t,k)})$, we have

$$F(\bar{w}^{(t)}) \leq F(\bar{w}^{(t-1)}) + \langle \nabla F(\bar{w}^{(t-1)}), \bar{w}^{(t)} - \bar{w}^{(t-1)}\rangle + \frac{\beta}{2}\|\bar{w}^{(t)} - \bar{w}^{(t-1)}\|^2$$

$$= F(\bar{w}^{(t-1)}) - \langle \nabla F(\bar{w}^{(t-1)}), \frac{\eta}{nq} \sum_{i \in S^{(t)}} \sum_{k=0}^{K-1} \nabla f_i(w_i^{(t,k)}) - Q^{(t)}\rangle$$

$$+ \frac{\beta}{2}\|\frac{\eta}{nq} \sum_{i \in S^{(t)}} \sum_{k=0}^{K-1} \nabla f_i(w_i^{(t,k)}) - Q^{(t)}\|^2$$

$$= F(\bar{w}^{(t-1)})$$

$$- \frac{\eta}{2nq} \cdot (\sum_{i \in S^{(t)}} \sum_{k=0}^{K-1} (\|\nabla F(\bar{w}^{(t-1)})\|^2 + \|\nabla f_i(w_i^{(t,k)})\|^2 - \|\nabla F(\bar{w}^{(t-1)}) - \nabla f_i(w_i^{(t,k)})\|^2))$$

$$+ \langle \nabla F(\bar{w}^{(t-1)}), Q^{(t)}\rangle + \frac{\beta}{2}\|\frac{\eta}{nq} \sum_{i \in S^{(t)}} \sum_{k=0}^{K-1} \nabla f_i(w_i^{(t,k)}) - Q^{(t)}\|^2.$$

(60)

With a similar reasoning as (53), we have the following by rearranging the terms in (60),

$$\frac{\eta K B_t}{2nq}\|\nabla F(\bar{w}^{(t-1)})\|^2 \leq F(\bar{w}^{(t-1)}) - F(\bar{w}^{(t)}) - \underbrace{(\frac{\eta}{2nq} - \frac{\beta\eta^2 B_t K}{(nq)^2}) \sum_{i \in S^{(t)}} \sum_{k=0}^{K-1} \|g_i^{(t,k)}\|^2}_{(A)}$$

$$+ \frac{\eta}{2nq} \sum_{i \in S^{(t)}} \sum_{k=0}^{K-1} \|\nabla F(\bar{w}^{(t-1)}) - g_i^{(t,k)}\|^2 + \beta\|Q^{(t)}\|^2.$$

(61)

For a sufficiently small learning rate $\eta$, term (A) is non-negative. Thus, to upper bound $\|\nabla F(w^{(t)})\|^2$, it suffices to keep track of $\|\nabla F(\bar{w}^{(t-1)}) - g^{(t,k)}\|^2$. Conditional on $\bar{w}^{(t-1)}$, take expectation on both sides of (54) and we have

$$\frac{\eta K}{2}\mathbb{E}[\|\nabla F(\bar{w}^{(t-1)})\|^2] \leq \mathbb{E}[F(\bar{w}^{(t-1)}) - F(\bar{w}^{(t)}) - (\frac{\eta}{2n} - \frac{\beta\eta^2 K}{n}) \sum_{i=1}^n \sum_{k=0}^{K-1} \|g_i^{(t,k)}\|^2$$

$$+ \frac{\eta}{2n} \sum_{i=1}^n \sum_{k=0}^{K-1} \|\nabla F(\bar{w}^{(t-1)}) - g_i^{(t,k)}\|^2 + \beta\|Q^{(t)}\|^2],$$

(62)

since $\mathbb{E}[B_t] = nq$.

By AM-GM inequality again,

$$
\begin{aligned}
&\sum_{i=1}^{n} \|\nabla F(\bar{w}^{(t-1)}) - g_i^{(t,k)}\|^2 \\
&\leq 2 \sum_{i=1}^{n} \left( \|\nabla F(\bar{w}^{(t-1)}) - \nabla f_i(\bar{w}^{(t-1)})\|^2 + \|\nabla f_i(\bar{w}^{(t-1)}) - \nabla f_i(w_i^{(t,k)})\|^2 \right) \\
&\leq 2 \left( n\tau + \beta^2 \sum_{i=1}^{n} \|\bar{w}^{(t-1)} - w_i^{(t,k)}\|^2 \right) \\
&= 2 \left( n\tau + \beta^2 \eta^2 \sum_{i=1}^{n} \| \sum_{l=0}^{k-1} g_i^{(t,l)}\|^2 \right) \leq 2 \left( n\tau + \beta^2 \eta^2 k \sum_{i=1}^{n} \sum_{l=0}^{k-1} \|g_i^{(t,l)}\|^2 \right).
\end{aligned}
\tag{63}
$$

Plugging (63), which suggests that

$$
\frac{\eta}{2n} \sum_{i=1}^{n} \sum_{k=0}^{K-1} \|\nabla F(\bar{w}^{(t-1)}) - g_i^{(t,k)}\|^2 \leq \eta \left( \tau K + \frac{\beta^2 \eta^2 K^2}{n} \sum_{i=1}^{n} \sum_{k=0}^{K-1} \|g_i^{(t,k)}\|^2 \right),
$$

back to (62), we have that

$$
\begin{aligned}
\frac{\eta K}{2} \mathbb{E}[\|\nabla F(\bar{w}^{(t-1)})\|^2] \leq &\mathbb{E}\Big[ F(\bar{w}^{(t-1)}) - F(\bar{w}^{(t)}) - \Big( \frac{\eta}{2n} - \frac{\beta \eta^2 K}{n} - \frac{\beta^2 \eta^3 K^2}{n} \Big) \sum_{i=1}^{n} \sum_{k=0}^{K-1} \|g_i^{(t,k)}\|^2 \\
&+ \eta \tau K + \beta \|Q^{(t)}\|^2 \Big],
\end{aligned}
\tag{64}
$$

Therefore, when $\frac{\eta}{2n} - \frac{\beta \eta^2 K}{n} - \frac{\beta^2 \eta^3 K^2}{n} \geq 0$, which requires that $\eta \leq \frac{1}{2\beta K}$, we have

$$
\mathbb{E}[\|\nabla F(\bar{w}^{(t-1)})\|^2] \leq 2 \cdot \mathbb{E}\Big[ \frac{F(\bar{w}^{(t-1)}) - F(\bar{w}^{(t)})}{\eta K} + \tau + \frac{\beta}{\eta K} \|Q^{(t)}\|^2 \Big].
\tag{65}
$$

Now, we sum up (65) both sides for $t = 1, 2, \cdots, T$ and average them, we have that

$$
\mathbb{E}\Big[ \frac{\sum_{t=1}^{T} \|\nabla F(\bar{w}^{(t-1)})\|^2}{T} \Big] \leq 2 \cdot \mathbb{E}\Big[ \frac{F(\bar{w}^{(t-1)})}{\eta T K} + \tau + \frac{\sum_{t=1}^{T} \beta \mathbb{E}[\|Q^{(t)}\|^2]}{\eta T K} \Big].
\tag{66}
$$

# C  Proof of Theorem 4.1: Utility of DP-LSGD in General Convex Optimization

We first focus on the clipped local update $\mathcal{CP}(\Delta w_i^{(t)}, c) = \mathcal{CP}(w_i^{(t,K)} - \bar{w}^{(t-1)}, c)$ in the $t$-th phase if the $i$-th sample gets selected. Since the local update before clipping is essentially the sum of gradient scaled by the learning rate $-\eta$, therefore,

$$\mathcal{CP}(w_i^{(t,K)} - \bar{w}^{(t-1)}, c) = \mathcal{CP}(-\eta \sum_{k=0}^{K-1} \nabla f_i(w_i^{(t,k)}), c) = -\eta_i^{(t)} \sum_{k=0}^{K-1} \nabla f_i(w^{(t,k)}), \quad (67)$$

where $\eta_i^{(t)} = \eta \cdot \min\{1, \frac{c}{\|\sum_{k=0}^{K-1} \nabla f_i(w_i^{(t,k)})\|}\}$ is determined by the clipping threshold, and thus $\eta_i^{(t)} \leq \eta$. Based on Definition 4.1,

$$\eta - \eta_i^{(t)} = \eta \cdot (1 - \frac{c}{c + \mathbf{1}(\|\Delta w_i^{(t)}\| > c) \cdot (\|\Delta w_i^{(t)}\| - c))} = \eta \cdot \frac{\Psi_i^{(t)}}{c + \Psi_i^{(t)}}, \quad (68)$$

where $\Psi_i^{(t)} = \max\{0, \|\Delta w_i^{(t)}\| - c\}$ represents the incremental norm of the local update from the $i$-th sample in the $t$-th phase. For simplicity, we will use $\Delta \Psi_i^{(t)}$ to denote $\frac{\Psi_i^{(t)}}{c + \Psi_i^{(t)}}$.

Now, we consider two virtual sequences:

    a) $w_i^{\prime(t,0)} = \bar{w}^{(t-1)}$ and $w_i^{\prime(t,k)} = w_i^{\prime(t,k-1)} - \eta_i^{(t)} \nabla f_i(w_i^{(t,k-1)})$, which represents a sequence of iterates based on the gradients $\nabla f_i(w_i^{(t,k-1)})$ but scaled by $\eta_i^{(t)}$ instead of constant $\eta$ for each $i$;

    b) We use $\hat{w}^{(t,k)} = \frac{1}{nq} \cdot \sum_{i=1}^{n} \mathbf{1}_i^{(t)} \cdot w_i^{\prime(t,k)}$ to represent the average of $w_i^{\prime(t,k)}$ for those indices $i$ selected in the $t$-th phase. Here, $\mathbf{1}_i^{(t)} = 1$ iff the $i$-th sample is selected in the $t$-th phase. Similarly, we define $\tilde{w}^{(t,k)} = \frac{1}{n} \cdot w_i^{\prime(t,k)}$ to be the average of all $w_i^{\prime(t,k)}$ for $i = 1, 2, \cdots, n$. It is not hard to observe that $\tilde{w}_i^{(t,K)} = \bar{w}^{(t-1)} + \mathcal{CP}(\Delta w_i^{(t)}, c)$, and consequently conditional on $\bar{w}^{(t-1)}$, $\mathbb{E}[\bar{w}^{(t)}] = \mathbb{E}[\hat{w}^{(t,K)}] = \tilde{w}^{(t,K)}$ since the independent DP noise satisfies that $\mathbb{E}[Q^{(t)}] = 0$.

In the following, we unravel $\|\tilde{w}^{(t,k)} - u\|^2$ for arbitrary $u$ and obtain

$$\|\hat{w}^{(t,k)} - u\|^2$$

$$= \|\hat{w}^{(t,k-1)} - \sum_{i=1}^{n} \frac{\eta_i^{(t)} \cdot \mathbf{1}_i^{(t)} \cdot \nabla f_i(w_i^{(t,k-1)})}{nq} - u\|^2$$

$$= \|\hat{w}^{(t,k-1)} - u\|^2 - \frac{2}{nq} \cdot \sum_{i=1}^{n} \eta_i^{(t)} \mathbf{1}_i^{(t)} \langle \tilde{w}^{(t,k-1)} - u, \nabla f_i(w_i^{(t,k-1)}) \rangle + \|\frac{\sum_{i=1}^{n} \eta_i^{(t)} \mathbf{1}_i^{(t)} \nabla f_i(w_i^{(t,k-1)})}{nq}\|^2.$$
$$(69)$$

We first work on the last term of (69). With the fact that $\eta_i^{(t)} \leq \eta$, conditional on $\bar{w}^{(t-1)}$,

$$\mathbb{E}[\|\frac{\sum_{i=1}^{n} \eta_i^{(t)} \mathbf{1}_i^{(t)} \nabla f_i(w_i^{(t,k-1)})}{nq}\|^2]$$

$$= \mathbb{E}[\|\frac{\sum_{i=1}^{n} \eta_i^{(t)} \mathbf{1}_i^{(t)} \nabla f_i(w_i^{(t,k-1)})}{nq} - \frac{\sum_{i=1}^{n} \eta_i^{(t)} \nabla f_i(w_i^{(t,k-1)})}{n} + \frac{\sum_{i=1}^{n} \eta_i^{(t)} \nabla f_i(w_i^{(t,k-1)})}{n}\|^2]$$

$$\leq 2 \cdot \mathbb{E}[\|\frac{\sum_{i=1}^{n} \eta_i^{(t)} (\mathbf{1}_i^{(t)} - q) \nabla f_i(w_i^{(t,k-1)})}{nq}\|^2] + 2 \cdot \|\frac{\sum_{i=1}^{n} \eta_i^{(t)} \nabla f_i(w_i^{(t,k-1)})}{n}\|^2$$

$$\leq \frac{2(q - q^2) \sum_{i=1}^{n} \|\eta_i^{(t)} \nabla f_i(w_i^{(t,k-1)})\|^2}{(nq)^2} + \frac{2 \sum_{i=1}^{n} \|\eta_i^{(t)} \nabla f_i(w_i^{(t,k-1)})\|^2}{n}$$

$$\leq \frac{4\eta^2 \sum_{i=1}^{n} \|\nabla f_i(w_i^{(t,k-1)})\|^2}{n}$$
$$(70)$$

which can be further bounded via Lemma A.1 as

$$4\eta^2\Big(\frac{3\beta^2\sum_{i=1}^n\|w_i^{(t,k-1)}-\tilde{w}^{(t,k-1)}\|^2}{n}+\min\{6\beta F(\tilde{w}^{(t,k-1)})-F(w^*),3\beta^2\|\tilde{w}^{(t,k-1)}-w^*\|^2\}+3\tau\Big).\tag{71}$$

Now, we move our focus to the second term of (69). Still, with a similar reasoning as Lemma A.2,

$$\mathbb{E}\Big[\frac{-2}{nq}\cdot\sum_{i=1}^n\mathbf{1}_i^{(t)}\eta_i^{(t)}\langle\tilde{w}^{(t,k-1)}-u,\nabla f_i(w_i^{(t,k-1)})\rangle\Big]$$

$$=\Big[\frac{-2}{n}\cdot\sum_{i=1}^n\eta(1-\Delta\Psi_i^{(t)})\langle\tilde{w}^{(t,k-1)}-u,\nabla f_i(w_i^{(t,k-1)})\rangle\Big]$$

$$\leq\frac{2}{n}\sum_{i=1}^n\eta(1-\Delta\Psi_i^{(t)})\big(f_i(u)-f_i(\tilde{w}^{(t,k-1)})+\frac{\beta}{2}\|w_i^{(t,k-1)}-\tilde{w}^{(t,k-1)}\|^2\big)$$

$$\leq 2\eta\big(F(u)-F(\tilde{w}^{(t,k-1)})\big)+\frac{\beta}{2n}\cdot\sum_{i=1}^n(1-\Delta\Psi_i^{(t)})\|w^{(t,k-1)}-\tilde{w}^{(t,k-1)}\|^2\big)$$

$$-\frac{2}{n}\cdot\sum_{i=1}^n\eta\Delta\Psi_i^{(t)}\big(F(u)-F(\tilde{w}^{(t,k-1)})\big)+\sum_{i=1}^n\frac{2}{n}\big(\eta\Delta\Psi_i^{(t)}\big)\cdot 2\gamma$$

$$\leq 2\eta(1-\frac{\sum_{i=1}^n\Delta\Psi_i^{(t)}}{n})\big(F(u)-F(\tilde{w}^{(t,k-1)})\big)+\big(\frac{\beta\eta}{n}\sum_{i=1}^n\|w_i^{(t,k-1)}-\tilde{w}^{(t,k-1)}\|^2\big)+\frac{4\eta\gamma\sum_{i=1}^n\Delta\Psi_i^{(t)}}{n}.\tag{72}$$

In the fourth line of (72), we use the $\gamma$-similarity assumption from Assumption 4.2. In the following, we will use $\Delta\bar{\Psi}^{(t)}=\frac{\sum_{i=1}^n\Delta\Psi_i^{(t)}}{n}$ for simplicity.

Next, we work on the upper bound of $\sum_{i=1}^n\|w_i^{(t,k-1)}-\tilde{w}^{(t,k-1)}\|^2$. Similar to Lemma A.3,

$$\sum_{i=1}^n\|\tilde{w}^{(t,k-1)}-w_i^{(t,k-1)}\|^2$$

$$=\sum_{i=1}^n\|\frac{\sum_{l=0}^{k-1}\sum_{j=1}^n\eta_j^{(t)}\nabla f_j(w_j^{(t,l)})}{n}-\eta\cdot\sum_{l=0}^{k-1}\nabla f_i(w_i^{(t,l)})\|^2$$

$$\leq 2\sum_{i=1}^n\big(\eta^2\|\frac{\sum_{l=0}^{k-1}\sum_{j=1}^n(\nabla f_j(w_j^{(t,l)})-\nabla f_i(w_i^{(t,l)}))}{n}\|^2+\|\frac{\sum_{l=0}^{k-1}\sum_{j=1}^n(\eta-\eta_j^{(t)})\nabla f_j(w_j^{(t,l)})}{n}\|^2\big)\tag{73}$$

For the first term in (73), we have studied it in Lemma A.3, where once $\eta^2<\frac{\beta^2}{24K^2}$,

$$\sum_{i=1}^n\|\eta\cdot\frac{\sum_{l=0}^{k-1}\sum_{j=1}^n\nabla f_j(w_j^{(t,l)})}{n}-\eta\cdot\sum_{l=0}^{k-1}\nabla f_i(w_i^{(t,l)})\|^2\leq 4\eta^2k^2n\tau.\tag{74}$$

Plugging (74) back to (73), since $(\eta-\eta_j^{(t)})^2\leq\eta^2$, and we apply the similar decomposition trick used in (71), we have that

$$\sum_{i=1}^n\frac{\|\tilde{w}^{(t,k-1)}-w_i^{(t,k-1)}\|^2}{n}\leq 8\eta^2k^2n\tau+\frac{1}{n}\cdot\frac{2k\eta^2\sum_{l=0}^{k-1}\sum_{i=1}^n\|\nabla f_i(w_i^{(t,l)})\|^2}{n}$$

$$\leq 8\eta^2k^2\tau$$

$$+\frac{6k\eta^2}{n}\sum_{l=0}^{k-1}\big(\beta^2\|\tilde{w}^{(t,l)}-w_i^{(t,l)}\|^2+\min\big\{2\beta\big(F(\tilde{w}^{(t,l)})-F(w^*)\big),\beta^2\|\tilde{w}^{(t,l)}-w^*\|^2\big\}+\tau\big)$$

$$\leq 14\eta^2k^2\tau+\frac{6k\eta^2}{n}\sum_{l=0}^{k-1}\big(\beta^2\|\tilde{w}^{(t,l)}-w_i^{(t,l)}\|^2+\min\big\{2\beta\big(F(\tilde{w}^{(t,l)})-F(w^*)\big),\beta^2\|\tilde{w}^{(t,l)}-w^*\|^2\big\}\big),\tag{75}$$

given that $n \geq 1$. Thus, when $\eta$ is selected small enough such that $\eta \leq \min\{\frac{\sqrt{n}}{\sqrt{30}K\beta}, \frac{1}{\sqrt{6}K}\}$, for any $k_0 \leq K$, by induction it is not hard to verifiy that

$$
\begin{aligned}
&\frac{\sum_{i=1}^{n} \|w_i^{(t,k_0-1)} - \tilde{w}^{(t,k_0-1)}\|^2}{n} \\
&\leq 15\eta^2 k_0^2 \tau + \frac{12\eta^2 k_0}{n} \Big( \sum_{l=0}^{k_0-1} \min\big\{ 2\beta\big(F(\tilde{w}^{(t,l)}) - F(w^*)\big), \beta^2 \|\tilde{w}^{(t,l)} - w^*\|^2 \big\} \Big).
\end{aligned}
\tag{76}
$$

Now, we put (71), (72) and (76) together, and go back to (69)

$$
\begin{aligned}
&[\eta(1 - \Delta\bar{\Psi}^{(t)})\big(F(\tilde{w}^{(t,k-1)}) - F(u)\big)] \leq \mathbb{E}[\|\hat{w}^{(t,k-1)} - u\|^2 - \|\hat{w}^{(t,k)} - u\|^2] + 4\eta\gamma\Delta\bar{\Psi}^{(t)} \\
&+ (12\eta^2\beta^2 + \beta\eta)\Big(15\eta^2 k^2\tau + \frac{12\eta^2 k}{n}\Big(\sum_{l=0}^{k-1} \min\big\{2\beta\big(F(\tilde{w}^{(t,l)}) - F(w^*)\big), \beta^2\|\tilde{w}^{(t,l)} - w^*\|^2\big\}\Big)\Big) \\
&+ 12\eta^2 \min\big\{2\beta\big(F(\tilde{w}^{(t,k-1)}) - F(w^*)\big), \beta^2\|\tilde{w}^{(t,l)} - w^*\|^2\big\} + 12\eta^2\tau
\end{aligned}
\tag{77}
$$

When $\eta$ is small enough such that $12\eta^2\beta^2 + \beta\eta \leq 2\beta\eta$, (77) can be simplified as

$$
\begin{aligned}
&[\eta(1 - \Delta\bar{\Psi}^{(t)})\big(F(\tilde{w}^{(t,k-1)}) - F(u)\big)] \leq \mathbb{E}[\|\hat{w}^{(t,k-1)} - u\|^2 - \|\hat{w}^{(t,k)} - u\|^2] + 4\eta\gamma\Delta\bar{\Psi}^{(t)} \\
&+ (10K^2\beta\eta^3 + 12\eta^2)\tau + \frac{24K\beta\eta^3}{n}\sum_{l=0}^{k-1} \min\big\{2\beta\big(F(\tilde{w}^{(t,l)}) - F(w^*)\big), \beta^2\|\tilde{w}^{(t,l)} - w^*\|^2\big\} \\
&+ 12\eta^2 \min\big\{2\beta\big(F(\tilde{w}^{(t,k-1)}) - F(w^*)\big), \beta^2\|\tilde{w}^{(t,l)} - w^*\|^2\big\}.
\end{aligned}
\tag{78}
$$

The remainder of the proof is almost the same as that for Theorem 4.1. On one hand, it is noted that

$$
1 - \Delta\bar{\Psi}^{(t)} = \sum_{i=1}^{n} \frac{1}{n} \cdot \frac{c}{c + \Psi_i^{(t)}} \geq \frac{c}{c + \frac{\Psi_i^{(t)}}{n}},
\tag{79}
$$

since $1/(1+x)$ is convex regarding $x$. Therefore, $\mathbb{E}[(1 - \Delta\bar{\Psi}^{(t)})] \geq \frac{c}{c+\mathcal{B}}$ and $\mathbb{E}[\Delta\bar{\Psi}^{(t)}] \leq \frac{\mathcal{B}}{c+\mathcal{B}}$ by Assumption 4.1 that $\mathbb{E}[\frac{\sum_{i=1}^{n} \Psi_i^{(t)}}{n}] \leq \mathcal{B}$.

Therefore, for sufficiently small $\eta = O(n/K^2)$ such that $24\eta^2\beta + \frac{48K^2\beta^2\eta^3}{n} \leq \frac{c\eta}{2(c+\mathcal{B})}$, summing up both sides of (77) for $k = 1, 2, \cdots, K$ and $t = 1, 2, \cdots, T$ with $u = w^*$, and take the zero-mean independent DP noise into accountant where $\bar{w}^{(t)} = \hat{w}^{(t,K)} + Q^{(t)}$, we have

$$
\begin{aligned}
&\mathbb{E}\Big[\frac{\sum_{t=1}^{T} \sum_{k=1}^{K-1} \frac{c}{2(c+\mathcal{B})}\big(F(\tilde{w}^{(t,k-1)}) - F(w^*)\big)}{TK}\Big] \\
&\leq \frac{\|\bar{w}^{(0)} - w^*\|^2}{TK\eta} + (30K^2\beta\eta^2 + 12\eta)\tau + \frac{4\gamma\mathcal{B}}{c+\mathcal{B}} + \frac{\sigma^2 d}{K\eta}.
\end{aligned}
\tag{80}
$$

To obtain the convergence guarantee of $\bar{w}^{(T)}$, we similarly imagine a virtual step where we implement one additional full gradient descent using the entire set and we have that

$$
\begin{aligned}
\|\tilde{w}^{(T+1,1)} - u\|^2 &= \|\bar{w}^{(T)} - u - \eta \cdot \frac{\sum_{i=1}^{n} \nabla f_i(\tilde{w}^{(T,K)})}{n}\|^2 \\
&\leq \|\bar{w}^{(T)} - u\|^2 - 2\eta\big(F(\bar{w}^{(T)}) - F(u)\big) + \eta^2\|\nabla F(\bar{w}^{(T)}) - \nabla F(w^*)\|^2 \\
&\leq \|\bar{w}^{(T)} - w^*\|^2 - 2\eta\big(F(\bar{w}^{(T)}) - F(u)\big) + \eta^2 \min\{\beta^2\|\bar{w}^{(T)} - w^*\|^2, 2\beta(F(\bar{w}^T) - F(w^*))\}).
\end{aligned}
\tag{81}
$$

Therefore, for small enough $\eta$, such that $\eta - \eta^2\beta > 0.5\eta$, we combine (80) and (81) with $u = w^*$, and have

$$
\begin{aligned}
&\mathbb{E}\Big[\frac{\sum_{t=1}^{T} \sum_{k=1}^{K} \frac{c}{2(c+\mathcal{B})}\big(F(\tilde{w}^{(t,k-1)}) - F(w^*)\big) + \frac{\mathcal{B}}{2(c+\mathcal{B})}\big(F(\bar{w}^{(T)}) - F(w^*)\big)}{TK+1}\Big] \\
&\leq \frac{\|\bar{w}^{(0)} - w^*\|^2}{(TK+1)\eta} + (30K^2\beta\eta^2 + 12\eta)\tau + \frac{4\gamma\mathcal{B}}{c+\mathcal{B}} + \frac{\sigma^2 d}{K\eta}.
\end{aligned}
\tag{82}
$$

685  Similarly, it is noted that conditional on $\bar{w}^{(t-1)}$, we still have that

$$\mathbb{E}[\|\hat{w}^{(t,k)} - u\|^2] = \mathbb{E}[\|\hat{w}^{(t,k)} - \tilde{w}^{(t,k)}\|^2] + \|\tilde{w}^{(t,k)} - u\|^2, \tag{83}$$

686  and for $\mathbb{E}[\|\hat{w}^{(t,k)} - \tilde{w}^{(t,k)}\|^2]$ for any $t$ and $k$, we use $\tilde{w}'^{(t,k)} = \frac{1}{n} \cdot \sum_{i=1}^{n} w_i^{(t,k)}$,

$$\mathbb{E}[\|\hat{w}^{(t,k)} - \tilde{w}^{(t,k)}\|^2] = \mathbb{E}[\|(\hat{w}^{(t,k)} - \bar{w}^{(t-1)}) - (\tilde{w}^{(t,k)} - \bar{w}^{(t-1)})\|^2]$$

$$= \mathbb{E}[\|\sum_{i=1}^{n} \frac{\eta_i^{(t)}}{\eta} \cdot \frac{\mathbf{1}_i^{(t)} - q}{nq} \cdot \sum_{l=0}^{k-1} \nabla f_i(w_i^{(t,l)})\|^2] \leq \frac{k}{n^2 q} \sum_{i=1}^{n} \sum_{l=0}^{k-1} \|\nabla f_i(w_i^{(t,l)})\|^2, \tag{84}$$

687  since $\eta_i^{(t)} \leq \eta$. Therefore, by (24), we also have that

$$\mathbb{E}[\|\hat{w}^{(t,k)} - \tilde{w}^{(t,k)}\|^2] \leq \frac{3K\eta^2}{nq}\left(4\beta^2 K^3 \tau \eta^2 + K\tau + \sum_{l=0}^{k-1} \beta^2 \|\tilde{w}^{(t,l)} - w^*\|^2\right) \tag{85}$$

688  Now, using (71) and (83), (78) can be rewritten as

$$[\eta(1 - \Delta\bar{\Psi}^{(t)})\left(F(\tilde{w}^{(t,k-1)}) - F(u)\right)]$$

$$\leq \mathbb{E}[\|\tilde{w}^{(t,k-1)} - u\|^2 - \|\tilde{w}^{(t,k)} - u\|^2 + \|\tilde{w}^{(t,k-1)} - \hat{w}^{(t,k-1)}\|^2 - \|\tilde{w}^{(t,k)} - \hat{w}^{(t,k)}\|]$$

$$+ \frac{\eta^2 K}{nq} \sum_{l=1}^{k} \left(\frac{3\beta^2 \sum_{i=1}^{n} \|w_i^{(t,l-1)} - \tilde{w}^{(t,k-1)}\|^2}{n} + \min\{6\beta F(\tilde{w}^{(t,k-1)}) - F(w^*), 3\beta^2\|\tilde{w}^{(t,k-1)} - w^*\|^2\} + 3\tau\right)$$

$$+ (10K^2\beta\eta^3 + 12\eta^2)\tau + \frac{24K\beta\eta^3}{n} \sum_{l=0}^{k-1} \min\left\{2\beta\left(F(\tilde{w}^{(t,l)}) - F(w^*)\right), \beta^2\|\tilde{w}^{(t,l)} - w^*\|^2\right\})$$

$$+ 12\eta^2 \min\left\{2\beta\left(F(\tilde{w}^{(t,k-1)}) - F(w^*)\right), \beta^2\|\tilde{w}^{(t,l)} - w^*\|^2\right\}. \tag{86}$$

689  On the other hand, if we select $u = \tilde{w}^{(t_0,k_0)}$ for some $t_0 \in [1:T]$ and $k_0 \in [0, K-1]$ in (86), when
690  $K^2 = O(nq)$,

$$\mathbb{E}[\frac{\sum_{(t,k)\in\mathcal{C}} \frac{c}{2(c+\mathcal{B})}\left(F(\tilde{w}^{(t,k)}) - F(\tilde{w}^{(t_0,k_0)})\right) + \frac{c}{2(c+\mathcal{B})}\left(F(\bar{w}^T) - F(\tilde{w}^{(t_0,k_0)})\right)}{(T - t_0 + 1)K - k_0 + 1}]$$

$$\leq O(1) \cdot \left\{ \frac{\frac{3K\eta}{nq}\left(4\beta^2 K^3\tau\eta^2 + K\tau + \sum_{l=0}^{k-1} \beta^2\|\tilde{w}^{(t,l)} - w^*\|^2\right)}{(T - t_0 + 1)K - k_0 + 1} \right.$$

$$\frac{K\beta^3\eta^2}{n}\left(\frac{\sum_{(t,k)\in\mathcal{C}} \sum_{l=0}^{K-1} \mathbb{E}[\|\tilde{w}^{(t,l)} - w^*\|^2]}{(T - t_0 + 1)K - k_0 + 1}\right) + (K^2\beta\eta^2 + \eta)\tau \tag{87}$$

$$\left. + \frac{\gamma\mathcal{B}}{(c+\mathcal{B})} + \frac{\sigma^2 d}{\eta} + \eta\beta^2 \frac{\sum_{(t,k)\in\mathcal{C}} \mathbb{E}[\|\tilde{w}^{(t,k-1)} - w^*\|^2] + \mathbb{E}[\|\bar{w}^{(T)} - w^*\|^2]}{(T - t_0 + 1)K - k_0 + 1} \right\},$$

691  where $\mathcal{C} = \left((t_0, k), k = k_0, \cdots, K-1\right) \cup \left((t,k), t = t_0 + 1, \cdots, T, k = 0, \cdots, K-1\right)$. In the
692  following, we may apply a similar reasoning as Lemma A.4 to derive the following results.

**Lemma C.1.** *Provided sufficiently small $\eta = o(1/K)$, for any $t \in [1:T]$ and $k \in [0:K-1]$*

$$\mathbb{E}[\|\tilde{w}^{(t,k)} - w^*\|^2] = O\left(\|\bar{w}^{(0)} - w^*\|^2 + TK\left(\eta\gamma\frac{\mathcal{B}}{c+\mathcal{B}} + \eta^3 K^2\tau + \eta^2\tau + \frac{K\tau\eta^2}{nq}\right) + T\sigma^2 d\right).$$

693

694  By Lemma (C.1),

$$\frac{24K\beta^3\eta^2}{n} \cdot \frac{\sum_{(t,k)\in\mathcal{C}} \sum_{l=0}^{K-1} \mathbb{E}[\|\tilde{w}^{(t,l)} - w^*\|^2] + \mathbb{E}[\|\bar{w}^{(T)} - w^*\|^2]}{(T - t_0 + 1)K - k_0}$$

$$\leq \frac{K^2\beta^3\eta^2}{n} \cdot O\left(\|\bar{w}^{(0)} - w^*\|^2 + TK\left(\eta\gamma\frac{\mathcal{B}}{c+\mathcal{B}} + \eta^3 K^2\tau + \eta^2\tau + \frac{K\tau\eta^2}{nq}\right) + T\sigma^2 d\right). \tag{88}$$

On the other hand, we have

$$
12\eta\beta^2 \frac{\sum_{t=t_0}^{T} \sum_{k=k_0+1}^{K-1} \mathbb{E}[\|\tilde{w}^{(t,k-1)} - w^*\|^2]}{(T - t_0 + 1)K - k_0}.
$$
$$
\leq \eta \cdot O\big(\|\bar{w}^{(0)} - w^*\|^2 + TK\big(\eta\gamma\frac{\mathcal{B}}{c + \mathcal{B}} + \eta^3 K^2\tau + \eta^2\tau + \frac{K\tau\eta^2}{nq}\big) + T\sigma^2 d\big).
$$

(89)

Now, we can apply the last iterate trick in Lemma A.5. Let $y_j = \frac{c}{2(c+\mathcal{B})}\mathbb{E}[\big(F(\tilde{w}^{(t,k)}) - F(w^*)\big)]$ for $j = (t-1)K+k+1$ for $t = 1, 2, \cdots, T$ and $k = 0, 1, \cdots, K-1$, and $y_{TK+1} = \frac{c}{2(c+\mathcal{B})}\mathbb{E}[F(\bar{w}^{(T)}) - F(w^*)]$.

$$
\begin{aligned}
y_{TK+1} &= \mathbb{E}\big[\frac{c}{2(c + \mathcal{B})}(F(\bar{w}^{(T)}) - F(w^*))\big] \\
&= \frac{\sum_{j=1}^{TK+1} y_j}{TK + 1} + \sum_{j=1}^{TK} \frac{1}{j + 1} \cdot \frac{\sum_{l=TK+1-j}^{TK+1}(y_l - y_{TK+1-j})}{j} \\
&\leq \tilde{O}\big((\eta + \frac{\eta^2 K^2}{n} + \frac{K^2\eta}{nq} + \frac{1}{TK\eta}) \cdot \|\bar{w}^{(0)} - w^*\|^2 \\
&\quad + TK(\frac{K^2\eta^2}{n} + \frac{K^2\eta}{nq} + \eta) \cdot \big((1 + K^2\eta + \frac{K}{nq})\eta^2\tau + \eta\frac{\gamma\mathcal{B}}{c + \mathcal{B}}\big) + \frac{K\eta}{nq}\big(\beta^2 K^3\tau\eta^2 + K\tau\big) \\
&\quad + (\frac{K^2\eta}{nq} + \frac{TK^2\eta^2}{n} + T\eta + 1/\eta)\sigma^2 d\big) \\
&= \tilde{O}\big((\frac{1}{\sqrt{TK}} + \frac{K}{nT})\|\bar{w}^{(0)} - w^*\|^2 + (\frac{K}{nT} + \frac{1}{\sqrt{TK}})(1 + \frac{K^{3/2}}{\sqrt{T}} + \frac{K}{nq})\tau + (K^2\eta^3 + \eta)\tau \\
&\quad + (\frac{K^{3/2}}{\sqrt{T}n} + 1)\frac{\gamma\mathcal{B}}{c + \mathcal{B}} + \sqrt{TK}\sigma^2 d\big) \\
&= \tilde{O}\big(\frac{\|\bar{w}^{(0)} - w^*\|^2}{\sqrt{TK}} + (\frac{1}{\sqrt{TK}} + \frac{K}{T})\tau + \frac{\gamma\mathcal{B}}{c + \mathcal{B}} + \sqrt{TK}\sigma^2 d\big).
\end{aligned}
$$

(90)

when we select $\eta = O(1/\sqrt{TK})$, $K = O(nq)$ and $K = O(T)$. This completes the proof.

## C.1 Proof of Lemma C.1

From (69), by letting $u = w^*$, given $\bar{w}^{(t-1)}$, we have that

$$
\begin{aligned}
&\|\tilde{w}^{(t,k)} - u\|^2 \\
&= \|\tilde{w}^{(t,k-1)} - \sum_{i=1}^{n} \frac{\eta_i^{(t)} \cdot \nabla f_i(w_i^{(t,k-1)})}{n} - w^*\|^2 \\
&= \|\tilde{w}^{(t,k-1)} - w^*\|^2 - \frac{2}{n} \cdot \sum_{i=1}^{n} \eta_i^{(t)}\langle\tilde{w}^{(t,k-1)} - w^*, \nabla f_i(w_i^{(t,k-1)})\rangle + \|\frac{\sum_{i=1}^{n} \eta_i^{(t)}\nabla f_i(w_i^{(t,k-1)})}{n}\|^2.
\end{aligned}
$$

(91)

By (72) and (70), (91) can be further bounded by

$$\|\tilde{w}^{(t,k)} - w^*\|^2$$

$$= \|\tilde{w}^{(t,k-1)} - w^*\|^2 + 2\eta(1 - \Delta\bar{\Psi}^{(t)})\big(F(w^*) - F(\tilde{w}^{(t,k-1)})\big) + \big(\frac{\beta\eta}{n}\sum_{i=1}^{n}\|w_i^{(t,k-1)} - \tilde{w}^{(t,k-1)}\|^2\big)$$

$$+ 4\eta\gamma\Delta\bar{\Psi}^{(t)} + \eta^2\big(\frac{3\beta^2\sum_{i=1}^{n}\|w_i^{(t,k-1)} - \tilde{w}^{(t,k-1)}\|^2}{n} + 6\beta(F(\tilde{w}^{(t,k-1)}) - F(w^*)) + 3\tau\big)$$

$$\leq \|\tilde{w}^{(t,k-1)} - w^*\|^2 - \big(2\eta(1 - \Delta\bar{\Psi}^{(t)}) - 6\beta\eta^2\big)\big(F(\tilde{w}^{(t,k-1)}) - F(w^*)\big)$$

$$+ (\eta\beta + 3\eta^2\beta^2)\frac{\sum_{i=1}^{n}\|w_i^{(t,k-1)} - \tilde{w}^{(t,k-1)}\|^2}{n} + 4\eta\gamma\Delta\bar{\Psi}^{(t)} + 3\eta^2\tau$$

$$\leq \|\tilde{w}^{(t,k-1)} - w^*\|^2 - \big(2\eta(1 - \Delta\bar{\Psi}^{(t)}) - 6\beta\eta^2\big)\big(F(\tilde{w}^{(t,k-1)}) - F(w^*)\big)$$

$$+ (\eta\beta + 3\eta^2\beta^2)\big(15\eta^2 k^2\tau + \frac{12\eta^2 k}{n}\big(\sum_{l=0}^{k-1}\beta\big(F(\tilde{w}^{(t,l)}) - F(w^*)\big)\big) + 4\eta\gamma\Delta\bar{\Psi}^{(t)} + 3\eta^2\tau. \tag{92}$$

On the other hand, as for $\|\bar{w}^{(t+1)} - w^*\|$, we have that

$$\mathbb{E}[\|\bar{w}^{(t)} - w^*\|^2] = \mathbb{E}[\|\bar{w}^{(t)} - \tilde{w}^{(t,K)}\|^2] + \mathbb{E}[\|\tilde{w}^{(t,K)} - w^*\|^2]$$

$$= \mathbb{E}[\|\frac{\sum_{k=1}^{K}\sum_{i=1}^{n}(1_i^{(1)} - q)\eta_i^{(t)}\nabla f_i(w_i^{(t,k-1)})}{nq}\|^2] + \mathbb{E}[\|\tilde{w}^{(t,K)} - w^*\|^2] + \sigma^2 d$$

$$\leq \frac{K\eta^2\sum_{k=1}^{K}\sum_{i=1}^{n}\|\nabla f_i(w_i^{(t,k-1)})\|^2}{n^2 q} + \mathbb{E}[\|\tilde{w}^{(t,K)} - w^*\|^2] + \sigma^2 d$$

$$\leq \frac{3K\eta^2\sum_{k=1}^{K}\big\{\sum_{i=1}^{n}\big(\beta^2\|w_i^{(t,k-1)} - \tilde{w}^{(t,k-1)}\|^2\big) + 2\beta n(F(\tilde{w}^{(t,k-1)}) - F(w^*)) + n\tau\big\}}{n^2 q}$$

$$+ \mathbb{E}[\|\tilde{w}^{(t,K)} - w^*\|^2] + \sigma^2 d$$

$$= O\big(\|\bar{w}^{(0)} - w^*\|^2 + tK\big(\eta\gamma\frac{\mathcal{B}}{c + \mathcal{B}} + (\eta^2 + \eta^3 K^2)\tau + \frac{K\tau\eta^2}{nq}\big) + t\sigma^2 d\big). \tag{93}$$

for sufficiently small $\eta = o(1/K)$ and $K = O(nq)$. Thus, with the above reasoning, we consider $t = T$ and $k = K$, and then we obtain a global upper bound.

# D  Utility of DP-LSGD in Strongly Convex Optimization

**Theorem D.1.** *For an arbitrary objective loss function $F(w) = \frac{1}{n} \cdot \sum_{i=1}^{n} f_i(w)$ where $f_i(w)$ is $\lambda$-strongly-convex and $\beta$-smooth, when $\eta < \min\{1/\beta, 2/(\beta + \lambda)\}$, Algorithm 1 with clipped local update (2) ensures that*

$$\mathbb{E}[\|\bar{w}^{(T)} - w^*\|^2] \leq \big(1 - (\eta\lambda)^2\big)^{TK}\|\bar{w}^{(0)} - w^*\|^2 + \frac{4(1 + \eta\lambda)^K \cdot \big(\frac{c^2}{nq} + \mathcal{B}^2 + \eta^2\tau K^2 + \sigma^2 d\big)}{((1 + \eta\lambda)^K - 1)(1 - (\eta\lambda)^2)^K}. \tag{94}$$

*Proof.* For simplicity, we use $G(w) = w - \eta\nabla F(w)$ to represent the output of gradient descent of function $F(w)$. Similarly, we use $G_i(w) = w - \eta\nabla f_i(w)$ to denote the gradient descent output of the $i$-th individual loss function $f_i(w)$.

**Lemma D.1** ([50]). *If $F(w)$ is convex and $\beta$-smooth, and $\eta \leq 2/\beta$, then the operation $G(w)$ is contractive, i.e.,*

$$\|G(w) - G(w')\| \leq \|w - w'\|,$$

*for arbitrary $w$ and $w'$. In addition, if $F(w)$ is $\lambda$-strongly convex and $\beta$-smooth, then if $\eta \leq 2(\beta + \lambda)$, then $G(w)$ is strictly contractive such that*

$$\|G(w) - G(w')\| \leq (1 - \frac{\eta\beta\lambda}{\beta + \lambda})\|w - w'\|.$$

In the $t$-th phase of Algorithm 1, conditional on the initialization $\bar{w}^{(t-1)}$, we first consider a virtual trajectory produced by applying full gradient descent on $F(w)$ with step size $\eta$ for $K$ iterations. We denote those iterates by $\tilde{w}^{(t,k)}$, for $k = 1, 2, \cdots, K$. Let $w^* = \arg\min_{w \in \mathcal{W}} F(w)$ be the global optimum, when $\eta < 1/\beta$,

$$\|\tilde{w}^{(t,k)}) - w^*\|^2 = \|\tilde{w}^{(t,k-1)} - w^* - \eta \nabla F(\tilde{w}^{(t,k-1)})\|^2 \tag{95}$$

$$\leq \|\tilde{w}^{(t,k-1)} - w^*\|^2 + \eta^2 \|\nabla F(\tilde{w}^{(t,k-1)})\|^2 - 2\eta(F(\tilde{w}^{(t,k-1)}) - F(w^*)) \tag{96}$$

$$\leq (1 - \eta\lambda)\|\tilde{w}^{(t,k-1)} - w^*\|^2 + (2\eta^2\beta - 2\eta)(F(\tilde{w}^{(t,k-1)}) - F(w^*)) \tag{97}$$

$$\leq (1 - \eta\lambda)\|\tilde{w}^{(t,k-1)} - w^*\|^2. \tag{98}$$

In (96), we use the property of strong convexity that

$$F(\tilde{w}^{(t,k-1)}) - F(w^*) \leq \langle \nabla F(\tilde{w}^{(t,k-1)}), \tilde{w}^{(t,k-1)} - w^* \rangle - \frac{\lambda}{2}\|\tilde{w}^{(t,k-1)} - w^*\|^2.$$

In (97), we use the smooth assumption that $\frac{1}{2\beta} \cdot \|\nabla F(\tilde{w}^{(t,k-1)})\|^2 \leq F(\tilde{w}^{(t,k-1)}) - F(w^*)$. Finally, in (98), as $\eta < 1/\beta$ and thus $2\eta(\eta\beta - 1) < 0$. Therefore,

$$\|\tilde{w}^{(t,K)} - w^*\|^2 \leq (1 - \eta\lambda)^K \|\bar{w}^{(t-1)} - w^*\|^2. \tag{99}$$

We will use $\gamma_1 = (1 - \eta\lambda)^K$ for simplicity.

Now, we consider to bound the deviation between $\tilde{w}^{(t,K)}$ and $\bar{w}^{(t)}$. In the following, we always assume $\eta < \min\{1/\beta, 2/(\beta + \lambda)\}$. It is noted that, based on the strict contraction property of $G$ and $G_i$, for any $u$ and $v$,

$$\|G_i(u) - G(v)\| = \|G_i(u) - G_i(v) + G_i(v) - G(v)\| \leq \|G_i(u) - G_i(v)\| + \|G_i(v) - G(v)\|$$

$$\leq (1 - \frac{\eta\beta\lambda}{\beta + \lambda})\|u - v\| + \eta\|\nabla f_i(v) - \nabla F(v)\|.$$

In the following, we use $\gamma_2 = (1 - \frac{\eta\beta\lambda}{\beta+\lambda})$ for simplicity. Similarly, for $\{G_1, G_2, \cdots, G_n\}$ on inputs $\{u_1, u_2, \cdots, u_n\}$, we have

$$\|\frac{\sum_{i=1}^n G_i(u_i)}{n} - G(v)\| \leq \gamma_2 \cdot \frac{\sum_{i=1}^n \|u_i - v\|}{n} + \|\frac{\sum_{i=1}^n G_i(v)}{n} - G(v)\|$$

$$= \gamma_2 \cdot \frac{\sum_{i=1}^n \|u_i - v\|}{n}. \tag{100}$$

At the $t$-th phase, from the initialization $\bar{w}^{(t-1)}$, $w_i^{(t,K)} = \underbrace{G_i \circ G_i \circ \cdots \circ G_i}_{k}(\bar{w}^{(t-1)})$. On the other hand, with the same start point $\bar{w}^{(t-1)}$, the virtual iterate $\tilde{w}^{(t,K)} = \underbrace{G \circ G \circ \cdots \circ G}_{k}(\bar{w}^{(t-1)})$.

Therefore, with a recursion reasoning,

$$\|\tilde{w}^{(t,K)} - \frac{\sum_{i=1}^n w_i^{(t,K)}}{n}\|$$

$$\leq \frac{\gamma_2 \cdot \sum_{i=1}^n \|w_i^{(t,K-1)} - \tilde{w}^{(t,K-1)}\|}{n}$$

$$\leq \frac{\gamma_2 \cdot \sum_{i=1}^n \left(\gamma_2 \|w_i^{(t,K-2)} - \tilde{w}^{(t,K-2)}\| + \eta\|\nabla f_i(\tilde{w}^{(t,K-1)}) - \nabla F(\tilde{w}^{(t,K-1)})\|\right)}{n} \tag{101}$$

$$\leq \|\bar{w}^{(t-1)} - \bar{w}^{(t-1)}\| + \frac{\eta \sum_{k=0}^{K-2} \gamma_2^{K-k} \sum_{i=1}^n \|\nabla f_i(\tilde{w}^{(t,k)}) - \nabla F(\tilde{w}^{(t,k)})\|}{n}$$

$$\leq \frac{\eta\sqrt{\tau}(1 - \gamma_2^K)}{1 - \gamma_2}.$$

Here, in (101), we apply Assumption 2.1 on the variance bound $\tau$, where the sampling noise of stochastic gradient satisfies $\|\sum_{i=1}^n (\nabla f_i(w) - \nabla F(w))\| \leq n\mathcal{B}$. Now, we further take the clipping

operation, i.i.d. sampling and DP noise into accountant. First, due to the clipping, stemmed from (101),

$$\|\frac{\sum_{i=1}^{n} \bar{w}^{(t-1)} + \mathcal{CP}(\Delta w_i^{(t)}, c)}{n} - \tilde{w}^{(t,K)}\| = \|\frac{\sum_{i=1}^{n} \bar{w}^{(t-1)} + \mathcal{CP}(w_i^{(t,K)} - \bar{w}^{(t-1)}, c)}{n} - \tilde{w}^{(t,K)}\|$$

$$\leq \|\frac{\sum_{i=1}^{n}(\bar{w}^{(t-1)} + \mathcal{CP}(w_i^{(t,K)} - \bar{w}^{(t-1)}, c) - w_i^{(t,K)})}{n}\| + \|\frac{\sum_{i=1}^{n} w_i^{(t,K)}}{n} - \tilde{w}^{(t,K)}\|)$$

$$\leq \mathcal{B} + \frac{\eta\sqrt{\tau}(1 - \gamma_2^K)}{1 - \gamma_2}. \tag{102}$$

In the following, we proceed to incorporate the sampling noise and DP noise into the deviation analysis. Let $\mu^{(t)} = \frac{\sum_{i=1}^{n} \mathcal{CP}(\Delta w_i^{(t)}, c)}{n}$ be the average of clipped local update at the $t$-th phase. Let $\mathbf{1}_i^{(t)}$ to be an indicator which equals 1 iff the $i$-th sample gets selected (independently with rate $q$). Then,

$$\mathbb{E}[\|\bar{w}^{(t)} - \tilde{w}^{(t,K)}\|] = \mathbb{E}[\|\bar{w}^{(t-1)} + \frac{\sum_{i=1} \mathbf{1}_i^{(t)} \cdot \mathcal{CP}(\Delta w_i^{(t)}, c)}{nq} + e^{(t)} - \tilde{w}^{(t,K)}\|] \tag{103}$$

$$\leq \mathbb{E}[\|\bar{w}^{(t-1)} + \frac{\sum_{i=1} \mathbf{1}_i^{(t)} \cdot \mathcal{CP}(\Delta w_i^{(t)}, c)}{nq} - \tilde{w}^{(t,K)}\|] + \sigma\sqrt{d} \tag{104}$$

$$= \mathbb{E}[\|\bar{w}^{(t-1)} + \frac{\sum_{i=1} \mathbf{1}_i^{(t)} \cdot \mathcal{CP}(\Delta w_i^{(t)}, c)}{nq} - \mu^{(t)} + \mu^{(t)} - \tilde{w}^{(t,K)}\|] + \sigma\sqrt{d} \tag{105}$$

$$\leq \mathbb{E}[\|\frac{\sum_{i=1}(\mathbf{1}_i^{(t)} - q) \cdot \mathcal{CP}(\Delta w_i^{(t)}, c)}{nq}\| + \|\bar{w}^{(t-1)} - \tilde{w}^{(t,K)} + \mu^{(t)}\|] + \sigma\sqrt{d} \tag{106}$$

$$\leq \sqrt{\frac{nc^2}{n^2 q}} + \mathcal{B} + \frac{\eta\sqrt{\tau}(1 - \gamma_2^K)}{1 - \gamma_2} + \sigma\sqrt{d}. \tag{107}$$

In (104), we use the fact that $Q^{(t)}$ is independent DP noise with zero mean and $\mathbb{E}[\|Q^{(t)}\|] = \sigma\sqrt{d}$. In (106), we use the triangle inequality. In (107), we use the convexity of $l_2$ norm function and it is noted that $(\mathbf{1}_i^{(t)} - q)$ for $i = 1, 2, \cdots, n$, are i.i.d. and of zero mean while $\|\mathcal{CP}(\Delta w_i^{(t)}, c)\| \leq c$.

So far, we have derived the expected deviation between $\bar{w}^{(t)}$ and $\tilde{w}^{(t,K)}$ at the end of the $t$-th phase conditional on $\bar{w}^{(t-1)}$. In the following, we will continue to incorporate such deviation to (99).

By applying the AM-GM inequality, $\|u - v\|^2 \leq (1 + z)\|u\|^2 + (1 + \frac{1}{z})\|v\|^2$ for any $z > 0$, on $\|\bar{w}^{(t)} - w^*\|^2 = \|(\tilde{w}^{(t,K)} - w^*) + (\bar{w}^{(t)} - \tilde{w}^{(t,K)})\|^2$, we have that

$$\mathbb{E}[\|\bar{w}^{(t)} - w^*\|^2] \leq (1 + z)\mathbb{E}[\|\tilde{w}^{(t,K)} - w^*\|^2] + (1 + \frac{1}{z})\|\bar{w}^{(t)} - \tilde{w}^{(t,K)}\|^2]$$

$$\leq (1 + z)\gamma_1 \mathbb{E}[\|\bar{w}^{(t-1)} - w^*\|^2] + (1 + \frac{1}{z})(\frac{c}{\sqrt{nq}} + \mathcal{B} + \frac{\eta\sqrt{\tau}(1 - \gamma_2^K)}{1 - \gamma_2} + \sigma\sqrt{d})^2$$

$$\leq (1 + z)\gamma_1 \mathbb{E}[\|\bar{w}^{(t-1)} - w^*\|^2] + 4(1 + \frac{1}{z})(\frac{c^2}{nq} + \mathcal{B}^2 + \frac{\eta^2\tau(1 - \gamma_2^K)^2}{(1 - \gamma_2)^2} + \sigma^2 d) \tag{108}$$

Based on (108) by recursion, we further obtain the following unconditional expectation

$$\mathbb{E}[\|\bar{w}^{(T)} - w^*\|^2] \leq ((1 + z)\gamma_1)^T \|\bar{w}^{(0)} - w^*\|^2 + \frac{4(1 + \frac{1}{z})}{1 - (1 + z)\gamma_1}(\frac{c^2}{nq} + \mathcal{B}^2 + \frac{\eta^2\tau^2(1 - \gamma_2^K)^2}{(1 - \gamma_2)^2} + \sigma^2 d)$$

$$\leq (1 - (\eta\lambda)^2)^{TK} \|\bar{w}^{(0)} - w^*\|^2 + \frac{4(1 + \eta\lambda)^K \cdot (\frac{c^2}{nq} + \mathcal{B}^2 + \eta^2\tau K^2 + \sigma^2 d)}{((1 + \eta\lambda)^K - 1)(1 - (\eta\lambda)^2)^K} \tag{109}$$

In (109), we select $z = (1 + \eta\lambda)^K - 1$, $\qquad\qquad\qquad\qquad\qquad\qquad\qquad\qquad\qquad\qquad\square$

# E Proof of Theorem 4.2: Utility of DP-LSGD in Non-Convex Optimization

To apply Theorem 3.2 on DP-LSGD, we may equivalently view the perturbation term $Q^{(t)}$ as formed by two parts. One is due to the local update clipping and the other is the DP noise added, denoted by $e^{(t)}$ in this proof. To be formal, $Q^{(t)}$ can be rewritten as follows,

$$
\begin{aligned}
Q^{(t)} &= \frac{\eta}{nq} \sum_{i \in S_t} \sum_{k=0}^{K-1} (1 - \frac{c}{\max\{\|\sum_{k=0}^{K-1} g_i^k\|, c\}}) g_i^k + e^{(t)} \\
&= \underbrace{\frac{\eta}{nq} \sum_{i=1}^{n} \sum_{k=0}^{K-1} 1_i^{(t)} (1 - \frac{c}{\max\{\|\sum_{k=0}^{K-1} g_i^k\|, c\}}) g_i^k}_{(A)} + e^{(t)}.
\end{aligned}
\tag{110}
$$

In (110), term (A) corresponds to the correction term due to the clipping, where equivalently the learning rate of the local update from each sample is scaled by a factor determined by the norm $\|\sum_{k=0}^{K-1} g_i^k\|$. $e^{(t)}$ is the independent DP noise added in the $t$-th phase. Therefore, conditional on $\bar{w}^{(t-1)}$, the expectation of $\|Q^{(t)}\|^2$ is in the following form,

$$
\begin{aligned}
\mathbb{E}[\|Q^{(t)}\|^2] &= \frac{\mathbb{E}[\|\sum_{i=1}^{n} \sum_{k=0}^{K-1} 1_i^{(t)} \eta(1 - \frac{c}{\max\{\|\sum_{k=0}^{K-1} g_i^k\|, c\}}) g_i^k\|^2]}{(nq)^2} + \sigma^2 d \\
&\leq \frac{\sum_{i=1}^{n} \mathbb{E}[\|\eta(1 - \frac{c}{\max\{\|\sum_{k=0}^{K-1} g_i^k\|, c\}}) \sum_{k=0}^{K-1} g_i^k\|^2]}{nq} + \sigma^2 d \\
&= \frac{\sum_{i=1}^{n} \mathbb{E}[(\Psi_i^{(t)})^2]}{nq} + \sigma^2 d = q\mathcal{B}^2 + \sigma^2 d.
\end{aligned}
\tag{111}
$$

Recall Definition 4.1, in (111), $\Psi_i^{(t)}$ is the incremental norm of the local update by $i$-th sample in the $t$-th phase, i.e., $\max\{\|\eta \sum_{k=0}^{K-1} g_i^k\| - c, 0\}$. Now, plugging the form of $\mathbb{E}[\|Q^{(t)}\|^2]$ in (111) back to Theorem 3.2, we obtain the utility bound claimed for DP-LSGD.

# F Additional Experiments and Experiment Setups

For all the experiments with respect to CIFAR10, we assume the training data set of 50,000 samples is private. Similarly, for SVHN, we assume the training data set of 73,257 samples is private. In Fig. 2 (a,b), we report the statistics of normalized incremental norm when we train ResNet 20 on SVHN. Very similar to our observation on CIFAR10, both the mean and the standard deviation of the normalized incremental norm in DP-LSGD is only about a half of those in DP-SGD, which suggest that DP-LSGD bears less influence from the clipping operator. As a consequence, in Fig. 2 (c), we can see DP-LSGD enjoys a faster convergence rate accompanying with a better utility-privacy tradeoff. Our code can be found in the following anonymous Github link: `https://anonymous.4open.science/r/DP-Local-SGD--262F/README.md`.

As for the hyper-parameter selection, in Table 1, for both the experiments on CIFAR10 and SVHN, the total number of phases $T$ is selected to be $1000, 1000, 1500, 1500, 2000$ and $2000$ for $\epsilon = 1.5, 2, 2.5, 3, 3.5$ and $4$, respectively. For DP-LSGD, $K$ is always fixed to be 10 and $\eta = 0.025$; while for DP-SGD, $K = 1, \eta = 1$.

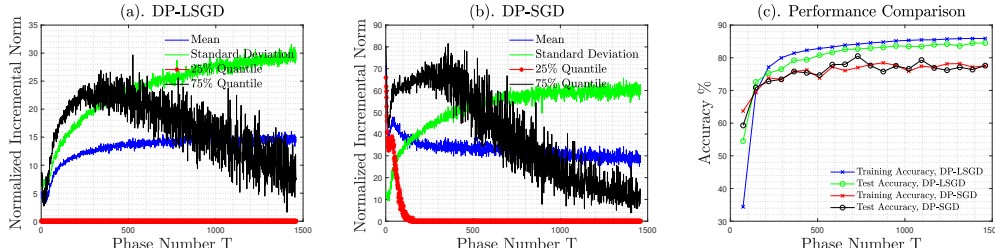

Figure 2: Training ResNet 20 on SVHN with DP-LSGD ($K = 10, \eta = 0.025, c = 1$) and DP-SGD ($K = 1, \eta = 1, c = 1$) under ($\epsilon = 2, \delta = 10^{-5}$)-DP, with expected batch size 1000.