# OpenReview forum: "Why Differentially Private Local SGD -- An Analysis of Synchronized-Only Biased Iterate"
_NeurIPS.cc/2023/Conference — Submitted to NeurIPS 2023_

### Official Review · Reviewer_7rJQ · 2023-07-05

**Soundness:** 3 good
**Presentation:** 3 good
**Contribution:** 3 good
**Rating:** 7
**Confidence:** 5

**Summary:**

This paper studies DP-LSGD and compares its performance with DP-SGD. The paper first provide the convergence result of FedAvg under the bounded variance assumption (Theorem 3.1 and 3.2) and provide the convergence analysis of DP-LSGD-GC under the bounded gradient assumption 4.1 and similarity assumption 4.2. for both convex and non-convex cases. The results imply that using multiple local updates, DP-LSGD converges faster to a neighborhood of the stationary point. Through numerical experiments, the paper demonstrates that DP-LSGD converges faster than DP-SGD with the same privacy budget and "communication" iterations.

**Strengths:**

Originality: this paper provides a novel analysis of DP-LSGD, which considers both clipping and DP-noise. The resulting convergence rate improves upon the existing FedGD algorithm.

Significance: This paper provides a DP algorithm (DP-LSGD) that outperforms DP-SGD with faster theoretical convergence to a neighborhood of stationary points.

Clarity: This paper provides a clear statement of the theorems, assumptions, and adequate numerical justification for the assumption used in the proof.


**Weaknesses:**

Assumption 4.1: This assumption assumes that the clipping error is bounded, which might be too strong. In Fig 1 (a), it seems like $\Phi$ is increasing as $t$ increases. Such an assumption simplifies the analysis of gradient clipping; Thus, it weakens the significance of the paper a bit.

Comparison with FedAvg: In general FedAvg considers local SGD updates, while the analyzed DP-LSGD algorithm considers local GD updates. Therefore, such a comparison is unfair. [R1] also provides the convergence rate of FedAvg, which matches the rate in this paper. Therefore, the convergence part without clipping is hard to be said to be an improvement.

It is unclear whether the numerical comparison is fair or not. It is hard to decide if the reported result for DP-SGD matches the SOTA results (e.g. in [R2]). The authors should report how the hyper-parameters are chosen and whether they are optimal for the algorithm.

The theoretical result suggested that $c = \Theta(\eta)$ and at the same time $B = O(c)$. However, it is unclear if these two results can be satisfied at the same time. The authors should also conduct numerical justification on different choices of $\eta$ and $c$ and report the corresponding $\Phi$.


[R1] Glasgow, M. R., Yuan, H., & Ma, T. (2022, May). Sharp bounds for federated averaging (local SGD) and continuous perspective. In International Conference on Artificial Intelligence and Statistics (pp. 9050-9090). PMLR.
[R2] De, S., Berrada, L., Hayes, J., Smith, S. L., & Balle, B. (2022). Unlocking high-accuracy differentially private image classification through scale. arXiv preprint arXiv:2204.13650.

**Questions:**

As listed above, please clarify
1) how $B$ changes with different $c$ and $\eta$ and
2) whether the reported performance of DP-SGD matches with SOTA results.
3) For non-convex cases, why the algorithm converges to saddle points? (I believe this should be a typo?)

**Limitations:**

The authors have addressed the empirical efficiency limitation of the proposed algorithm, which I believe is the largest limitation.

---

> ### Author Rebuttal · Authors · 2023-08-09
>
> Thanks for your comments which also inspire us to properly describe the strength of DP-LSGD.
>
> 1). On Assumption 4.1: We apologize for the confusion. Actually we do not assume a bounded clipping error; instead we only assume its second moment is bounded. We will stress this in a revision. The details of how we address this challenge and avoid the bounded gradient assumption compared to prior works can be found in Item 1) of our response to Reviewer SLyE.
>
> 2). On local stochastic gradient and comparison to prior works: We have partially explained this in the global response and response to Reviewer 624r, where our results can easily capture the local stochastic gradients case; and when we compare with existing works, we also compare to the case of full local gradient. In addition, thanks for pointing out the reference “Sharp Bounds for Federated Averaging and Continuous Perspective”, which provides nice lower bounds for federated learning. We will add comparisons to it in a revision. As a summary, we totally agree the improvement from the optimization side is not the key contribution or the main focus of this paper. All our results are presented to explain the better convergence rate and clipping bias from DP-LSGD and hopefully provide systematic instructions on future improvement of DP optimization.
>
> 3). Comparison with state-of-art [36]: Thanks for this very sharp question. On one hand, in the experiments reported, we have optimized the hyper-parameter selections, including the clipping threshold $c$ and the iteration/phase number $K$ and $T$ for all the cases of DP-SGD and DP-LSGD. We will elaborate on this in Appendix F of a revision. In addition, as for the comparison with [36], we need to mention that the success of [36] heavily relies on the scaling with a very large batchsize. In [36] for CIFAR10, besides many other advanced deep learning tricks, one dominant issue is the large subsampling rate with extensive ($16\times$) data self augmentation. As discussed in the conclusion section, [36] selects a batchsize over 15,000 in each iteration of DP-SGD to sharpen the signal-to-noise ratio in a subsampled Gaussian mechanism. In contrast, all our reported results are produced using a small batch size of 1000, partially because our machine memory can only support such a setup for ResNet20. However, we want to mention, when we implement [36] in the same setup of a small 1000 batch size, when $\epsilon=8$, [36] only produces 70.6\%, comparable to our case using DP-LSGD when $\epsilon=4$. In the same setup, when $\epsilon=2$, DP-LSGD achieves 64.0\% accuracy on CIFAR10, while [36] can only achieve 60.2\%. Thus, we believe the framework of DP-LSGD with connections to federated learning is a promising direction for future DP optimization research.
>
> 4). Dependence on $c$, $B$ and $\eta$. Thanks for this very insightful question. First, as explained in Item 3) of our response to Reviewer WAtK, $B$ in Theorems 4.1 and 4.2 is an arbitrary term, which can be dependent on $K$. In practice, when $K$ is large with a larger step size $\eta$, the norm of the local update also becomes larger, which basically carries more information, and we also need to correspondingly increase the clipping threshold $c$ to produce the optimal performance. Usually, $c$ is selected as the average of the local update norm. We have added additional experimental results in the attachment.
>
> For CIFAR10, Table 2 attached showcases the average $l_2$-norm of the local updates across different combinations of $K \in$ {1, 4 ,8, 12, 16, 20} and $\eta \in$ {0.01, 0.02, 0.03, 0.04, 0.05} over the initial $T=100$ phases. On one hand, for a given stepsize (within each column), a discernible trend emerges: the rate of increase in the $l_2$ norm of the local update decelerates as $K$ escalates. In other words, the ratio between the norm of the local update and the value of $K$ diminishes as the number of local gradient descent steps, $K$, grows. This observation lends credence to our assertion that the sampling noises originating from local gradients—despite their interdependence and evaluation on the same datapoint—tend to cancel out substantially. On the other hand, when focusing on a fixed value of $K$ (within each row), the norm of the local update maintains a linear proportionality with the step size $\eta$, which matches our intuition
>
> In Table 3, we further compare the incremental norm bound $B$ with the corresponding clipping bias bound across various selections of $K$ and $\eta$. We adopt a clipping threshold in a form $c = 25 \cdot K \eta$, where the value of $c$ is scaled with $K$ and $\eta$. The choice of the constant $25$ is informed by our empirical tests, which demonstrate that this particular clipping parameter selection generally yields the optimal performance in the context of the CIFAR10 dataset. Within Table 3, we present the ratio $B/c$, which captures the bound on the clipping bias as outlined in Theorem 4.1. Evidently, larger values of $K$ tend to yield more concentrated local updates and enhanced clipping efficiency. Based on our observations, in real-world applications of DP-LSGD, a suitable choice is often $K=10$ along with $\eta=0.025$.
>
> As revealed by Table 2, the $l_2$ norm of local updates exhibits a linear relationship with both $K$ and $\eta$. In cases where $K$ and $\eta$ are excessively large, it becomes necessary to correspondingly increase the value of $c$ in order to mitigate clipping bias. However, this simultaneously introduces disruptive high-dimensional DP noise. On the other hand, even for the non-private scenario without noise perturbation, LSGD typically demands a comparatively smaller learning rate to prevent local updates from excessively diverging. For substantially large $\eta$, we observe that the variance of local updates will significantly increase and the convergence becomes less stable.
>
> Please feel free to let us know if you have any other questions! Thanks again.

---

> > ### Comment · Reviewer_7rJQ · 2023-08-11
> >
> > 1. Bounding second-order moment is equivalent to bounding both its magnitude and variance of the clipping error. $E(x^2) = (Ex)^2 + Var(x) \leq B \rightarrow (Ex)^2\leq B and Var(x)\leq B$.
> >
> > Overall, I think my questions have been addressed. This is a delicate algorithm with promising theoretical result, yet hard to be used for practical large-scale DNN training.

---

> > > ### Author Response · Authors · 2023-08-11
> > >
> > > Thanks so much for your prompt reply! Hopefully we can optimize our code with a much smaller memory requirement in the near future. We  appreciate your support for our new research direction to apply federated learning methods to systematically improve clipped DP learning!

---

### Official Review · Reviewer_624r · 2023-07-06

**Soundness:** 2 fair
**Presentation:** 3 good
**Contribution:** 2 fair
**Rating:** 4
**Confidence:** 4

**Summary:**

This paper proposed a unified analysis of the convergence of (DP)-Local SGD, which covers (DP) parallel SGD as a special case with K=1, for both convex and non-convex optimization. Under this unified analysis, one can identify error effects due to non-iid objectives, clipping, and DP noises and the convergence rate to the error neighborhood.



**Strengths:**

(1) The introduction section provides a brief yet insightful summary of related subjects including "relation between local SGD and parallel SGD",  "sensitivity and privacy analysis methodology of DP", "effect and convergence limitation of clipping" etc.

(2) The unified framework that covers both local SGD and parallel SGD is attractive and makes sense at a high level.

**Weaknesses:**

(1) The problem setting of this paper assumes each sampled worker is running gradient descent instead of stochastic gradient descent as the full gradient is available in equation (1) though later an additive stochastic noise is added after the aggregation of local updates in equation (2) and (3).   This problem setting is much simpler than the standard setting of local SGD as less divergence across involved local workers is introduced.  The comparison between the analysis from this paper, e.g., Thm 3.2, and the state-of-the-art is no longer fair.  (All the state-of-the-art compared in this paper considered stochastic optimization.  And it is well known that GD for deterministic non-convex has O(1/T) convergence while SGD for stochastic non-convex only has O(1/\sqrt{T}) convergence.)


(2) The assessment in line 313 that DP-LSGD converges faster with O(1/T) than DP-SGD corresponding to K=1 with O(1/\sqrt{T}) is unfair as the convergence rate from Thm 4.2 requires K=\Theta(T) such that the overall computation/iteration is TK = O(T^2) to attain an error decay like 1/T.  This is effectively the same O(1/\sqrt{S}) convergence where S is the number of computation steps/iterations.


(3) The new assumption in Assumption 4.1 that involves \Phi in Definition 4.1 does not seem much different from a bounded 2nd-order moment assumption as \Phi_i measures how much the norm of update is larger than the clipping threshold.  (Given that a bounded 2nd-order movement further implies a first-order moment by the inequality E[||X||] \leq \sqrt{E[||X||^2]}, I don't see how this new assumption can relax the widely used bounded gradient assumption in the literature.) Could the author discuss whether the new assumption strengthens or relaxes the standard assumptions?




**Questions:**

See my questions in the "Weaknesses" section.

**Limitations:**

I do not see any potential negative societal impact of this paper.

---

> ### Author Rebuttal · Authors · 2023-08-09
>
> Thanks for your comments.
>
> 1). Stochastic local gradient and the convergence rate of federated learning: Thanks for this very sharp question. We first answer the question about the generalization of our results with local stochastic gradients. We totally agree, in a complete picture of DP-LSGD, there exist three types of noise: (a) noise from sampling over nodes (in each phase, different nodes will be randomly selected to participate in the computation); (b) noise from sampling over local samples from each selected node (local stochastic gradient); (c) DP noise for perturbation. In this paper, to provide a clear comparison between DP-LSGD and DP-SGD in the centralized DP model, which is equivalent to a federated learning where each node has a single datapoint, we mainly take a) and c) into account since the local stochastic gradient b) is reduced to the full local gradient when there is only one local datapoint. However, as explained in the global response, our results indeed study a more generic scenario where DP-LSGD is perturbed by possibly biased noise and we always take a generic noise $Q$ in our analysis. If one applies additional local stochastic gradients accompanied by an independent zero-mean sampling noise of variance bounded by $\sigma^2_s$, then as pointed out by Reviewer WAtk, an additional term $O(\frac{\eta K\sigma^2_s}{n})$ will appear in Equation (4) in the paper.  $O(\frac{\eta K\sigma^2_s}{n})$ captures the effect of accumulated sampling error from $K$ local stochastic gradients, and theoretically can be merged with the DP noise.
>
> Second, we want to discuss more about the convergence rate of federated learning and its connection to DP-(L)SGD. We totally agree that in the centralized case, $O(1/T)$ convergence rate of full GD and the $O(1/\sqrt{T})$ rate of SGD have been well understood. However, things become much more complicated in the federated learning case, especially with heterogeneous data. Even in the non-private realm, there are many open questions about the optimal convergence rate when combining both (a) and (b) in Item 1) above, with sampling over both users and local samples. However, with careful error-feedback or variance reduction, LSGD with only (a), where we allow subsampling over users but a selected user still applies full local gradient descent, can achieve $O(1/T)$ convergence, for example Scaffold [30] and FedLin [43]. This suggests, theoretically without noise and clipping, LSGD using full local gradient can outperform standard SGD $O(1/\sqrt{T})$ rate with proper assumptions. In this paper, still in the same setup for clipped DP-LSGD with (a,c), we show it can converge to a neighborhood of minimum at rate $O(1/T)$.
>
> A more important open question is that when the DP-noise is sufficiently small, can clipped DP-LSGD converge at a rate of $O(1/T)$ to the optimum “without clipping bias”? That is why we argue, in the conclusion section, to connect both the research in DP-SGD and federated learning and study whether the variance reduction method used for handling data heterogeneity can cancel out the clipping bias. Though we have tried to incorporate those methods in, such as Scaffold [30] and FedLin [43], into clipped LSGD, unfortunately, we find that current methods cannot be trivially modified to produce bounded sensitivity and thus enable efficient utility-privacy tradeoffs. But we believe this will be a promising direction to systematically improve DP-SGD for deep learning.
>
> 2). $O(1/T)$ convergence rate: Thanks for this very sharp question. We apologize for the confusion and as pointed out by Reviewer 7rJQ, the correct statement should be: DP-LSGD converges faster at the same privacy budget or “communication” iterations. As we explained in the introduction, DP optimization under the current analysis "white-box" framework shares many common questions/concerns with federated learning. The DP noise (computed using DP composition) added to each local update is determined by the total number of phases $T$, and thus for a smaller noise, we want a faster convergence rate in terms of $T$, which is equivalent to the communication overhead concern in federated learning, where we want faster convergence in “less communication” rounds $T$. We totally agree with your point that our results do not improve the entire iterations required, where DP-LSGD and DP-SGD still theoretically require the same order of computation complexity. But the local iterations in DP-LSGD will not contribute to the noise bound. We will properly modify this statement in a revision.
>
> 3). Technical and theoretical improvement: Due to the length limits, the technical challenges behind only assuming second-moment bounded local updates can be found in the global response and Item 1) of our response to Review SLyE. Moreover, we believe our most important contribution is not proving the convergence using the weaker second moment assumption, but the usable and explainable theory. We are not against the analysis of DP-(L)SGD using bounded gradients. Our concerns are that after assuming Lipschitz continuity, we may not properly characterize the difference between the clipping bias produced by DP-LSGD and DP-SGD: either no clipping happens when $c$ is larger than the Lipschitz constant, or we will not be able to explain what we can learn from the nonintuitive convergence rate/bias depending on the unknown or even nonexistent Lipschitz constant to further improve DP optimization.
>
> Please feel free to let us know if you have any other questions. Thanks again.

---

> ### Author Response · Authors · 2023-08-17
> **Follow-ups to Reviewer 624r**
>
> Dear Reviewer 624r
>
> We want to thank you again for your comprehensive comments. As we approach the final days of the discussion phase, we are eager to ascertain whether we have effectively addressed your concerns and whether any further inquiries remain.
>
> In the rebuttal, we have shown that our framework is even more generic which characterizes the effect of biased perturbation. Notably, we highlight that the scenario involving local stochastic gradients with independent zero-mean sampling noise is seamlessly encompassed as a special case.  We also explain our motivation without including the stochastic gradient, since we try to provide a clear comparison between DP-LSGD and DP-SGD in centralized DP scenario where equivalently each client only has one datapoint.
>
> Moreover, we have provided enhanced clarity regarding the benefits inherent to clipped DP-LSGD, which converges faster at a lower privacy budget compared to DP-SGD We hope that this clarification will also offer a more intuitive understanding of our pursuit in unifying the analysis framework. DP-SGD and federated learning share similar underlying concerns expressed through different terminologies—composite privacy leakage in DP and communication overhead in federated learning. This perspective illuminates the potential for cross-pollination of distributed optimization concepts to systematically enhance DP learning.
>
> Our efforts further extend to the elucidation of the motivations underpinning our introduced assumptions. We do not arbitrarily propose these assumptions and our goal is not simply to improve prior works with these weaker assumptions, either. We develop theory to explain the practice, where we have established a foundation for comprehending the empirical convergence phenomena of clipped SGD. We show DP-LSGD produces more concentrated local updates, consequently resulting in heightened clipping efficiency. As you pointed out, technically our assumptions merely require the bounded second moment for local updates.
>
> Finally, if your concerns are all properly addressed, we really hope that the reviewer can positively re-evaluate our work to support this research direction. We appreciate your inputs and we thank you for your time spent reviewing.

---

### Official Review · Reviewer_WAtK · 2023-07-11

**Soundness:** 3 good
**Presentation:** 4 excellent
**Contribution:** 3 good
**Rating:** 5
**Confidence:** 5

**Summary:**

This paper propose a differentially-private local stochastic gradient descent both for centralized and distributed settings. The authors argue that the proposed method has less number of clipping and in turn produce less clipping bias compared to its counterpart DP-SGD which do not involve local steps. They also show that DP-LSGD converges sublinearly to a ball of the optimum, which is claimed to be faster than that of DP-SGD, and exhibit a better utility-privacy tradeoff.

**Strengths:**

- The authors characterize the convergence performance by regarding clipping noises as biased noises and assuming that the incremental norm of local update be bounded, which is otherwise not easy to deal with.
- The authors prove that the proposed DP-LSGD converges faster than that DP-SGD and empirically show that it also has a better utility-privacy trade-off.
- The paper is technically sound and well organized.


**Weaknesses:**

- Assumption 4.1 and 4.2 seem restrictive to the reviewer. In particular, Assumption 4.1 seems not practical in the sense that one can not ensure the boundedess of the incremental norm of $\nabla w$ without knowing in advance the basic convergence of the algorithm (note that the algorithm may diverge, making $\nabla w$ unbounded); Assumption 4.2 is assumed to be hold for any value of w instead of the optimum w*, which is more common to be adopted.

- Theorem 4.1 and 4.2: the authors claim that DP-LSGD enjoys faster convergence to a neighborhood of the global optimum/ saddle point than DP-SGD; however, local sample-level differential privacy is guaranteed for DP-SGD, but not for DP-LSGD. For a fair comparison, each client for DP-LSGD should clip the calculated gradient and add the DP-noise at each SGD step in local update to satisfy the local sample-level differential privacy, which will inevitably degrade the convergence performance of the algorithm. In that case, what are the advantages of DP-LSGD compared to DP-SGD and does DP-LSGD still produce less clipping bias than DP-SGD?

- The selection of many parameters such as B, \eta and c lack of intuition; also, the experiment does not corroborate the theoretical result very well; for instance, the clipping bias captured by $\mathcal{B}$ is independent of $K$ (c.f., Theorem 4.1) and thus can not reduced with increasing value of $K$, which is inconsistent with the claim that DP-LSGD Produces Less Bias.


**Questions:**

- It is not surprising that the authors can avoid the assumption of bounded gradient with the introduction of bounded variance of stochastic gradient, which is well known in the existing literature; what are the new technical novelty in the convergence analysis?

- How one can determine the clipping threshold c and $\mathcal{B}$ such that $\mathcal{B}$ is at the same order as c?

- In Theorem 3.2, the authors claim that their iteration complexity to reach an error is tighter than the state-of-the-art results in [a] when there is no perturbation. This comparison seems unfair in that this paper only considers gradient variance among clients (Assumption 2.1) instead of the gradient sampling variance among datapoints, while both of them are considered in [a]. Note that, in this case, $\eta^2$ in the second term of (4) will become $\eta$.

[a] Sai Praneeth Karimireddy, Satyen Kale, Mehryar Mohri, Sashank Reddi, Sebastian Stich, and Ananda Theertha Suresh. Scaffold: Stochastic controlled averaging for federated learning. In International Conference on Machine Learning, pages 5132–5143. PMLR, 2020.


**Limitations:**

please refer to weaknesses and questions.

---

> ### Author Rebuttal · Authors · 2023-08-09
>
> Thanks for your comments.
>
> 1). Regarding Assumptions 4.1 and 4.2: With regards to Assumption 4.1, we apologize for the confusion caused. Actually, we did not assume the incremental norm is globally bounded but only a bounded second moment. It is worth noting that Assumption 4.1 can be seen as being equivalent to the condition where the second moment of the local update is confined. To illustrate, let us consider the scenario where we set $c=1$, and the expected value of the incremental norm—essentially the norm of the local update extending beyond the unit $l_2$-ball, is bounded by $2$. As a result, the expected value of the $l_2$ norm of the local update is bounded by $3$. The rationale behind our selection of the incremental norm for elaboration stems from it is better to intuitively capture the heightened concentration of local updates produced by DP-LSGD.
>
> Second, for Assumption 4.2, we totally agree this is a bit artificial but we only use this for DP-(L)SGD in convex optimization. Since we do not assume bounded gradient or Lipshitz continuity, to handle generic biased perturbation $\Delta$, we have to assume certain similarity between functions, otherwise we cannot bound the utility loss $f_i(x)-f_j(x+\Delta)$ with only a smooth assumption. Since we are the first work to study such biased clipping error without a bounded gradient assumption, we did not find other replaceable assumptions from prior works. But we will definitely consider more practical assumptions in our future work.
>
> 2). Comparison between DP-SGD and DP-LSGD in different models: As partially explained in the global response, when we present our results, we consider a very generic perturbation term $Q^{(t)}$ across the iterations in Equations (2) and (3) in the paper, which can capture many kinds of noise: such as clipping bias, sampling noise, compression/quantification error and DP noise. Thus, for different privacy models, the only difference is that the injected noise will be different. For example, if each local update is clipped to 1, for a total of $T$ phases/communications/releases, and for a centralized dataset with $n$ samples, DP-(L)SGD adds $O(\frac{q \sqrt{T}}{nq})$ noise to the aggregate updates given a sampling rate $q$; for sample-wise local DP, a node with $m$ local samples will add a noise $O(\frac{q_0 \sqrt{T}}{mq_0})$ to its local update produced by either DP-SGD or DP-LSGD, where $q_0$ is the local sampling rate over the $m$ local samples; for strict local DP where each node has a single datapoint and does not trust anyone else, it needs to add a noise $O(\sqrt{T})$ to the released local update, still either by DP-SGD or DP-LSGD. Thus, our results can capture the utility-privacy tradeoff in any privacy model by just plugging in different noise scales $\sigma$. More importantly, in the same setup with the same clipping threshold $c$, the noise injected for DP-SGD is identical to that for DP-LSGD. The only difference is whether one adopts DP-SGD to clip and expose the local gradient or applies DP-LSGD to clip and expose the local update formed by $K$ local gradients. Hence, back to your question, though in this paper we focus on centralized DP, DP-LSGD still outperforms DP-SGD in other privacy models in the same setup. We will add comments on this in a revision.
>
> 3). Relationship between $B$ and $K$: We are thankful for this sharp question. We need to mention that the $B$ in Theorems 4.1 and 4.2 is an arbitrary term, which can be a generic function of $K$ and is not necessarily some constant. We apologize for this confusion and will change the notation from $B$ to $B(K)$ for clarity.
>
> 4). New techniques to study DP-(L)SGD with only second-moment bounded gradient/local update: Thanks for this insightful comment. We totally agree that for non-private LSGD, the convergence rate with second-moment bounded gradients has been extensively studied. However, it remains challenging to study it in DP-(L)SGD with clipping or generally biased perturbation, and we are the first to address it. Due to the length limits, please refer to our responses to Reviewer SLyE.  Also as partially explained in the global response, our goal is not to simply improve existing works with weaker assumptions, but to develop meaningful theory using simulatable or explainable quantities, such as the incremental norm, to instruct systematic improvement.
>
> 5). How to select $c$ such that it is of the same order as $B(K)$. This is a very insightful question. The reason we consider such a selection or situation is twofold. First, it is based on our empirical observations on the optimal hyper-parameters in practice. We find that either for DP-SGD and DP-LSGD, the optimal selection of clipping threshold $c$ is usually close to the average/median of $l_2$-norm of local updates. A similar observation has also been made in prior adaptive clipping works, such as “Differentially Private Learning with Adaptive Clipping”. Second, we assume it is also mainly to simplify Equation (5) in the paper to provide more intuition about the asymptotic utility-privacy tradeoff. The clipping bias can be captured by $\frac{B(K)}{c}$ and theoretically $B(K)$ is not necessarily in the same order of $c$. Thus, given a proper selection of $c$, the more concentrated local updates will produce a smaller ratio $\frac{B(K)}{c}$, leading to a smaller clipping bias. More details can be found in our attachment and our item 4) response to Reviewer 7rJQ.
>
> 6). Stochastic local gradient: Thanks for this sharp question. We have partially explained this in the global response where we still provide fair comparison with prior works in the same full-batch gradient setup. Due to the length limit, further details can be found in our item 1) response to Reviewer 624r.
>
> Please feel free to let us know if you have any other questions. Thanks again.

---

> > ### Comment · Reviewer_WAtK · 2023-08-21
> >
> > Thank you for the detailed response which has addressed most of the reviewer's concerns. The reviewer's remaining concerns are still on  the restrictiveness of Assumptions 4.1 and the proper selection of certain important parameters, and would thus maintain the current score.

---

> > > ### Author Response · Authors · 2023-08-21
> > > **Additional comments on Assumption 4.1 and parameter selection**
> > >
> > > Thank you very much for your response, and we are delighted that we have successfully addressed most of your concerns.
> > >
> > > 1. Regarding Assumption 4.1 (second-moment bounded incremental norm), we explained in our rebuttal that this assumption is technically equivalent to having a local update or gradient with bounded second moment. This is a well-established concept in non-private optimization research. But we are totally agree our work could be possibly further improved and please do not hesitate to share your thoughts or any further suggestions on how we could potentially relax this assumption to enhance our results. On another note, we hope that we have clarified our motivation behind considering the incremental norm. Our intention is not solely to improve existing private optimization work by utilizing this weaker second-moment assumption, though it presents several technical challenges as we explained. More importantly, we aim to develop valuable theory that can explain and guide the field of DP learning. The incremental norm provides a clear and intuitive way to understand and control clipping errors.
> > >
> > > 2. Regarding parameter selection, we fully agree with your perspective. While we have presented numerous asymptotic analyses regarding hyper-parameter selections and the achieved sharper convergence rates associated with privacy-utility tradeoffs, we acknowledge that practical deep learning often requires fine-tuning. In practice, constants do matter, even though DP-LSGD inherently offers the potential for more concentrated local updates and improved clipping efficiency. This is precisely why we emphasize our released code, which stands as the first PyTorch platform to implement DP-LSGD for practical deep learning tasks with competitive running times. This allows us to fine tune the optimal parameter and produce state-of-the-art performance. Additionally, we have included extra experiments in our attached document to illustrate what optimal hyper-parameter selections will look like in practice. For instance, we highlight that the clipping threshold $c$ should be proportional to the step size $\eta$ and the number of local iterations $K$.
> > >
> > > In summary, we would like to express our sincere gratitude once again for your invaluable feedback. We truly appreciate your support for this innovative research direction by leveraging federated learning techniques to systematically enhance clipped DP learning. If you have any further questions or require additional information, please do not hesitate to reach out to us. Your input is highly valued.

---

### Official Review · Reviewer_SLyE · 2023-07-21

**Soundness:** 2 fair
**Presentation:** 1 poor
**Contribution:** 2 fair
**Rating:** 3
**Confidence:** 2

**Summary:**

The paper focuses on Differentially-Private Local SGD (DP-LSGD), and studies its advantages over the foundational technique of DP-SGD. In particular, the authors show why DP-LSGD provides higher clipping efficiency and less clipping bias compared to DP-SGD. The authors start by showing a convergence analysis on the released iterates of LSGD under perturbations and a bounded variance assumption on the stochastic gradients. Next, they generalize the results to DP-LSGD, and show that DP-LSGD has a faster convergence rate near an optimum point compared to DP-SGD. Lastly, they show that DP-LSGD behaves as an efficient variance reduction of local update, and enables more efficient clipping compared to DP-SGD.


**Strengths:**

1. The authors focus on the important problem of improving the privacy-utility trade-offs for DP Learning.


**Weaknesses:**

1. The paper contains many theoretical results (Sections 3-5), and I have not been able to verify the correctness of any of the proofs in the Appendix, but after reading the paper it is not even clear to me whether the proofs use any techniques/ideas that are novel (and might be of independent interest), or use methods from prior works to obtain novel results for (DP-)LSGD.
2. The empirical evaluation is very limited, focusing only on image-classification settings (CIFAR10 and SVHN datasets). Given that the focus of the paper is on (DP-)LSGD which is a building block of (DP) Federated Learning, it might be useful to have experiments on FL benchmark datasets, e.g., StackOverflow, EMNIST, etc.?


**Questions:**

Listed in the weaknesses section.

**Limitations:**

Yes.

---

> ### Author Rebuttal · Authors · 2023-08-09
>
> Thanks for your comments.
>
> 1). First, we apologize for the confusion caused about the main contributions and technical novelty of our paper. As we have partially explained in the global response, the key theoretical contributions are mainly twofold. On one hand, technically, to our knowledge, this is the first work which presents a unified convergence analysis with clipping bias description of DP-(L)SGD without assuming bounded gradient (local update) for both convex and non-convex optimization. On the other hand, as an independent contribution beyond privacy-preserving learning, for general convex optimization, to our knowledge, we also provide the first last-iterate convergence analysis without assuming bounded gradient. Due to the page limits, we did not thoroughly compare our techniques with prior works, which is also partially because we use very different methods compared to, say [16, 32], which must count on the assumption of bounded gradient. Hereafter, we briefly outline how we mitigate the need for a bounded gradient assumption in characterizing the clipping bias.
>
> We employ distinct strategies for convex, strongly convex, and non-convex scenarios. The non-convex instance is comparatively straightforward, wherein we present a generic analysis applicable to arbitrary perturbations $Q$ with bounded second-order moment, even if biased. A pivotal step is Equation (54) in Appendix B. By relying on the smoothness assumption, we demonstrate that, in expectation, a sufficiently small step size $\eta$ restricts both the local update drift stemming from data heterogeneity and the cumulative biased perturbation $Q$ across iterations. Importantly, we show the rate to the local minimum neighborhood is $O(1/T)$. The convex case poses a more formidable challenge. To counteract the impact of clipping bias, since local update clipping essentially corresponds to a projection operation, expressible equivalently via a scaled step size, we approach it as if each node (user) employs an individual step size for local updates in LSGD with clipping.  However, without assuming global bounds on gradients, we do not have a global lower bound for each step size. Nevertheless, by invoking Janson's inequality alongside the convexity of the utility function, we demonstrate that, once the second moment (or incremental norm) is bounded, clipping LSGD propels advancement toward the global optimum vicinity in an expectation sense. In scenarios of strong convexity, we harness an important property of the gradient descent operator, which is (strictly) contractive in (strongly) convex optimization. This allows us to present more compelling results, ensuring that the divergence introduced by clipping-induced bias will remain bounded. A more comprehensive explanation of these principles will be furnished in a revision.
>
> Concurrently, we regard the more important contribution of our paper as the implications derived from our principal theorems. In addition to demonstrating that DP-LSGD can achieve faster convergence within the same privacy budget in comparison to DP-SGD, we also expound upon the concept of clipping bias. The heightened concentration of local updates and the diminished clipping bias inherent in LSGD pave the way to systematically enhance DP learning. This involves connecting existing techniques from distributed optimization.
>
> 2). As for the experiments, as requested, we have added new results on EMNIST, which can be found in the Table 1 of the attachment, where we further test and compare DP-SGD and DP-LSGD on training ResNet20 for EMNIST dataset. EMNIST is an extension of MNIST to the more complicated handwritten letters, and still we assume the training set of totally 125,000 samples as the private data. We consider 6 scenarios of various security parameters where $\epsilon=$ {1.5, 2, 2.5, 3, 3.5, 4} with a fixed $\delta= 10^{-5}$. We optimize the hyperparameter selections of DP-SGD as follows. We search for the optimal step size $\eta \in $ {0.25,0.5,1,1.25,1.5}, the clipping threshold $c \in$ {0.5, 1, 1.5, 2, 2.5, 3}, and the number of iterations(communications/releases) $T \in$ {500, 1000, 1500, 2000} to produce the best test accuracy. Finally, we determine the optimal option as $\eta=0.5$ and $c=1$. Correspondingly, the $T$ for the 6 different $\epsilon=$ {1.5, 2, 2.5, 3, 3.5, 4} is selected as {1000, 1000, 1500, 1500, 1500, 2000}, respectively. As for DP-LSGD, we select $K=10$, $\eta = 0.025$, $c=2$, with the same selection of $T$. We record their performances in Table 1, where in each case we conduct 5 independent trials and report the median test accuracy. As a benchmark, in the non-private scenario without noise, i.e., $\epsilon=\infty$, we achieve a 94.5% test accuracy in 2000 iterations. From Table 1, we can see DP-LSDG still has obvious advantage over DP-SGD in the same setup, even for this relatively simpler learning task.
>
> Please feel free to let us know if you have any other questions. Thanks again.

---

> > ### Comment · Reviewer_SLyE · 2023-08-15
> > **Acknowledging the rebuttal**
> >
> > I have read the authors' rebuttal, and thank them for responding to the raised concerns. The 2nd point addressed by the authors by conducting experiments on EMNIST is helpful, and it would be useful to see it in a future iteration of the paper. Since I have not been able to verify correctness of the proofs, and the implications derived from principal theorems are an important contribution of the paper, as a result I am decreasing the confidence of my review.

---

> > > ### Author Response · Authors · 2023-08-15
> > >
> > > Thank you sincerely for your response and for spending time reading our rebuttals. We are committed to incorporating these supplementary experimental findings into our revised work. Regarding the theoretical part, please do not hesitate to inform us of any ways in which we can facilitate a deeper comprehension of our proposed methodologies for characterizing the clipping bias and the last-iterate convergence. Notably, our approach hinges on the weak assumption of second-moment-bounded local updates.
> > >
> > > In this context, we aim to present you with more insights into how DP-LSGD effectively mitigates sampling noise and produces more concentrated local updates, which finally leads to less clipping error. If your schedule permits, please also take a look at Table 2 and 3 enclosed within the accompanying document as well.
> > >
> > > Table 2 presents the mean $l_2$-norm of local updates across varying combinations of local steps ($K$) and step sizes ($\eta$) throughout the initial $T=100$ phases of training ResNet 20 on Cifar10 using DP-(L)SGD. One important and intuitive pattern emerged is that: as the number of local gradient descent steps ($K$) increases, the rate of escalation in the $l_2$ norm of local updates decelerates.  Said another way, the ratio between the local update norm and the value of $K$ diminishes as $K$ grows. This finding lends support to our point that the inherent sampling noises in local updates, even if they are dependent and evaluated on the same data point, exhibit a propensity to largely nullify each other. When we focus on a fixed $K$ value (within each row), it becomes evident that the local update norm maintains a linear correlation with the step size ($\eta$), aligning with our expectations.
> > >
> > > Turning to Table 3, we delve into a comprehensive comparison between the incremental norm bound ($B$) and the corresponding clipping bias bound across various choices of $K$ and $\eta$. Our approach involves a clipping threshold selected as $c = 25 \cdot K \eta$, wherein the value of $c$ scales in relation to $K$ and $\eta$. The rationale behind the selection of the constant $25$ is grounded in our empirical evaluations, which indicate that this particular clipping parameter choice consistently yields optimal results on the CIFAR10 dataset. Within Table 3, we present the $B/c$ ratio, which captures the clipping bias bound described in Theorem 4.1. Evidently, larger values of $K$ tend to yield more concentrated local updates and bolster the efficiency of clipping.
> > >
> > > Lastly, if you do not have additional concerns, we sincerely hope that you could consider re-evaluating our work positively and perhaps even contemplate an adjustment to your assessment score. Thank you again.

---

### Official Review · Reviewer_y8pk · 2023-07-26

**Soundness:** 3 good
**Presentation:** 3 good
**Contribution:** 3 good
**Rating:** 7
**Confidence:** 3

**Summary:**

This submission studies the Differentially-Private Local Stochastic Gradient Descent (DP-LSGD), and shows that  DP-LSGD with multiple local iterations can produce more concentrated local updates and   less clipping bias compared to DP-SGD, assuming that the stochastic gradient is of bounded variance.  The main contribution of this submission is to show that DP-LSGD has a faster convergence rate  compared to DP-SGDThe authors also add the experiments to  show that
DP-LSGD produces a better  utility-privacy tradeoff  than DP-SGD.

**Strengths:**

1. This submission develops the connections between the clipping bias and the second moment of local updates, which is something new in the research of differentially private optimization.
2. This submission shows that DP-LSGD can converge faster compared to regular DP-SGD.
3. The experimental results ( the comparison between DP-SGD and DP-LSGD) in this submission look convincing, and this paper is well-written.


**Weaknesses:**

1.The implementation inefficiency (local update in parallel at a cost of large memory) is a minor issue here.

**Questions:**

N/A

---

> ### Author Rebuttal · Authors · 2023-08-09
>
> Thanks for your positive assessment for our work. As you mentioned, we believe a more efficient implementation of DP-LSGD would be a promising direction for further work on DP optimization/learning, especially from a system engineering perspective. We present the first step, though at the cost of relatively large memory. However, after careful optimization, our released code has competitive running time compared to Opacus, the well-developed DP-SGD simulator. We will think about further improvement. Please feel free to let us know if you have any other questions. Thanks again.

---

> > ### Comment · Reviewer_y8pk · 2023-08-18
> > **Acknowledging the rebuttal**
> >
> > I have read the authors' rebuttal. Overall, I think my questions have been addressed.

---

> > > ### Author Response · Authors · 2023-08-18
> > >
> > > Thank you so much for your reply and we are glad that we have addressed all your concerns. We appreciate your support for our new research direction to connect federated learning techniques and DP learning for systematical improvement.

---

### Official Review · Reviewer_1PJn · 2023-07-27

**Soundness:** 3 good
**Presentation:** 4 excellent
**Contribution:** 3 good
**Rating:** 5
**Confidence:** 1

**Summary:**

The authors provide a unified analysis of the clipping bias and the utility loss in privacy-preserving gradient methods for centralized and distributed setups. The conclusion shows that LSGD behaves as an efficient variance reduction of local update, where multiple local GDs with a small learning rate cancel out substantial sampling noise and enable more efficient clipping compared to DP-SGD.

**Strengths:**

1. The authors build the connections between the clipping bias and the second moment of local updates. This initializes a new direction to systematically instruct private learning by connecting the research of variance reduction in distributed optimization.

2. The authors conduct analysis on both convex and non-convex ERM problems with a fairly mild assumption of the bounded stochastic gradient variance.

**Weaknesses:**

1. I understand this is a theoretical paper, but the authors should claim the experimental setup more clearly. It is unclear whether the setting is IID or non-IID. The authors should illustrate the consistency of the empirical support under both IID and non-IID settings, which are the most concern to the FL community.

**Questions:**

Please check above.

**Limitations:**

The authors discuss the limitations of their work in the last section.

---

> ### Author Rebuttal · Authors · 2023-08-09
>
> Thanks for your comments. We apologize for the confusion caused about our experimental setup. Our experiments focus on the application of DP-SGD and DP-LSGD in the centralized setup, which is essentially equivalent to a federated learning model of $n$ nodes (users), each holding a single distinct data point. Our experimental setup, strictly speaking, should belong to the non-i.i.d. case. As our results only assume local updates of a bounded second moment, theoretically they can be applied to study both i.i.d. and non-i.i.d. scenarios allowing data heterogeneity. Also, as mentioned in the global response, our results also capture both centralized DP and local DP, where the only difference is the different noise scale $\sigma$ required. In addition, we want to mention that our released codes can easily handle a more generic data partition (not necessarily one datapoint for each individual) to simulate a generic distributed optimization setup with automatic noise determination. We will consider adding more experiments about local/user-level DP in a truly federated learning setup in a revision. Please feel free to let us know if you have any other questions. Thanks again.

---

### Author Rebuttal · Authors · 2023-08-09

We would like to express our gratitude to all reviewers for their insightful and helpful comments. In this global response, we address the common concerns raised by the reviewers. We begin by describing what we think the three key contributions and technical innovations of this paper are.

The first and foremost contribution is an "explainable" and intuitive theory that characterizes the clipping bias of DP-(L)SGD and why DP-LSGD performs better. We do not merely seek weaker assumptions to produce stronger convergence analysis. Rather, we conduct experiments to compare the statistics of local updates with/out clipping in the first place and try to understand the convergence phenomena of DP-(L)SGD. Subsequently, we formulated a theory to effectively describe these observations. The goal of this paper is not simply to relax the bounded gradient assumption in prior works to the weaker second-moment bounded scenario, though this is technically challenging as explained below. Instead, we try to only use explainable and simulatable terms to describe the convergence and clipping bias. For instance, Definition 4.1 (incremental norm) and Assumption 4.1 are not arbitrarily proposed: DP-LSGD exhibits higher clipping efficiency where the local updates are more concentrated with relatively small incremental norm. This enables better exploitation of the clipping budget, and we show it produces provably reduced clipping bias (Theorems 4.1 and 4.2).

Second, from a technical perspective, this is the first work that characterizes the convergence rate and clipping bias of DP-(L)SGD without assuming globally bounded local updates/gradients. The key challenge in removing this assumption and only assuming a bounded second moment is primarily from that the clipping bias, now, can be unbounded. Indeed, this can be even intractable across iterations if we use existing analysis methods, such as [16,32,33], which must count on a global upper bound. To address this, we develop a tighter analysis on the average case. We show that, with a very careful selection of the step size $\eta$, in expectation, clipped (L)SGD still makes progress towards the minimum. More details are in our reply to Reviewer SLyE. Additionally, for general convex optimization under perturbation, we provide stronger last-iterate convergence (Theorems 3.1 and 4.1) as opposed to the conventional amortized convergence. To our knowledge, this is the first work that does not assume bounded gradient, which could be of independent interest to general last-iterate optimization research, even outside the realm of privacy concerns.

The third aspect lies in the implications and practicality of our results. Our key motivation behind all presented results is to provide useful theory to systematically instruct improvement over DP optimization, especially for deep learning. In particular, we want to point out the essentially similar nature of DP-SGD and federated learning, and many ideas in (non-private) distributed learning can benefit DP-SGD. Clipped DP-SGD, being the most widely-used private optimizer, has undergone extensive study over the past decade, primarily focusing on empirical approaches by optimizing hyper-parameters (e.g., clipping threshold and network architecture). Even state-of-the-art results [36] align with this line. However, the lack of theory to systematically instruct improvement has gradually become the bottleneck for DP deep learning research. In this paper, LSGD, as a natural variance reduction, is the first step in a new direction to use federated learning methods to systematically improve the clipping bias; and as discussed in the paper, once the privacy issues of error feedback/correction methods can be properly addressed, more advanced federated learning acceleration tricks can be used to provably improve DP-SGD. Moreover, for DP practitioners, we want to mention our released code, the first Pytorch platform for DP-LSGD, has competitive running time as that of Opacus, the well-developed DP-SGD simulator.  While due to the higher memory consumption, currently we can only run DP-LSGD on medium neural networks (ResNet20) with a relatively small batchsize (1,000) on our machines.

As for the local stochastic gradient, to give a more clear comparison with DP-SGD, we describe the main algorithm in a form where each node uses the full local gradients rather than stochastic one (since in centralized DP-SGD, it is equivalent to that each user only has one datapoint.) But as briefly mentioned in Section 2, the scenario of applying local stochastic gradients can still be easily captured by our results. Please note that in all our theorems, we always take a generic noise $Q$ (see Equation (2) and (3) in the paper) into account. This $Q$ can be clipping error, DP noise, the stochastic gradient error or even compression/quantification error. In particular, for the stochastic gradients, the sampling noises are independent of zero mean and thus can be merged with the DP noise.

As for the comparison with state-of-the-art non-private LSGD results, we apologize for the confusion caused, but we do provide a fair comparison in the same full gradient setup. For example in Scaffold [30, Theorem 1] we first set $\sigma=0$ and compare.

Correspondingly, since we provide the generic analysis for the convergence of biased iterate, captured by the generic $Q$, we are able to handle any privacy setup, including the centralized DP, the local sample-wise DP, and the strict local DP, where the only difference is their different sensitivities and DP noises required. Thus, in this paper, we do not restrict ourselves to any particular privacy setup but show a unified analysis. We will elaborate on this in the individual responses.

Finally, we have added additional experimental results on the EMNIST dataset and about the relationship among the incremental norm $B$, clipping threshold $c$ and the stepsize $\eta$. Details can be found in the attachment.

---

> ### Author Response · Authors · 2023-08-16
> **Please let us know if we have addressed your concerns**
>
> Dear reviewers,
>          We extend our sincere gratitude to your exceptionally comprehensive comments and invaluable suggestions! As we draw closer to the deadline for the author-reviewer discussion phase, if your schedule permits, could you please read our rebuttals and let us know if we have addressed your questions or concerns? Your insights are pivotal to us and we would highly appreciate your feedbacks. Thank you again!

---

### Decision · Program_Chairs · 2023-09-21

**Decision:**

Reject

**Comment:**

While the paper got scores with wide variation, there were reviewers who raised the concern from a theoretical novelty perspective, and lack of federated benchmarks. While the authors included details about EMNIST experiment in the rebuttal which partially answered the concern about the lack of FL benchmarks, The reviewers and I do not think it still sufficient for the paper to get accepted. Additionally, the reviewers and I think the paper needs to highlight the theoretical novelty more.